# Repeated Random Sampling for Minimizing the Time-to-Accuracy of Learning

**Patrik Okanovic**[1,*] **Roger Waleffe**[2,*] **Vasilis Mageirakos**[1] **Konstantinos E. Nikolakakis**[3]
**Amin Karbasi**[3,4] **Dionysis Kalogerias**[3] **Nezihe Merve Gürel**[5] **Theodoros Rekatsinas**[1,†]
[1]ETH Zürich  [2]University of Wisconsin-Madison  [3]Yale  [4]Google Research  [5]TU Delft

## Abstract

Methods for carefully selecting or generating a small set of training data to learn from, i.e., data pruning, coreset selection, and dataset distillation, have been shown to be effective in reducing the ever-increasing cost of training neural networks. Behind this success are rigorously designed, yet expensive, strategies for identifying the most informative training examples out of large datasets. In this work, we revisit these methods to understand if the additional computational costs associated with such strategies are justified from the perspective of time-to-accuracy, which has become a critical efficiency measure of deep neural network training over large datasets. Surprisingly, we find that many of the recently proposed methods underperform what we call *Repeated Sampling of Random Subsets* (RSRS or RS2), a powerful yet overlooked extension of the standard random baseline that learns from repeatedly sampled data throughout training instead of a fixed random subset. We test RS2 against thirty-two state-of-the-art data pruning and distillation methods across four datasets including ImageNet. Our results demonstrate that RS2 significantly reduces time-to-accuracy, particularly in practical regimes where accuracy, but not runtime, is similar to that of training on full dataset. For example, when training ResNet-18 on ImageNet, with 10% of the dataset each epoch RS2 reaches an accuracy of 66% versus 69% when training with the full dataset. The best competing method achieves only 55% while training $1.6\times$ slower than RS2. Beyond the above meta-study, we discuss the theoretical properties of RS2 such as its convergence rate and generalization error. Our primary goal is to highlight that future works that aim to minimize total training cost by using subset selection, need to consider 1) the total computation cost (including preparing the subset) and 2) should aim to outperform a simple extension of random sampling (i.e., RS2).

## 1 Introduction

Deep learning is continually achieving impressive results, from image classification (He et al., 2016; Dosovitskiy et al., 2020) to speech recognition (Chiu et al., 2018) and natural language processing (Brown et al., 2020; Radford et al., 2019; OpenAI, 2023). Much of this success can be attributed to training large neural networks over datasets with millions or billions of examples (Russakovsky et al., 2015; Gokaslan & Cohen, 2019; Brown et al., 2020; Radford et al., 2021). However, these network and dataset sizes lead to model training that requires weeks or months and yields significant monetary and computational costs (Mindermann et al., 2022; Brown et al., 2020). Such costs nearly prohibit model refinement through hyperparameter or neural architecture search. As a result, there has been an arms race to minimize the required training time to reach a given accuracy, i.e., time-to-accuracy.

To reduce time-to-accuracy, recent works focus on decreasing the amount of training data used for model learning during each epoch. More specifically, given a large, labeled dataset, these works aim to maximize end-model accuracy and minimize runtime when training for multiple rounds, where training within each round is performed only on a small set of examples equal in size to a fraction $r$ of the full dataset. The set of examples used for training at each round can be either chosen once before learning begins or periodically recomputed between rounds based on model updates (e.g., as

---

*Equal contribution. †Currently at Apple.
Correspondence to `patrik.okanovic@inf.ethz.ch` and `waleffe@wisc.edu`.

in Mirzasoleiman et al. (2020); Killamsetty et al. (2021b)). Existing methods in this framework span two main categories: 1) *data pruning* methods which aim to reduce time-to-accuracy by selecting a subset of the most informative examples for training (Welling, 2009; Bachem et al., 2015; Bateni et al., 2014; Chen et al., 2010; Paul et al., 2021; Sorscher et al., 2022); 2) *dataset distillation* methods which generate small sets of synthetic examples to summarize the full dataset (Wang et al., 2018; Yu et al., 2023). We review these methods in Section 2. Briefly, these methods have demonstrated strong competitiveness in minimizing the time-to-accuracy of learning and influenced a large number of subsequent works, including ours. However, several challenges persist in the pursuit of minimizing time-to-accuracy using these methods. One major challenge lies in the time efficiency aspect, as there is a notable overhead associated with subset selection. For example, many methods require pretraining an auxiliary model on the full dataset for a few epochs in order to select the subset, a task which we find can take roughly 250 minutes on ImageNet (Figure 3b) while training itself with $r = 10$ and 30% takes only 400 and 1200 minutes respectively. Even when efforts are made to address this issue, score-based data pruning has fallen short in achieving high accuracy compared to a simple baseline—which randomly samples a static subset of the full dataset once at the beginning of training—particularly in practical regimes (e.g., $r \in [10, 30]\%$) that enable ML practitioners to efficiently perform tasks such as hyperparameter search (Ayed & Hayou, 2023; Sorscher et al., 2022).

With these challenges in mind, in this work, we revisit the random sampling baseline for subset selection and study an intuitive and powerful extension of this method for optimizing the time-to-accuracy of learning. Random sampling is already a competitive baseline due to its ability to select representative data examples for training, thus preventing overfitting. Typically, a static subset of the complete dataset is sampled once before the learning begins (Guo et al., 2022; Park et al., 2022). However, we believe that a stronger instance of this baseline would repeatedly sample data instances at each epoch, as this allows the learner to explore more previously unseen examples throughout training. Random exploration has already proven advantageous for data pruning methods (Ayed & Hayou, 2023), allowing them to calibrate for distribution shift (caused by discarded examples). Moreover, adversarial training has also experienced time-to-accuracy reduction with random exploration (Kaufmann et al., 2022). Surprisingly, however, this method has yet to be established as a data pruning baseline for standard training of deep neural networks. Motivated by this gap, in this work we study *Repeated Sampling of Random Subsets* (RSRS = RS2), a data pruning baseline which samples a subset of data uniformly at random for each training epoch. The main contribution of this paper is a in-depth study of RS2 and a comprehensive evaluation of to what extent it is competitive, and can even outperform, expensive and complex data selection algorithms introduced in the recent literature; when the goal is to minimize total training time including subset selection, i.e., time-to-accuracy. While instantiations of RS2 have been considered for a variety of contexts in recent works (Bartoldson et al., 2023; Safran & Shamir, 2020; Wang et al., 2019; Wu et al., 2020), to our knowledge, this work is the first to show that RS2 surpasses *all* state-of-the-art data pruning and distillation methods in accuracy and end-to-end runtime across a wide range of subset sizes. This finding makes RS2 a strong baseline to beat for minimizing time-to-accuracy. In Section 3, we provide a detailed explanation of RS2, and in Section 4 we discuss its theoretical properties.

We extensively evaluate the time-to-accuracy of RS2 and compare it against 24 proposed data pruning and eight dataset distillation methods from the literature (Section 5). We find that RS2 outperforms existing methods with respect to runtime and accuracy across varying subset selection sizes and datasets, including CIFAR10, CIFAR100, ImageNet30, and ImageNet. For example, when training ResNet-18 with $r = 10\%$ on ImageNet, RS2 yields a model with 66% accuracy, 11 points higher than the next-best method and only 3.5 points less than training with the entire dataset every round. Yet, RS2 reaches this accuracy $9\times$ faster than standard full-dataset training and $1.6\times$ faster than the next-best method. Finally, we present an extension of RS2 beyond supervised learning by benchmarking its performance on self-supervised pretraining of GPT2 (Radford et al., 2019). Our evaluation shows that RS2 can potentially be a competitive baseline in this setting. Overall, our findings highlight that future works that aim to minimize neural network training cost need to consider both the end-model accuracy and runtime, including subset selection, and should aim to outperform RS2.

## 2 PRELIMINARIES

We first present a unified framework for the problem of reducing time-to-accuracy by training on less data each epoch and then review existing data pruning and distillation methods.

**Problem Statement**    Given a large, labeled dataset $S = \{\mathbf{x}_i, y_i\}_{i=1}^N$, where each training example consists of an input feature vector $\mathbf{x}_i$ and a given ground truth label $y_i$, our goal is to minimize runtime and maximize accuracy when training for $X$ rounds, with the training of each round performed on a set of examples $S'$ with size $|S'| = r \cdot |S|$ for $r \in (0, 1]$.

*We highlight two points:* First, it is assumed that $X$ is chosen such that training proceeds for the same number of rounds as when training on the full dataset, otherwise the computational benefits are reduced (e.g., $r = 50\%$ with $X = 200$ is the same amount of computation as $r = 100\%$ with $X = 100$). Second, note that the subset $S'$ may be static (e.g., as in Sorscher et al. (2022)) or vary across rounds (e.g., as in Mirzasoleiman et al. (2020); Killamsetty et al. (2021b)). Given the primary goal of minimizing time-to-accuracy, either choice is valid so long as the time to generate the subset $S'$ at each round is included in the overall runtime.

**Related Work**    To minimize time-to-accuracy, *data pruning* methods attempt to find a subset (also called a *coreset*) of informative examples $S' \subset S$ (Guo et al., 2022; Park et al., 2022). Numerous metrics have been proposed to quantify importance: Uncertainty based methods such as Least Confidence, Entropy, and Margin (Sachdeva et al., 2021) assume examples with lower confidence will have higher impact on training. Loss and error based methods operate on a similar principle (e.g., Forgetting Events (Toneva et al., 2018), GraNd, EL2N (Paul et al., 2021), and others (Bachem et al., 2015; Munteanu et al., 2018; Dasgupta et al., 2019; Liu et al., 2021)). Other techniques for subset selection, such as CRAIG (Mirzasoleiman et al., 2020) and GradMatch (Killamsetty et al., 2021a), focus on gradient matching, where the goal is to construct a subset of examples such that a weighted sum of the model gradients on the subset matches the overall gradient on the full dataset. A different class of methods focuses on feature geometry for data subset selection (e.g., Herding (Welling, 2009; Chen et al., 2010), K-Center Greedy (Sener & Savarese, 2018), and prototypes (Sorscher et al., 2022)). Additional data pruning algorithms attempt to find the training examples closest to the decision boundary (e.g., Adversarial Deepfool (Ducoffe & Precioso, 2018) and Contrastive Active Learning (Liu et al., 2021)), pose subset selection as a bilevel optimization problem (e.g., Retrieve (Killamsetty et al., 2021c) and Glister (Killamsetty et al., 2021b)), or connect subset selection to maximization of a submodular function (e.g., GraphCut, Facility Location, and Log Determinant (Iyer et al., 2021)). Active learning methods (which aim to minimize labeling cost by selecting a subset given a large unlabeled dataset), can also be used in the presence of a labeled dataset when the goal is to reduce time-to-accuracy (Park et al., 2022). We refer the reader to recent surveys (Guo et al., 2022) for more details on the above methods and for comparisons between them.

In contrast to data pruning which assumes $S'$ to be a subset of $S$, *dataset distillation* methods use $S$ to generate a small set of synthetic examples $S'$ that aims to summarize $S$. Dataset distillation methods can be split into three groups: 1) Performance matching methods (Wang et al., 2018; Deng & Russakovsky, 2022) aim to optimize the synthetic examples in $S'$ such that models trained on $S'$ achieve the lowest loss on the original data $S$. 2) Parameter matching techniques (Zhao et al., 2021; Lee et al., 2022; Kim et al., 2022; Cazenavette et al., 2022) focus instead on matching the parameters of a network trained on $S'$ with those of a network trained on $S$ by training both models for a number of steps. 3) Finally, the distribution matching approach (Zhao & Bilen, 2023) to dataset distillation attempts to obtain synthetic examples in $S'$ such that the distribution of $S'$ matches the distribution of $S$. We refer the reader to (Yu et al., 2023) for a detailed survey on dataset distillation.

## 3    RS2: Repeated Random Sampling to Reduce Time-to-Accuracy

We describe the RS2 framework in the context of data pruning and discuss how it yields efficient training by reducing the amount of training data used at each round of model learning.

**Repeated Sampling of Random Subsets (RS2)**    As discussed in Section 2, we assume access to a large, labeled dataset $S$ and aim to reduce time-to-accuracy by training for $X$ rounds, with the training of each round performed on a subset $S' \subset S$ (of size $|S'| = r \cdot |S|$). *We define RS2 as follows: at each round, sample $S'$ randomly from $S$ (Algorithm 1).* Thus, the only differences between RS2 and existing SOTA data pruning methods (Section 2) are: 1) the subset is sampled randomly rather than based on example importance and 2) the subset is resampled before each round, compared to many prior methods which opt to select only a static subset once at the beginning. We next discuss two variants of the sampling strategy and the importance of appropriate learning rate scheduling.

**RS2 With Replacement**     The simplest version of RS2 samples $S'$ with replacement across rounds—sampling can be stratified. This means that examples included in the subset of previous rounds are replaced in $S$ and eligible to be resampled when constructing $S'$ for the current round, i.e., $S'$ is always constructed by sampling uniformly from all examples in $S$ (Algorithm 1, Lines 7-8).

**RS2 Without Replacement**     A second variant of RS2 samples $S'$ without replacement across rounds. That is, examples in $S$ that have been included in the subset during previous rounds are not considered when sampling $S'$ for the current round. This continues until all examples from $S$ have been included in $S'$ at some round, at which point the process repeats. RS2 without replacement can be implemented as follows (Algorithm 1, Lines 3-6): Given a random permutation of the full dataset $S = \{\mathbf{x}_i, y_i\}_{i=1}^N$, we select the first $rN$ examples as the subset for the first round, then the next $rN$ examples as the subset for the second round, and so on. After iterating over the full dataset, we generate a new permutation and repeat. *Observe that as a consequence of sampling $S'$ without replacement,*

---

**Algorithm 1** RS2 General Algorithm

**Require:** Dataset $S = \{\mathbf{x}_i, y_i\}_{i=1}^N$, selection ratio $r \in (0, 1]$, batch size $b$, initial model $w^0$, $X$ rounds, sampling mode $mode = $ 'with repl.' or 'w/o repl.'
1: $T \leftarrow \lceil N/b \rceil$;  $t \leftarrow 1$
2: **for** round $j = 1$ to $X$ **do**
3:     **if** $mode == $ 'w/o repl.' **then**
4:         **if** $t\%T == 0$ **then**
5:             $shuffle(S)$          ▷ Shuffle full dataset
6:         $S' \leftarrow S[(j-1) \cdot rN : j \cdot rN]$
7:     **if** $mode == $ 'with repl.' **then**
8:         $S' \leftarrow randomly\_sample\_subset(S, \ r)$
9:     **for** $k = 1$ to $r \cdot T$ **do**
10:        batch $m \leftarrow S'[(k-1) \cdot b : k \cdot b]$
11:        $w^t \leftarrow train(w^{t-1}, \ m)$;  $t \leftarrow t + 1$
    **return** $w^t$

---

*this version of RS2 across $X$ rounds is equivalent to training on the full dataset $S$ for $r \cdot X$ rounds. Thus, RS2 without replacement can also be viewed as training on the full dataset for fewer epochs (with an adjusted learning rate as described next).*

**RS2 Hyperparameters**     For both RS2 variants, we assume that training proceeds using the same hyperparameters (e.g., batch size, etc.) as those used when training on the full dataset with one exception: the learning rate schedule. The reason for this is that state-of-the-art training procedures often slowly decay the learning rate after each SGD step, as shown in Figure 1. We include a vertical line showing the number of SGD iterations when running RS2 for $r = 10\%$—data pruning with RS2 (or any method)

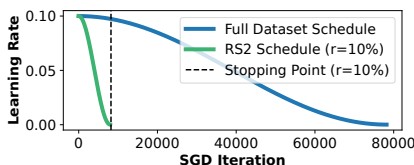

Figure 1: Learning rate schedules on CI-FAR10 with and without data pruning.

leads to fewer SGD iterations (we train for the same $X$ rounds, but on fewer examples per round). Thus, if the pruning method retains the same learning rate schedule, the learning rate will not decay enough to achieve high end-model accuracy. For example, on CIFAR10 with $r = 10\%$, RS2 with the full dataset schedule reaches just $83.9\%$ accuracy compared to $95.5\%$ when training without pruning. As such, we train both RS2 variants with the full dataset learning rate schedule adapted to decay faster, inversely proportional to the subset size $r$ (e.g., green line in Figure 1). This is standard across existing data pruning methods (Guo et al., 2022; Smith & Topin, 2019). In this case, on CIFAR10 with $r = 10\%$, RS2 reaches $89.7\%$ and $91.7\%$ with and without replacement respectively. While RS2 without replacement generally reaches the highest accuracy of the two variants, we do not claim that one is strictly better. We refer the reader to existing works regarding this question (Haochen & Sra, 2019; Lu et al., 2022; De Sa, 2020).

## 4    THEORETICAL PROPERTIES OF RS2

With time-to-accuracy in mind, we now study the relevant theoretical properties of RS2. We first comment on its convergence rate and then provide its generalization error.

**RS2 Convergence Rate**     As familiar readers may identify, RS2 without replacement, under the assumption of nonconvex loss with bounded, $\beta$-Lipschitz gradients, converges to an approximate critical point of the expected loss (Wang & Srebro, 2019). Following the convergence result for mini batch SGD with Nesterov's accelerated gradient update (Ghadimi & Lan, 2016), the convergence rate of RS2 without replacement can be shown to have a scaling factor $r$ in front of the total number of iterates compared to the full dataset SGD convergence rate (Ghadimi & Lan, 2016), while the bound remains consistent with respect to all the other parameters (we defer to Appendix E for details).

**RS2 Generalization Error** We now provide an upper bound on the generalization error of RS2. For this analysis, we relax the update rule from Algorithm 1 to a standard gradient update without momentum. Recall that as $r$ decreases, RS2 results in a smaller total number of gradient steps after $X$ rounds compared to $r = 1$. While this may lead to an increase in optimization error, the generalization error is expected to be smaller than that of the full dataset schedule (shorter training time gives a smaller generalization error). This phenomenon has been characterized rigorously in prior works (Hardt et al., 2016) for vanilla SGD with batch size $b = 1$, however it does not directly apply to larger mini batch sizes and general selection rules. As such, we show an extension of known generalization error bounds that also holds for RS2 with mini batch size $b$. Before we proceed, we first introduce some notation for brevity. We define the training dataset $S \triangleq (z_1, z_2, \ldots, z_N)$, for which $z_i \triangleq (\mathbf{x}_i, y_i)$ for $i \in \{1, \ldots, N\}$ and the (empirical) loss $l(w) \triangleq \frac{1}{N} \sum_{i=1}^{N} f(w, z_i)$, where $f : \mathbb{R}^d \times \mathcal{Z} \to \mathbb{R}^+$. Let $z_1, z_2, \ldots, z_N, z$ be i.i.d random variables with respect to an unknown distribution $\mathcal{D}$. Then for any stochastic algorithm $A$ with input $S$, and output $A(S)$, the generalization error $\epsilon_{\text{gen}}$ is defined as the difference between the empirical and population loss (Hardt et al., 2016):

$$\epsilon_{\text{gen}}(f, \mathcal{D}, A) \triangleq \mathbb{E}_{S,A,z}\big[f(A(S), z)\big] - \mathbb{E}_{S,A}\Big[\frac{1}{N} \sum_{i=1}^{N} f(A(S), z_i)\Big]. \tag{1}$$

We now proceed with an upper bound on the generalization error of RS2. The next result follows from recent work (Nikolakakis et al., 2023), and applies to RS2 without momentum and batch size $b$.

**Theorem 4.1** (Generalization error of standard gradient RS2). *Let the function $f$ be nonconvex, $L_f$-Lipschitz and $\beta_f$-smooth. Then the generalization error of the standard gradient RS2 algorithm with a decreasing step-size $\eta_t \leq C/t$ (for $C < 1/\beta_f$), is bounded as:*

$$|\epsilon_{\text{gen}}(f, \mathcal{D}, \text{RS2})| \leq \frac{1}{N} \cdot 2Ce^{C\beta_f} L_f^2 (r \cdot T \cdot X)^{C\beta_f} \min\Big\{1 + \frac{1}{C\beta_f}, \log(e \cdot r \cdot T \cdot X)\Big\}. \tag{2}$$

The proof of Theorem 4.1 is deferred to Appendix F. Observe that, as for the convergence rate, the generalization error of RS2 remains the same as for that of the full dataset (Nikolakakis et al., 2023), except that the number of iterates for RS2 is scaled by $r$. We note that Theorem 4.1 applies to both RS2 with and without replacement—it relies on the fact that the batch at each iteration is selected *non-adaptively and in a data-independent fashion*. However, most exisiting data pruning methods adopt data-dependent strategies, which recent work (Ayed & Hayou, 2023) has shown may worsen generalization due to discarding many training examples, resulting in inferior performance compared to random sampling. This theoretical insight (that data-independent sampling allows for improved generalization due to selecting diverse, unbiased samples), is the reason RS2 outperforms existing methods. Our conducted experiments in Section 5 support this hypothesis.

## 5 EVALUATION

We evaluate RS2 on four common benchmarks for supervised learning and compare against data pruning and distillation methods. We show that:

1. Across a wide range of selection ratios, RS2 reaches higher accuracy than all existing methods.
2. For a given selection ratio, RS2 also trains the fastest, and thus has the fastest time-to-accuracy.
3. In the presence of noisy labels, RS2 is the most robust data pruning method; It achieves the highest end-model accuracy and lowest relative drop in performance vs. training on the clean dataset.

We also end this section by discussing an initial extension of RS2 beyond conventional supervised learning to self-supervised training settings for text.

### 5.1 EXPERIMENTAL SETUP

We summarize the setup used in the experiments. More details can be found in Appendix B.

**Datasets, Models, and Metrics** We benchmark RS2 against baseline methods using CIFAR10 (Krizhevsky et al., 2009), CIFAR100 (Krizhevsky et al., 2009), ImageNet30 (a subset

---

Source code: `https://github.com/PatrikOkanovic/RS2`

of ImageNet) (Hendrycks et al., 2019), and ImageNet (Russakovsky et al., 2015) itself. We train ResNet models (He et al., 2016) representative of modern state-of-the-art convolutional neural networks. We use the same models/datasets as those used in the recent published works on SOTA data pruning methods (Guo et al., 2022; Killamsetty et al., 2021a; Mirzasoleiman et al., 2020; Paul et al., 2021; Sachdeva et al., 2021; Sorscher et al., 2022; Toneva et al., 2018); this choice enables a direct comparison with prior work. For all experiments we measure subset selection overhead, overall training time (including the total time for subset selection across all rounds and the total training time on selected subsets), and end-model accuracy. We believe that the consistency of our findings across all models/datasets provides sufficient experimental evidence that RS2 is a strong baseline that has been overlooked in literature.

**Baselines** We compare RS2 against 24 data pruning methods and eight dataset distillation methods from the literature. A full list and their abbreviations can be found in Appendix B. All baselines are used for the smallest dataset (i.e., CIFAR10), but some methods do not scale to larger datasets (e.g., ImageNet). We utilize existing open-source implementations and results where applicable (Park et al., 2022; Guo et al., 2022) and implement RS2 within the same code for equal comparison.

**Training Details** We use standard hyperparameters for each dataset from prior works known to achieve high accuracy. We use the same hyperparameters for all methods where applicable (e.g., batch size, initial learning rate, number of training rounds, etc.). This means that, for the same subset selection size $r$, RS2 and all baseline methods train for the same number of SGD iterations using the same learning rate schedule; the only difference is *how* the examples in each mini batch are generated. More details are provided in Appendix B.3.

## 5.2 END-TO-END DATA PRUNING EXPERIMENTS

We discuss end-to-end comparisons of RS2 with data pruning baselines on supervised learning benchmarks. Results are shown in Figures 2-3 and Table 1.

**Accuracy** We first focus on end-model accuracy of RS2 compared to existing methods. We include comparisons both to baselines which we modify to re-select the subset each round and to the original methods which select a fixed subset—this enables us to compare directly to many recent works that aim to reduce total training cost (i.e., time-to-accuracy) through data pruning and propose to select static subsets (e.g., Killamsetty et al. (2021a); Paul et al. (2021); Sachdeva et al. (2021); Toneva et al. (2018)). Evaluating the performance of data pruning baselines in both static and repeated sampling modes allows us to highlight the importance of 1. selecting subsets *randomly* (RS2 vs. baselines in per-round sampling mode) and 2. selecting subsets *repeatedly* (RS2 vs. baselines in static mode).

*1. Accuracy Vs Baselines with Static Subset Selection* In Figure 2 we show the end-model accuracy of RS2 compared to existing methods on CIFAR10 and ImageNet for varying selection ratios. We use the combined baseline methods and setting from recent studies (Guo et al., 2022; Park et al., 2022) together with newer prototype-based data pruning methods (Sorscher et al., 2022). We include a further discussion of results and the exact accuracies for these Figures in Tables 3 and 4 in Appendix C.1. The repeated sampling of RS2 leads to accuracy improvements over existing methods: For example, on CIFAR10 with 10% of the data each epoch, RS2 without replacement achieves 91.7% accuracy while the next closest baseline reaches just 86.1%. Similar results hold on CIFAR100 and ImageNet30 (Appendix C.2) as well as on CIFAR10 with ViT architectures rather than ResNets (Appendix C.4). Figure 2b shows that RS2 also outperforms existing methods on the much larger ImageNet dataset: RS2 end-model accuracy with $r = 10\%$ is 66% while the next closest baseline trains to only 55%. Moreover, the end-model accuracy of RS2 is actually on par with training on the full dataset for non-trivial selection ratios (e.g., $r \in [10, 30]\%$), offering a potential practical solution to reduce the cost of training in some applications (see also runtime reductions below).

*2. Accuracy Vs Baselines with Repeated Subset Selection* Next, we extend baselines to also perform repeated sampling. Our goal is to examine if the prior observations are attributed only to the fact that RS2 performs repeated sampling while the above baselines do not. We implement repeated sampling for baselines in two different ways. For a given baseline $M$ 1) *if* $M$ computes a numerical importance score for all training examples to select a subset, we compute those scores once at the beginning of training and view them as a categorical distribution over the dataset. We then define $M$-RS as follows: at each round, we *resample* the subset according to the categorical distribution. 2) We define $M$-RC as follows: In between *every* round, we use the weights of the current model being trained to either

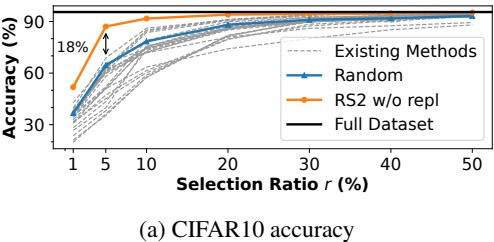 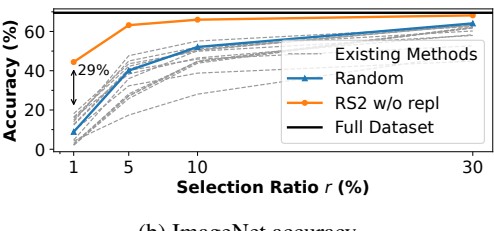

| (a) CIFAR10 accuracy | (b) ImageNet accuracy |

Figure 2: Accuracy achieved by data pruning methods (baselines select static subsets) when training ResNet-18 on CIFAR10 and ImageNet. RS2 outperforms existing methods.

Table 1: Accuracy achieved by data pruning methods with per-round sampling when training ResNet-18 on CIFAR10. The training subset is update for all methods after each round, either by resampling from a static example importance distribution (RS, left) or by recomputing subsets based on the model weights after each epoch (RC, right). Repeated Sampling of Random Subsets (RS2) outperforms repeated sampling based on example importance. Best method bolded; Next best underlined.

| Selection Ratio ($r$) | 5% | 10% | 30% | Selection Ratio ($r$) | 5% | 10% | 30% |
|---|---|---|---|---|---|---|---|
| CD-RS | - | - | - | CD-RC | 75.2±2.2 | 83.1±0.7 | 87.5±0.2 |
| Herding-RS | - | - | - | Herding-RC | 30.1±2.6 | 40.6±8.4 | 81.0±0.9 |
| K-Center Greedy-RS | - | - | - | K-Center Greedy-RC | 78.1±1.5 | 82.3±0.5 | 86.3±0.4 |
| Least Confidence-RS | 67.6±5.1 | 83.4±4.9 | 93.7±0.4 | Least Confidence-RC | 44.8±12 | 76.7±3.9 | 88.3±0.3 |
| Entropy-RS | 85.2±0.9 | 89.8±0.4 | 94.4±0.3 | Entropy-RC | 41.4±6.9 | 78.4±2.9 | 86.9±0.1 |
| Margin-RS | 84.3±2.7 | 90.4±1.0 | 94.4±0.2 | Margin-RC | 79.7±1.4 | 82.8±1.4 | 86.8±0.2 |
| Forgetting-RS | 81.9±3.1 | 88.3±2.4 | 94.0±0.1 | Forgetting-RC | 28.7±0.8 | 40.7±6.5 | 78.8±4.3 |
| GraNd-RS | 86.2±2.1 | 90.1±0.9 | **94.5±0.1** | GraNd-RC | 15.5±1.8 | 24.1±6.0 | 75.2±5.0 |
| CAL-RS | 81.1±3.0 | 86.6±0.7 | 93.3±0.1 | CAL-RC | 66.7±1.7 | 74.5±0.8 | 84.8±0.4 |
| Craig-RS | - | - | - | Craig-RC | 70.3±13 | 80.3±0.8 | 85.5±0.3 |
| Glister-RS | - | - | - | Glister-RC | 72.5±0.6 | 81.4±0.7 | 86.6±0.5 |
| SP-Easy-RS | 84.0±4.3 | 88.4±0.1 | 93.6±0.3 | SP-Easy-RC | - | - | - |
| RS2 w repl (stratified) | 86.6±0.5 | 89.8±0.4 | **94.5±0.1** | RS2 w repl (stratified) | 86.6±0.5 | 89.8±0.4 | **94.5±0.1** |
| RS2 w/o repl | **87.1±0.8** | **91.7±0.5** | 94.3±0.2 | RS2 w/o repl | **87.1±0.8** | **91.7±0.5** | 94.3±0.2 |

*recompute* example importance scores—the examples with the highest scores are then selected for the subset at the next round—or directly re-select the subset for training. We do not present -RC results for baselines which generate subsets independently of the model weights (as the subset would remain static). Full implementation details for our repeated sampling baselines are included in Appendix B.4. Note that the latter set of methods (RC) are unlikely to be able to improve the efficiency of training as they generally require computing the model forward pass for every example between each round (to compute importance scores). Results on CIFAR10 (for computational considerations we do not run these methods on ImageNet) are shown in Table 1. While updating the subset for existing methods each round improves their accuracy, RS2 still reaches the highest end-model accuracy.

**Training Time** We now study the training time (time-to-accuracy) of RS2 compared to existing methods on CIFAR10 and ImageNet. We train all methods from scratch on NVIDIA 3090 GPUs and use all baselines which do not give GPU out-of-memory. For the experiments reported here, we run baseline as they were originally proposed (i.e., with static subset selection). This allows us to quantify the overhead of selecting a *single* subset with existing methods compared to repeatedly selecting many random subsets with RS2. We show the time-to-accuracy on CIFAR10 in Figure 3a and on ImageNet in Figure 3b using $r = 10\%$ for both datasets.

More time-to-accuracy results are included in Appendix C.6 and C.7. We report the total time for subset selection on CIFAR10 for baselines with static subset selection in Appendix Table 8 and for baselines which utilize per-round sampling in Appendix Table 9. We also include the time-to-accuracy measurements on CIFAR10 and ImageNet for different selection ratios in Appendix Tables 10-14.

Figures 3a and 3b show that RS2 provides the fastest time-to-accuracy when compared to previous data pruning methods. Note that the repeated subset selection in RS2 leads to negligible overhead compared to training on a static random subset (Figure 3) and to the total training time: For example, the total subset selection time for RS2 on CIFAR10 with $r = 10\%$ is less than one second, yet the total runtime is 750 seconds. Existing methods, however, are primarily limited by the fact that they require *pretraining* an auxiliary model on the full dataset for a few epochs in order to rank example importance: On ImageNet the fastest baseline begins training after 250 minutes, yet training itself only requires 400 minutes. Even if the pretraining overhead is amortized by fixing the subset for the

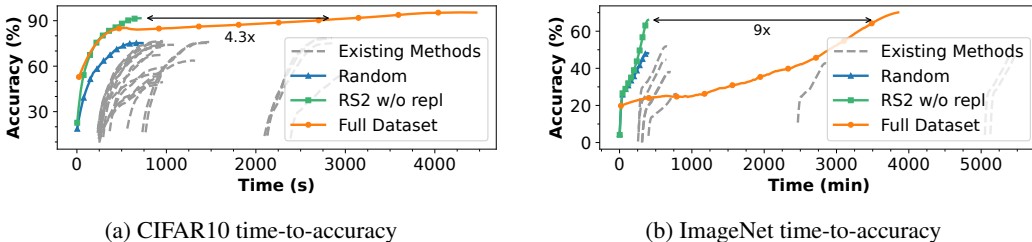

(a) CIFAR10 time-to-accuracy          (b) ImageNet time-to-accuracy

Figure 3: Time-to-accuracy for RS2 vs. existing data pruning methods (with static subset selection), a static random subset, and standard training on the full dataset. We use a selection ratio of $r = 10\%$. RS2 is both the fastest and highest accuracy data pruning method.

Table 2: Accuracy achieved by dataset distillation methods, RS2, and Random data pruning when training a ConvNet model. We select the specified number of images per class (Img/Cls) corresponding to the given selection ratio on the full dataset. Best method bolded. Next best underlined.

| | Img/Cls | Ratio % | Random | Dataset Distillation Methods | | | | | | | | RS2 w/ repl | Full Dataset |
| | | | | DD | LD | DC | DSA | DM | CAFE | CAFE+DSA | TM | | |
|---|---|---|---|---|---|---|---|---|---|---|---|---|---|
| CIFAR10 | 1 | 0.02 | 14.4±2.0 | - | 25.7±0.7 | 28.3±0.5 | 28.8±0.7 | 26.0±0.8 | 30.3±1.1 | 31.6±0.8 | 46.3±0.8 | **54.7±0.5** | |
| | 10 | 0.2 | 36.8±1.2 | 36.8±1.2 | 38.3±0.4 | 44.9±0.5 | 52.1±0.5 | 48.9±0.6 | 46.3±0.6 | 50.9±0.5 | 65.3±0.7 | **72.7±0.1** | 84.8±0.1 |
| | 50 | 1 | 43.4±1.0 | - | 42.5±0.4 | 53.9±0.5 | 60.6±0.5 | 63.0±0.4 | 55.5±0.6 | 62.3±0.4 | 71.6±0.2 | **76.5±0.3** | |
| CIFAR100 | 1 | 0.2 | 4.2±0.3 | - | 11.5±0.4 | 12.8±0.3 | 13.9±0.3 | 11.4±0.3 | 12.9±0.3 | 14.0±0.3 | 24.3±0.3 | **37.4±0.4** | |
| | 10 | 2 | 14.6±0.5 | - | - | 25.2±0.3 | 32.3±0.3 | 29.7±0.3 | 27.8±0.3 | 31.5±0.2 | 40.1±0.4 | **43.9±0.4** | 56.2±0.3 |
| | 50 | 10 | 30.0±0.4 | - | - | - | 42.8±0.4 | 43.6±0.4 | 37.9±0.3 | 42.9±0.2 | **47.7±0.2** | 44.6±0.3 | |
| Tiny ImageNet | 1 | 0.2 | 1.4±0.1 | - | - | - | - | 3.9±0.2 | - | - | 8.8±0.3 | **23.5±0.2** | |
| | 10 | 2 | 5.0±0.2 | - | - | - | - | 12.9±0.4 | - | - | 23.2±0.2 | **27.4±0.1** | 37.6±0.4 |
| | 50 | 10 | 15.0±0.4 | - | - | - | - | 24.1±0.3 | - | - | 28.0±0.3 | **28.6±0.4** | |

remaining rounds, or by resampling from the importance distribution after each round (-RS baselines in Table 1), the initial overhead of these methods is still orders of magnitude higher than the total overhead of RS2 across all rounds (e.g., Table 8-9). Moreover, Figures 3a and 3b highlight the practical potential of RS2 to reduce the computational cost of training high-accuracy models: For CIFAR10, RS2 reaches 91.7% accuracy 4.3× faster than standard training on the full dataset, while for ImageNet, RS2 reaches 66% accuracy 9× faster than standard training.

**Takeaway** The above results show that RS2 outperforms existing data pruning methods with respect to end-model accuracy. RS2 also has the lowest subset selection overhead resulting in the best time-to-accuracy across small (CIFAR10) and large (ImageNet) datasets.

## 5.3 COMPARISON TO DATASET DISTILLATION

We compare RS2 to dataset distillation methods which generate small sets of synthetic examples. Our experiments on CIFAR10, CIFAR100, and Tiny ImageNet (for computational reasons) are shown in Table 2. We use no data augmentation for these experiments due to small selection ratios. We train ConvNet models for consistency with existing dataset distillation evaluations. While dataset distillation methods generally outperform data pruning methods (e.g., in Table 2 a static random subset on CIFAR10 with $r = 1\%$ reaches 43.4% accuracy while dataset distillation methods reach up to 71.6%), they have drawbacks. The synthetic sets are model-specific and computationally expensive to generate. For instance, the best performing method, Trajectory Matching (TM), requires 133, 317, and 433 minutes to generate 50 images per class on CIFAR10, CIFAR100, and Tiny ImageNet, respectively. In comparison, RS2 requires just seven, 33, and 187 minutes for end-to-end training in these settings. Yet RS2 outperforms Trajectory Matching with respect to end-model accuracy for eight of the nine selection ratio/dataset combinations in Table 2.

## 5.4 EXPERIMENTAL EXTENSIONS

We consider two practical extensions beyond the supervised benchmarks considered in the data pruning literature: We aim to 1) provide evidence of RS2's robustness against noisy labels compared to other data pruning methods and 2) explore its possible application to unsupervised learning settings.

**Robustness of RS2 to Noisy Labels** One aspect that is usually underexplored in literature is the robustness of standard data pruning baselines, which assume clean, noise-free labels. Here, we aim to challenge this assumption and compare standard pruning methods against RS2 in a practical scenario where this assumption fails. We present a detailed discussion of our experiments in Appendix C.8

but summarize our main findings here: For this experiment, we use CIFAR10 and randomly flip $p$ percentage of the labels in the dataset—we vary $p$ in $\{10\%, 30\%, 50\%\}$. Different data pruning methods are used over this noisy dataset and we measure end-model accuracy and raw accuracy drop compared to no noise ($p = 0$). In summary, we find that RS2 achieves higher end-model accuracy in the presence of noisy labels compared to existing data pruning methods. For example, with 30% of the training examples mislabeled, RS2 without replacement achieves 74.4% accuracy while the next closest baseline—our modified per-round subset selection version of supervised prototypes with easy examples (SP-Easy-RS)—achieves just 63.4%. Moreover, RS2 is generally the most robust method in that it suffers the lowest relative drop in performance when presented with noisy labels. All results are reported in Appendix Table 15.

**RS2-based Pretraining of Language Models**     In the supervised training setting we find that RS2 can lead to measurable end-to-end cost reductions without significant drop in accuracy. We now extend our evaluation to unsupervised training aiming to understand if a similar finding holds. To this end, we focus on pretraining of large language models (LLMs), a cost intensive procedure. We focus on the popular GPT architecture and specifically, explore if repeated random sampling (i.e., RS2) can help reduce the cost of training GPT2 (Radford et al., 2019) on OpenWebText (Gokaslan & Cohen, 2019) from scratch, without significant loss in end-model quality. A detailed discussion can be found in Appendix C.9. Here, we present a summary of our findings.

We consider the standard metrics of *accuracy* and *perplexity* over the LAMBADA (Paperno et al., 2016) and WikiText103 (Merity et al., 2016) benchmarks. In addition, we measure the monetary cost required to train GPT2—we choose to train GPT2 due to monetary restrictions. We compare RS2-based training of GPT2 against 1) training over the full dataset, and 2) training over a random sample of the dataset (a basic baseline considered as a lower bound). We find that for LAMBADA, using a selection ratio of $r = 30\%$, RS2-based training of GPT2 yields a relative accuracy drop of only $2.8\%$ and a perplexity increase of only $0.5\%$, while offering a $3\times$ cost reduction from \$5,200, when the full dataset is used for training, to \$1,560. For comparison, a static random sample, yields an accuracy drop of $4.2\%$ and a perplexity increase of $3.3\%$. Details are reported in Appendix Table 16.

We believe that this extension highlights a promising direction for reducing the cost of LLM training, and that RS2 offers a competitive baseline for future unsupervised data pruning. Moreover, our findings seem to corroborate the very recent findings of Marion et al. (2023), which identify that static random sampling is a competitive data pruning baseline for LLM pretraining, while also offering a stronger baseline. Finally, we believe that repeated random sampling is complementary to methods that aim to reduce LLM training costs by focusing on high-quality data points (Zhou et al., 2023; Gunasekar et al., 2023). A more extensive study of data pruning for LLM training that will combine the aforementioned ideas between selecting high-quality examples in a randomized manner (to promote diversity) is definitely an exciting future direction.

## 6   CONCLUSION

We showed that training on random subsets repeatedly sampled (RS2) from a large dataset results in reduced runtime and higher end-model accuracy when compared to existing data pruning and distillation methods. While RS2 may provide a practical solution for reducing time-to-accuracy, e.g., for hyperparameter or neural architecture search, we also hope that our findings serve as a baseline for future research to minimize time-to-accuracy through data subset selection. RS2 is likely to be a weaker baseline when the goal is different from minimizing time-to-accuracy, e.g., if the goal is to reduce dataset storage overhead or to minimize the cost of labeling examples for training by selecting a subset from a large, unlabeled dataset. We believe that other methods we consider (e.g., active learning methods (Park et al., 2022; Ren et al., 2021)) are more practical for these settings.

Finally, we are excited for future work to address the next questions: How can we further close the gap between RS2 and training on the full dataset? Interesting sub directions to answering this question include: 1) further study of importance sampling-based methods for reducing time-to-accuracy and 2) improving the subset training procedure (independent of the method) to benefit the end-model accuracy. The key issue with the latter is that training on a subset results in fewer total SGD iterations when compared to training on the full dataset for the same number of rounds. Can we overcome this limitation of data pruning without eliminating the runtime benefits? We believe that new research into these questions can enable further reductions in time-to-accuracy.

**Ethics statement.** We do not see potential ethical issues with the contributions of the paper. In contrast, we evaluate a performant extension of random sampling for environmentally-friendly training and offer an extensive comparison of it to the existing data pruning methods.

**Reproducibility statement.** The reproducibility of RS2 spans theoretical and experimental perspectives. We provide complete proofs of all the theoretical results in appendices. We upload the source codes for implementing RS2 and reproducing experiment results as supplementary materials.

### ACKNOWLEDGMENTS

We would like to thank the anonymous reviewers for their constructive comments on our paper. This work was supported by DARPA under grant ASKEM HR001122S0005. The U.S. Government is authorized to reproduce and distribute reprints for Governmental purposes notwithstanding any copyright notation thereon. Any opinions, findings, and conclusions or recommendations expressed in this material are those of the authors and do not necessarily reflect the views, policies, or endorsements, either expressed or implied, of DARPA or the U.S. Government. This work is also supported with funding from the Wisconsin Alumni Research Foundation. Amin Karbasi acknowledges funding in direct support of this work from NSF (IIS-1845032), ONR (N00014- 19-1-2406), and the AI Institute for Learning-Enabled Optimization at Scale (TILOS).

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

APPENDIX

## A  A MOTIVATING EXPERIMENT FOR REPEATEDLY SAMPLING RANDOM SUBSETS

We have shown in the main body of the paper that Repeated Sampling of Random Subsets (RS2) allows for faster training and more accurate models when compared to existing data pruning and dataset distillation techniques. In this section, we discuss a simple experiment that helped motivate our work.

Existing data pruning methods are primarily based on the intuition that a small subset $S'$ of 'difficult' (Toneva et al., 2018; Paul et al., 2021) (or sometimes 'easy' (Sorscher et al., 2022)) examples contained in the full dataset $S$ are close to (far from) the decision boundary and thus likely to be the most informative for learning. During our initial investigation into data pruning methods, we empirically studied this intuition. Calculating the distance between a training example and the decision boundary, however, can be challenging because the decision boundary is not known until training completes, and because the location of the decision boundary in high dimensional space can be computationally intensive to compute. Thus, we consider the following proxy measurement: To decide whether a training example $x$ is close to the decision boundary, we find the nearest neighbor (e.g., $L_2$ distance) from the full dataset and check whether it has the same label as $x$. If not, then the decision boundary in the input feature space must be between the two points (i.e., they are 'close' to the boundary).

We evaluated the above proxy measurement for all examples in the CIFAR10 dataset to decide whether each one was close to the decision boundary. Surprisingly, we found that the nearest neighbor for 65% of the training examples had a different label than the example itself. In other words, in the raw feature space, this experiment provides some evidence that a majority of examples may be needed for learning the final decision boundary. This observation motivates RS2 as a strong data pruning baseline because it satisfies two desired properties: 1) it maximizes overall data coverage by periodically resampling the subset and 2) it provides representative examples from the dataset without overfitting. We remark that a majority of points are unlikely to be on the decision boundary if we first encode the input examples $x$ into a more semantically meaningful feature space. Learning such an encoding, however, requires first learning a decision boundary over the raw features and must be done during the model training itself. We leave a detailed study of this experiment, and the implications of this observation on selecting hard/easy examples for importance-sampling based data pruning to future work.

## B  ADDITIONAL DETAILS ON EXPERIMENTAL SETUP

We expand on the experimental setup described in Section 5.1 of the main body of the paper.

### B.1  DATA PRUNING BASELINES

We consider the following 24 data pruning baselines. We refer the reader to existing studies for more detailed descriptions of these methods (Guo et al., 2022).

1. Random: standard baseline; sample a static random subset of the dataset once before training
2. Contextual Diversity (CD) (Agarwal et al., 2020)
3. Herding (Welling, 2009; Chen et al., 2010)
4. K-Center Greedy (Sener & Savarese, 2018)
5. Least Confidence (Sachdeva et al., 2021)
6. Entropy (Sachdeva et al., 2021)
7. Margin (Sachdeva et al., 2021)
8. Forgetting (Toneva et al., 2018)
9. GraNd (Paul et al., 2021)
10. Contrastive Active Learning (CAL) (Liu et al., 2021)
11. Craig (Mirzasoleiman et al., 2020)

12. GradMatch (Killamsetty et al., 2021a)
13. Glister (Killamsetty et al., 2021b)
14. Facility Location (FL) (Iyer et al., 2021)
15. GraphCut (Iyer et al., 2021)
16. Active Learning with confidence-based example informativeness (AL (Conf)) (Park et al., 2022)
17. Active Learning with loss-based example informativeness (AL (LL)) (Park et al., 2022)
18. Active Learning with margin-based example informativeness (AL (Margin)) (Park et al., 2022)
19. Self-supervised prototypes with easy examples (SSP-Easy) (Sorscher et al., 2022)
20. Self-supervised prototypes with hard examples (SSP-Hard) (Sorscher et al., 2022)
21. Supervised prototypes with easy examples (SP-Easy) (Sorscher et al., 2022)
22. Supervised prototypes with hard examples (SP-Hard) (Sorscher et al., 2022)
23. Moderate-DS (Xia et al., 2023)
24. Area under the margin (Coverage-centric Coreset Selection) (AUM (CCS)) (Zheng et al., 2023)
25. Coresets for Data-efficient Deep Learning (CREST) (Yang et al., 2023b)

## B.2 DATASET DISTILLATION BASELINES

We compare against the following eight dataset distillation methods.

1. Dataset Distillation (DD) (Wang et al., 2018)
2. Flexible Dataset Distillation (LD) (Bohdal et al., 2020)
3. Dataset Condensation (DC) (Zhao et al., 2021)
4. Differentiable Siamese Augmentation (DSA) (Zhao & Bilen, 2021)
5. Distribution Matching (DM) (Zhao & Bilen, 2023)
6. Aligning Features (CAFE) (Wang et al., 2022)
7. Aligning Features + Differentiable Siamese Augmentation (CAFE+DSA) (Wang et al., 2022)
8. Trajectory Matching (TM) (Cazenavette et al., 2022)

## B.3 ADDITIONAL TRAINING DETAILS

For all experiments (except GPT2 due to cost considerations) we conduct three runs using different random seeds and report the average accuracy and runtime. We include additional details on the hyperparameters and hardware used below.

**Hyperparameters** We use the following hyperparameters for our experiments: For CIFAR10 and CIFAR100 experiments, we use SGD as the optimizer with batch size 128, initial learning rate 0.1, a cosine decay learning rate schedule (Loshchilov & Hutter, 2016), momentum 0.9, weight decay 0.0005, and 200 training epochs. For data augmentation, we apply random cropping and horizontal flipping with four-pixel padding on the $32 \times 32$ training images. For ImageNet30 and ImageNet, we use the same hyerparameters as above except for a larger batch size on ImageNet (256). We also use different data augmentation: training images are randomly resized and cropped to $224 \times 224$ with random horizontal flipping. Further details can be found in the source code.

**Hardware Setup** We run image classification experiments on a university cluster with job isolation and NVIDIA RTX 3090 GPUs. We run GPT2 experiments using AWS P3 GPU instances with eight NVIDIA V100 GPUs (as GPT2 experiments require more compute power). Utilizing the former allows us to reduce the cost of our experiments (e.g., compared to training entirely using AWS), but introduces the potential for increased variance compared to training with completely dedicated hardware—Even though all experiments run with exclusive access to one GPU and a set of CPU cores, cluster load can influence runtime measurements. We observe small variance across multiple runs of the same experiment on small datasets (e.g., on CIFAR10 the three run standard deviation is generally less than one percent of the total runtime), but larger variance on ImageNet, likely do to an increased load on the shared file system and longer experiment runtimes. As such, we calculate the runtime of

each method on ImageNet as follows: We calculate the minimum time per mini-batch using all runs across *all* methods, and then use this value to compute individual method runtimes by multiplying by the total number of batches during training and adding any necessary overheads for subset selection. More specifically, we have: the total runtime of any method $T_{\text{total}} = T_{\text{total\_subset\_selection}} + T_{\text{total\_training\_time}}$ with $T_{\text{total\_training\_time}} = T_{\text{global\_minimum\_batch\_runtime}} \times \text{total\_number\_of\_batches}$. Note that this means runtimes differ only due to subset selection overhead as expected (once a subset has been selected, all methods train on the same number of examples per round using the same hardware, and thus should have the same per round training time). Furthermore, we calculate $T_{\text{total\_subset\_selection}}$ as the minimum subset selection time observed across three runs of each method. The above runtime calculation allows us to minimize the affect of cluster noise on our experiments and ensure a fair comparison for the ImageNet time-to-accuracy reported in the paper.

## B.4    BASELINES WITH REPEATED SUBSET SELECTION

We expand on Section 5.2 and provide additional details for our implementations of baseline data pruning methods with repeated (per-round) subset selection. Accuracy results on CIFAR10 when training with these methods are shown in Table 1.

### B.4.1    DESCRIPTIONS OF -RS BASELINES

The -RS methods in Table 1 (left) start from an initial importance distribution over all training examples and then sample a subset for learning from that static distribution each epoch. The initial distribution is generated by pretraining an auxiliary model(s) on the full dataset for 10 epochs and then using this model to quantify example importance according to the corresponding baseline method of interest (in the same manner as the baseline would quantify example importance using a partially pretrained auxiliary model when generating a static subset at the beginning of training Guo et al. (2022)). We do not present -RS results for baselines methods which do not generate importance scores for all examples.

For each baseline, to create a categorical distribution over the dataset we apply the $softmax$ function to the baseline-specific example scores that are normally used for subset selection. Specifically, given a dataset $S = \{\mathbf{x_i}, y_i\}_{i=1}^{N}$:

**Least Confidence-RS**    We calculate $s_{least\,confidence}(\mathbf{x_i}) = 1 - \max_{j=1,\dots,C} P(\hat{y}_i = j|\mathbf{x_i})$ for each data point using the partially pretrained auxiliary model. The example importance scores for each instance are taken to be $-s_{least\,confidence}(\mathbf{x_i})$.

**Entropy-RS**    We calculate the example importance scores as in Least Confidence-RS, but replace $s_{least\,confidence}(\mathbf{x_i})$ with $s_{entropy}(\mathbf{x_i}) = -\sum_{j=1}^{C} P(\hat{y}_i = j|\mathbf{x_i}) \log P(\hat{y}_i = j|\mathbf{x_i})$.

**Margin-RS**    We calculate $s_{margin}(\mathbf{x_i}) = 1 - \min_{y_i \neq \hat{y}_i}(P(\hat{y}_i|\mathbf{x_i}) - P(y_i|\mathbf{x_i}))$ for each data point using the partially pretrained auxiliary model. The example importance scores are taken to be $-s_{margin}(\mathbf{x_i})$.

**Forgetting-RS**    We create the categorical distribution such that the probability of sampling a data point $\mathbf{x_i}$ is proportional to the number of times $\mathbf{x_i}$ is missclassified while partially pretraining the auxiliary model. A missclassification is defined as follows: $\mathbf{x_i}$ is classified incorrectly in the current epoch after having been correctly classified in the previous epoch.

**GraNd-RS**    We calculate the categorical distribution such that the probability of sampling a data point $\mathbf{x_i}$ is proportional to the average contribution from $\mathbf{x_i}$ to the decline in the training loss while pretraining the auxiliary model.

**CAL-RS**    We calculate the categorical distribution based on a heuristic for how far each data point is from the decision boundary. Examples that have a predictive likelihood that diverges most from their neighbors in the embedding space have a higher probability of being sampled (they are assumed to be closer to the decision boundary and thus more important) (see Liu et al. (2021)).

**SP-Easy-RS**    We create the distribution based on the distance between each data point $\mathbf{x_i}$ and its corresponding class mean in the embedding space (see Sorscher et al. (2022)). That is the probability of sampling a data point $x_i$ is higher if, in the embedding space, $x_i$ is closer to the mean of all embeddings with the same class label.

### B.4.2 Descriptions of -RC Baselines

In between each round, the -RC methods in Table 1 (right) use the current weights of the model being trained to reselect the subset according to the baseline of interest's standard subset selection algorithm. That is, each epoch the current model weights are used to either 1) recompute example importance—in this case the examples with the highest importance scores are then selected for the subset at the next round—or 2) directly re-select the subset for training (e.g., in the case of submodular methods). We do not use any pretraining or an auxiliary model for the -RC methods and do not present -RC results for baselines which generate subsets independently of model weights. We show the pseudocode for -RC methods in Algorithm 2. Sampling in Line 4 of Algorithm 2 is specific for each baseline and depends on the current state of the model.

---

**Algorithm 2** -RC Baselines

---

**Require:** Dataset $S = \{x_i, y_i\}_{i=1}^N$, selection ratio $r \in (0, 1]$, batch size $b$, initial model $w^0$, $X$ rounds, baseline method $M$
1: $T \leftarrow \lceil N/b \rceil$
2: $t \leftarrow 1$
3: **for** round $j = 1$ to $X$ **do**
4:     $S' \leftarrow M.sample\_subset(S, r; w^t)$
5:     **for** $k = 1$ to $r \cdot T$ **do**
6:         batch $m \leftarrow S'[(k-1) \cdot b : k \cdot b]$
7:         $w^t \leftarrow train(w^{t-1}, m)$
8:         $t \leftarrow t + 1$
    **return** $w^t$

---

### B.4.3 Comparison to CREST

In addition to our -RS and -RC baselines above, we evaluated the repeated subset selection method CREST under the same conditions as presented in Table 1. CREST achieves accuracy rates of $87.1\%$, $90.4\%$, and $92.4\%$ for selection ratios $r$ of $5\%$, $10\%$, and $30\%$ respectively. It is noteworthy that CREST distinguishes itself from -RC methods by undergoing multiple rounds of training on the same subset before resampling.

### B.4.4 Total Unique Examples Seen By Repeated Subset Selection

We compare the total number of unique examples from the full dataset selected throughout training (as part of some subset) when using RS2 and select per-round baselines described above. Result on CIFAR10 with a selection ratio of $r = 10\%$ are shown in Figure 4. By definition, RS2 without replacement is the fastest method to utilize all training examples, however other baselines which reselect the subset each round (e.g., Margin-RS, Margin-RC, and Craig-RC) also explore nearly all training examples in the full dataset (e.g., >99%). Modifying baseline methods which select static subsets to re-select the subset each round leads to more of exploration of full dataset and contributes to the improved accuracy of these baselines compared to their static counterparts (e.g., Table 1 vs. Table 3).

## C Additional Experimental Results

Here we include additional evaluation result comparing Repeated Sampling of Random Subsets (RS2) to existing data pruning and dataset distillation methods. These results extend those presented in Section 5 of the main paper. We briefly discuss each result (table) in turn and how it connects to the arguments made in Section 5.

### C.1 End-to-End Data Pruning Accuracy on CIFAR10 and ImageNet

In Tables 3-4 we show the end-model accuracy of RS2 and existing data pruning methods for varying selection ratios on CIFAR10 and ImageNet respectively. The numbers in these tables were used to create Figure 2 in the main body of the paper. Recall from the discussion of Figure 2 in Section 5.2

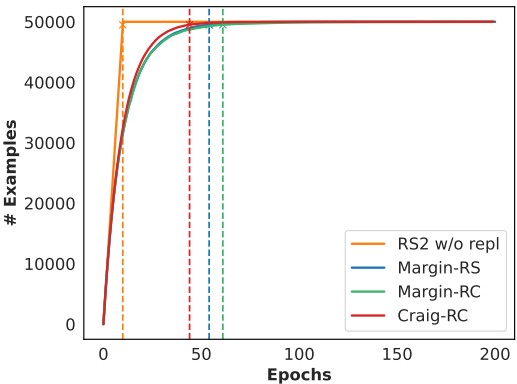

Figure 4: The total number of unique examples from the full dataset selected for training (as part of some subset) when using RS2 and select baseline methods with repeated subset selection. Dashed lines indicate the epoch at which each method saw 99% of the full dataset. RS2 without replacement is the fastes method to explore the full dataset, but baseline methods with repeated subset selection also explore >99% of the examples in the full dataset over the course of training.

that we use the combined baseline methods from recent studies (Park et al., 2022; Guo et al., 2022) together with newer prototype-based data pruning methods (Sorscher et al., 2022). Recall also that for these tables, we use the setting proposed by these works: for all baselines, we sample a static subset once before training starts. We use all baseline methods for CIFAR10, but some methods do not scale to the larger ImageNet dataset. We show in Table 8 that active learning already takes more than eight hours for subset selection in some settings on CIFAR10 and we are not aware of a scalable implementation of prototype-based methods that would allow for training on ImageNet.

As in Figure 2, Tables 3-4 show that the repeated sampling of RS2 leads to accuracy improvements compared to existing data pruning methods which sample a static subset (see discussion in Section 5.2). RS2 outperforms existing method across selection ratios. Interestingly, in the low compression regime ($r > 30\%$) our results support the intuition of many data pruning methods—that 'harder' samples benefit model training. For example, in Table 3, for $r = 50\%$, SSP-Hard (self-supervised prototypes subset selection with hard examples (Sorscher et al., 2022)) reaches 93.3% accuracy while SSP-Easy (self-supervised prototypes subset selection with easy examples) reaches 92.7%. RS2 however, matches or outperforms all existing baseline methods—those which select 'hard' or 'easy' samples. For example, in the same setting as above, RS2 reaches 95.2% accuracy. While RS2 also generally outperforms existing methods in the high compression regime ($r \leq 10\%$), for extreme compression ratios, like $r = 0.1\%$ on ImageNet, we find RS2 to be inferior to existing methods. We hypothesize that this occurs because in these extreme regimes, only a few examples are shown to the model for each class and these examples likely have large variance when using repeated random sampling coupled with data augmentation. In this setting, it may be best to select a static subset of only the easiest examples as highlighted in recent work (Sorscher et al., 2022), however the significance of this regime is debatable given the low end-model accuracy of all methods. Improving the performance in these regimes is of interest for future work.

Table 3: Accuracy achieved by different data pruning methods when training ResNet-18 on CIFAR10 for different subset selection sizes. Best method bolded; Next best underlined.

| Selection Ratio ($r$) | 1% | 5% | 10% | 20% | 30% | 40% | 50% | 100% |
|---|---|---|---|---|---|---|---|---|
| Random | 36.7±1.7 | 64.5±1.1 | 78.4±0.9 | 88.1±0.5 | 91.0±0.3 | 91.9±0.2 | 93.2±0.3 | 95.5±0.2 |
| CD | 23.6±1.9 | 38.1±2.2 | 58.8±2.0 | 81.3±2.5 | 90.8±0.5 | 93.3±0.4 | 94.3±0.2 | 95.5±0.2 |
| Herding | 34.8±3.3 | 51.0±3.1 | 63.5±3.4 | 74.1±2.5 | 80.1±2.2 | 85.2±0.9 | 88.0±1.1 | 95.5±0.2 |
| K-Center Greedy | 31.1±1.2 | 51.4±2.1 | 75.2±1.7 | 87.3±1.0 | 91.2±0.6 | 92.2±0.5 | 93.8±0.5 | 95.5±0.2 |
| Least Confidence | 19.8±2.2 | 36.2±1.9 | 57.6±3.1 | 81.9±2.2 | 90.3±0.4 | 93.1±0.5 | 94.5±0.1 | 95.5±0.2 |
| Entropy | 21.1±1.3 | 35.3±3.0 | 57.6±2.8 | 81.9±0.4 | 89.8±1.6 | 93.2±0.2 | 94.4±0.3 | 95.5±0.2 |
| Margin | 28.2±1.0 | 43.4±3.3 | 73.2±1.3 | 85.5±0.9 | 91.3±0.5 | 93.6±0.3 | 94.5±0.2 | 95.5±0.2 |
| Forgetting | 35.2±1.6 | 52.1±2.2 | 79.0±1.0 | 89.8±0.9 | 92.3±0.4 | 93.6±0.4 | 93.8±0.3 | 95.5±0.2 |
| GraNd | 26.7±1.3 | 39.8±2.3 | 75.4±1.2 | 88.6±0.6 | 92.4±0.4 | 93.3±0.5 | 94.2±0.4 | 95.5±0.2 |
| CAL | 37.8±2.0 | 60.0±1.4 | 71.8±1.0 | 80.9±1.1 | 86.0±1.9 | 87.5±0.8 | 89.4±0.6 | 95.5±0.2 |
| Craig | 31.7±1.1 | 45.2±2.9 | 60.2±4.4 | 79.6±3.1 | 88.4±0.5 | 90.8±1.4 | 93.3±0.6 | 95.5±0.2 |
| GradMatch | 30.8±1.0 | 47.2±0.7 | 61.5±2.4 | 79.9±2.6 | 87.4±2.0 | 90.4±1.5 | 92.9±0.6 | 95.5±0.2 |
| Glister | 32.9±2.4 | 50.7±1.5 | 75.7±1.0 | 86.3±0.9 | 90.1±0.7 | 91.5±0.5 | 93.3±0.6 | 95.5±0.2 |
| FL | 38.9±1.4 | 60.8±2.5 | 74.7±1.3 | 85.6±1.9 | 91.4±0.4 | 93.2±0.3 | 93.9±0.2 | 95.5±0.2 |
| GraphCut | 42.8±1.3 | 65.7±1.2 | 74.0±1.5 | 86.3±0.9 | 90.2±0.5 | 91.5±0.4 | 93.8±0.5 | 95.5±0.2 |
| AL (Conf) | 35.2±1.5 | 60.6±3.1 | 83.6±0.7 | 90.5±0.4 | 93.8±0.4 | 94.8±0.3 | 95.1±0.3 | 95.5±0.2 |
| AL (LL) | 37.5±4.3 | 63.1±2.0 | 85.0±0.9 | 91.2±0.7 | 93.8±0.6 | 94.4±0.5 | 95.0±0.4 | 95.5±0.2 |
| AL (Margin) | 36.7±0.8 | 62.2±1.1 | 84.5±0.7 | 91.0±0.5 | 93.9±0.4 | 94.5±0.3 | **95.3±0.2** | 95.5±0.2 |
| SSP-Easy | 35.6±1.7 | 62.1±1.2 | 72.0±0.8 | 85.9±0.4 | 90.0±0.2 | 91.5±0.4 | 92.7±0.0 | 95.5±0.2 |
| SSP-Hard | 34.2±1.1 | 58.0±2.4 | 74.3±1.7 | 86.1±1.3 | 90.3±0.4 | 91.9±0.3 | 93.3±0.2 | 95.5±0.2 |
| SP-Easy | 37.1±1.4 | 59.8±0.5 | 72.3±2.9 | 85.1±1.0 | 89.6±0.2 | 91.6±0.2 | 92.7±0.2 | 95.5±0.2 |
| SP-Hard | 35.0±0.7 | 60.9±1.8 | 74.1±1.1 | 86.3±0.3 | 89.8±0.6 | 91.5±0.3 | 93.0±0.3 | 95.5±0.2 |
| Moderate-DS | 36.4±3.0 | 64.9±3.2 | 79.0±0.8 | 87.0±0.3 | 90.0±0.0 | 91.6±0.0 | 92.6±0.1 | 95.5±0.2 |
| AUM (CCS) | 39.6±3.2 | 68.7±4.1 | 86.1±0.6 | 90.9±0.3 | 93.0±0.2 | 89.7±0.8 | 94.9±0.2 | 95.5±0.2 |
| RS2 w/ repl | 51.1±3.5 | 86.7±0.8 | 89.7±0.2 | 93.5±0.3 | 94.2±0.1 | 94.6±0.2 | 95.1±0.2 | 95.5±0.2 |
| RS2 w/ repl (stratified) | 51.1±4.5 | 86.6±0.5 | 89.8±0.4 | 93.4±0.1 | **94.5±0.1** | **94.8±0.1** | 95.1±0.3 | 95.5±0.2 |
| RS2 w/o repl | **51.8±2.0** | **87.1±0.8** | **91.7±0.5** | **94.0±0.5** | 94.3±0.2 | 94.7±0.1 | 95.2±0.1 | 95.5±0.2 |

Table 4: Accuracy achieved by different data pruning methods when training ResNet-18 on ImageNet for different subset selection sizes. Repeatedly Sampling Random Subsets (RS2) considerably outperforms existing methods for realistic selection ratios. Best method bolded; Next best underlined.

| Select Ratio ($r$) | 0.1% | 0.5% | 1% | 5% | 10% | 30% | 100% |
|---|---|---|---|---|---|---|---|
| Random | 0.76±0.01 | 3.78±0.14 | 8.85±0.46 | 40.09±0.21 | 52.1±0.22 | 64.11±0.05 | 69.52±0.45 |
| CD | 0.76±0.01 | 1.18±0.06 | 2.16±0.18 | 25.82±2.02 | 43.84±0.12 | 62.13±0.45 | 69.52±0.45 |
| Herding | 0.34±0.01 | 1.7±0.13 | 4.17±0.26 | 17.41±0.34 | 28.06±0.05 | 48.58±0.49 | 69.52±0.45 |
| K-Center Greedy | 0.76±0.01 | 1.57±0.09 | 2.96±0.24 | 27.36±0.08 | 44.84±1.03 | 62.12±0.46 | 69.52±0.45 |
| Least Confidence | 0.29±0.04 | 1.03±0.25 | 2.05±0.38 | 27.05±3.25 | 44.47±1.42 | 61.8±0.33 | 69.52±0.45 |
| Entropy | 0.31±0.02 | 1.01±0.17 | 2.26±0.3 | 28.21±2.83 | 44.68±1.54 | 61.82±0.31 | 69.52±0.45 |
| Margin | 0.47±0.02 | 1.99±0.29 | 4.73±0.64 | 35.99±1.67 | 50.29±0.92 | 63.62±0.15 | 69.52±0.45 |
| Forgetting | 0.76±0.01 | 4.69±0.17 | 14.02±0.13 | 47.64±0.03 | 55.12±0.13 | 62.49±0.11 | 69.52±0.45 |
| GraNd | 1.04±0.04 | 7.02±0.05 | 18.1±0.22 | 43.53±0.19 | 49.92±0.21 | 57.98±0.17 | 69.52±0.45 |
| CAL | **1.29±0.09** | 7.5±0.26 | 15.94±1.3 | 38.32±0.78 | 46.49±0.29 | 58.31±0.32 | 69.52±0.45 |
| Craig | 1.13±0.08 | 5.44±0.52 | 9.4±1.69 | 32.3±1.24 | 38.77±0.56 | 44.89±3.72 | 69.52±0.45 |
| GradMatch | 0.93±0.04 | 5.2±0.22 | 12.28±0.49 | 40.16±2.28 | 45.91±1.73 | 52.69±2.16 | 69.52±0.45 |
| Glister | 0.98±0.06 | 5.91±0.42 | 14.87±0.14 | 44.95±0.28 | 52.04±1.18 | 60.26±0.28 | 69.52±0.45 |
| FL | 1.23±0.03 | 5.78±0.08 | 12.72±0.21 | 40.85±1.25 | 51.05±0.59 | 63.14±0.03 | 69.52±0.45 |
| GraphCut | 1.21±0.09 | 7.66±0.43 | 16.43±0.53 | 42.23±0.6 | 50.53±0.42 | 63.22±0.26 | 69.52±0.45 |
| RS2 w/ repl | 0.17±0.03 | 16.35±0.56 | 44.45±0.07 | 45.4±7.18 | 64.87±0.10 | 68.23±0.07 | 69.52±0.45 |
| RS2 w/ repl (stratified) | 0.18±0.02 | **33.66±0.13** | **46.96±0.13** | 62.32±0.08 | 64.92±0.10 | **68.24±0.08** | 69.52±0.45 |
| RS2 w/o repl | 0.19±0.02 | 18.2±0.35 | 44.42±0.04 | **63.2±0.07** | **66.0±0.18** | 68.19±0.06 | 69.52±0.45 |

## C.2 END-TO-END DATA PRUNING ACCURACY ON CIFAR100 AND IMAGENET30

In Table 5, we include additional end-model accuracy results for RS2 and existing data pruning methods on two datasets, CIFAR100 and ImageNet30, not included in the main paper due to space considerations. For these experiments, we include a representative set of baseline methods which sample static subsets, together with our modified version of the recent prototype-based data pruning method which utilizes repeated subset selection between each round (SP-Easy-RS) (see Section 5.2). Thus, Table 5 extends the end-model accuracy results presented previously for CIFAR10 and ImageNet in Figure 2 and Tables 1, 3, and 4. Observe that RS2 also outperforms existing methods on these datasets. For example, in the high compression regime ($r = 10\%$), RS2 without replacement reaches 73% accuracy on CIFAR100, while the best baseline method, our per-round prototype-based data pruning method reaches only 66%. Existing methods which sample static subsets only once before training begins reach just 36% in this setting.

## C.3 END-TO-END DATA PRUNING ACCURACY ON MEDMNIST

To demonstrate the application of RS2 in real-world, class-imbalanced settings, we present end-model accuracy results for RS2 alongside selected baselines on four datasets from MedMNIST (Yang et al., 2023a) in Table 6. RS2 achieves the highest end-model accuracy on each dataset. Moreover, for selection ratio $r = 10\%$, RS2 achieves comparable accuracy to training on the entire dataset in every round. Specifically, it achieves $90.7\%$ compared to $92.2\%$ on PathMNIST, $72.7\%$ compared to $75.5\%$ on DermaMNIST, $95.3\%$ compared to $96.3\%$ on BloodMNIST, and $90.2\%$ compared to $92.0\%$ on OrganCMNIST.

## C.4 END-TO-END DATA PRUNING ACCURACY FOR VISUAL TRANSFORMERS (VIT)

In Table 7, we show end-model accuracy results for RS2 and selected baselines for a different neural network architecture, i.e. ViT (Dosovitskiy et al., 2021). Specifically, we train ViT with patch size four on CIFAR10. RS2 achieves higher end-model accuracy than selected baselines across varying selection ratios. For example, RS2 achieves 67% accuracy with $r = 10\%$ while the next closest baseline (GraphCut) achieves 59.6%. These results show that our initial findings extend to ViT models as well, i.e., RS2 is a strong baseline which outperforms existing data pruning methods.

Table 5: Accuracy achieved by select data pruning methods when training ResNet-18 on CIFAR100 and ImageNet30. Best method bolded; Next best underlined.

| Dataset | Select Ratio ($r$) | 10% | 20% | 30% | 40% | 50% | 60% | 70% | 80% | 90% | 100% |
|---|---|---|---|---|---|---|---|---|---|---|---|
| CIFAR100 | Random | 32.0±0.9 | 53.6±0.6 | 63.6±0.5 | 67.2±0.5 | 71.0±0.3 | 73.1±0.4 | 75.2±0.2 | 76.1±0.3 | 77.5±0.2 | 78.7±0.2 |
| | K-Center Greedy | 33.9±1.5 | 56.2±0.9 | 64.5±0.6 | 69.8±0.4 | 72.1±0.5 | 74.3±0.4 | 75.8±0.3 | 77.2±0.2 | 77.8±0.2 | 78.7±0.2 |
| | Margin | 18.7±2.1 | 38.2±1.6 | 58.1±0.8 | 65.1±0.6 | 70.1±0.5 | 73.3±0.3 | 75.4±0.3 | 76.9±0.4 | 78.5±0.2 | 78.7±0.2 |
| | Forgetting | 35.4±1.0 | 54.7±0.9 | 64.6±0.7 | 68.6±0.8 | 71.5±0.4 | 73.7±0.5 | 75.5±0.3 | 76.1±0.3 | 76.9±0.3 | 78.7±0.2 |
| | GraNd | 30.8±1.9 | 49.4±1.0 | 62.8±0.9 | 68.1±0.6 | 70.5±0.3 | 72.5±0.4 | 74.5±0.3 | 76.4±0.2 | 77.8±0.2 | 78.7±0.2 |
| | Glister | 36.4±1.0 | 55.5±1.0 | 63.9±0.8 | 69.1±0.7 | 71.2±0.6 | 73.5±0.4 | 75.0±0.3 | 76.9±0.2 | 77.6±0.2 | 78.7±0.2 |
| | GraphCut | 36.3±1.1 | 56.0±0.8 | 65.5±0.6 | 69.5±0.4 | 71.1±0.4 | 73.8±0.4 | 75.4±0.2 | 76.4±0.2 | 78.0±0.2 | 78.7±0.2 |
| | AL (Conf) | 36.1±1.6 | 55.7±1.0 | 65.8±0.7 | 70.6±0.5 | 73.7±0.4 | 76.1±0.5 | 77.1±0.3 | 78.0±0.2 | 78.4±0.2 | 78.7±0.2 |
| | AL (LL) | 33.1±1.9 | 55.3±1.3 | 64.9±0.8 | 70.3±0.7 | 73.1±0.5 | 75.9±0.5 | 77.0±0.3 | 78.2±0.3 | 78.5±0.2 | 78.7±0.2 |
| | AL (Margin) | 36.0±1.0 | 57.3±0.5 | 66.0±0.6 | 70.4±0.5 | 73.6±0.5 | 76.1±0.4 | 77.2±0.3 | 78.2±0.3 | 78.5±0.2 | 78.7±0.2 |
| | SSP-Easy | 32.8±2.0 | 50.0±1.5 | 62.5±1.5 | 67.9±0.3 | 70.2±0.2 | 73.4±0.3 | 75.0±0.7 | 76.3±0.6 | 77.4±0.1 | 78.7±0.2 |
| | SSP-Hard | 29.7±1.5 | 53.3±0.6 | 63.2±0.5 | 67.8±0.2 | 71.3±0.2 | 72.9±0.2 | 74.8±0.1 | 75.9±0.8 | 77.1±0.2 | 78.7±0.2 |
| | SP-Easy | 33.6±0.9 | 53.0±2.0 | 63.0±1.0 | 67.4±1.0 | 70.5±0.3 | 73.3±0.2 | 74.9±0.2 | 76.3±0.6 | 76.9±0.3 | 78.7±0.2 |
| | SP-Hard | 31.2±2.7 | 53.6±0.4 | 63.0±0.6 | 68.0±0.8 | 71.1±0.3 | 73.0±0.4 | 74.6±0.8 | 75.8±0.9 | 77.4±0.4 | 78.7±0.2 |
| | SP-Easy-RS | 66.1±1.8 | 72.7±0.6 | 74.6±0.5 | 75.5±0.2 | 76.3±0.3 | 76.9±0.4 | 77.6±0.1 | 78.0±0.1 | 78.3±0.3 | 78.7±0.2 |
| | RS2 w/ repl | 68.8±1.5 | 74.4±0.1 | **76.1±0.3** | 76.8±0.1 | **77.6±0.2** | 77.7±0.0 | **78.3±0.3** | **78.4±0.2** | **78.7±0.1** | 78.7±0.2 |
| | RS2 w/ repl (stratified) | 68.6±2.1 | 74.6±0.7 | 75.9±0.2 | 76.7±0.2 | 77.7±0.1 | 77.7±0.1 | 78.1±0.3 | 78.2±0.2 | 78.3±0.3 | 78.7±0.2 |
| | RS2 w/o repl | **73.0±0.3** | **74.9±0.7** | 76.1±0.5 | **77.1±0.1** | 77.5±0.4 | **78.0±0.1** | 78.3±0.2 | 78.3±0.2 | 78.4±0.3 | 78.7±0.2 |
| ImageNet30 | Random | 69.3±0.7 | 83.7±0.5 | 86.9±0.4 | 90.3±0.3 | 92.2±0.3 | 93.0±0.2 | 94.6±0.3 | 95.2±0.2 | 95.4±0.2 | 96.1±0.1 |
| | K-Center Greedy | 69.7±0.9 | 84.1±0.5 | 88.9±0.4 | 91.6±0.3 | 93.4±0.2 | 94.4±0.3 | 95.1±0.2 | 95.3±0.2 | 95.6±0.2 | 96.1±0.1 |
| | Margin | 56.9±1.1 | 77.3±0.7 | 83.7±0.5 | 90.5±0.4 | 92.9±0.2 | 94.4±0.3 | 95.1±0.2 | **95.8±0.2** | **96.0±0.1** | 96.1±0.1 |
| | Forgetting | 64.1±0.9 | 85.4±0.7 | 87.3±0.5 | 90.9±0.3 | 93.6±0.4 | 94.8±0.2 | 94.9±0.2 | 95.1±0.2 | 95.3±0.2 | 96.1±0.1 |
| | GraNd | 69.3±0.9 | 85.7±0.5 | 90.0±0.5 | 92.4±0.4 | 93.6±0.3 | 94.7±0.4 | 95.1±0.2 | 95.5±0.2 | 95.7±0.1 | 96.1±0.1 |
| | Glister | 72.4±0.7 | 82.9±0.5 | 87.0±0.4 | 91.2±0.3 | 92.7±0.3 | 93.3±0.3 | 94.2±0.2 | 95.0±0.2 | 95.8±0.2 | 96.1±0.1 |
| | GraphCut | 71.9±0.6 | 83.0±0.3 | 88.5±0.3 | 91.2±0.3 | 92.9±0.2 | 93.7±0.3 | 94.4±0.2 | 95.3±0.2 | 95.6±0.2 | 96.1±0.1 |
| | AL (Conf) | 70.7±1.1 | 87.0±0.5 | 90.3±0.5 | 93.1±0.4 | 94.3±0.3 | 95.1±0.2 | 95.5±0.4 | 95.7±0.2 | **96.0±0.1** | 96.1±0.1 |
| | AL (LL) | 68.4±1.5 | 85.5±0.7 | 89.3±0.6 | 93.1±0.5 | 94.7±0.2 | 95.3±0.2 | 95.6±0.3 | **95.8±0.2** | **96.0±0.2** | 96.1±0.1 |
| | AL (Margin) | 71.9±0.9 | 86.7±0.5 | 90.1±0.4 | 93.3±0.4 | 94.5±0.3 | 95.1±0.2 | 95.6±0.3 | **95.8±0.2** | **96.0±0.2** | 96.1±0.1 |
| | SSP-Easy | 71.3±0.5 | 81.5±2.0 | 87.4±0.7 | 90.2±0.3 | 92.0±0.5 | 93.1±0.4 | 94.2±0.2 | 94.9±0.1 | 95.3±0.2 | 96.1±0.1 |
| | SSP-Hard | 70.4±1.7 | 83.0±0.7 | 87.4±0.3 | 91.1±0.2 | 92.9±0.2 | 93.2±0.7 | 94.5±0.2 | 94.9±0.4 | 95.2±0.2 | 96.1±0.1 |
| | SP-Easy | 70.0±1.5 | 82.4±0.3 | 87.1±1.3 | 89.9±0.6 | 92.0±0.4 | 93.4±0.2 | 94.3±0.3 | 94.6±0.2 | 95.4±0.0 | 96.1±0.1 |
| | SP-Hard | 68.0±1.2 | 81.6±0.3 | 87.6±0.7 | 90.8±0.8 | 92.7±0.6 | 93.7±0.3 | 94.3±0.2 | 94.8±0.4 | 95.3±0.2 | 96.1±0.1 |
| | SP-Easy-RS | 89.1±0.9 | 92.3±0.1 | 93.2±0.5 | 93.8±0.4 | 94.5±0.3 | 95.0±0.3 | 94.9±0.2 | 95.4±0.3 | 95.7±0.2 | 96.1±0.1 |
| | RS2 w/ repl | 91.7±0.6 | 93.7±0.2 | 94.2±0.6 | **94.9±0.2** | **95.3±0.2** | 95.0±0.1 | 95.4±0.4 | 95.6±0.2 | 95.9±0.1 | 96.1±0.1 |
| | RS2 w/ repl (stratified) | 91.7±0.6 | 93.6±0.3 | **94.8±0.4** | **94.9±0.3** | 95.2±0.1 | **95.4±0.3** | 95.5±0.0 | 95.6±0.3 | 95.9±0.2 | 96.1±0.1 |
| | RS2 w/o repl | **92.0±0.4** | **94.0±0.4** | 94.6±0.2 | 94.5±0.3 | 95.2±0.3 | 95.3±0.2 | **95.7±0.1** | 95.8±0.3 | 95.8±0.1 | 96.1±0.1 |

Table 6: Accuracy achieved by different data pruning methods when training ResNet-18 with $r = 10\%$ on MedMNIST datasets. Best method bolded; Next best underlined.

| Dataset | PathMNIST | DermaMNIST | BloodMNIST | OrganCMNIST |
|---|---|---|---|---|
| Random | 85.8±0.8 | 68.5±0.6 | 86.3±1.6 | 84.9±0.7 |
| CD | 77.9±4.4 | 68.0±0.5 | 68.8±6.5 | 71.3±3.2 |
| Herding | 55.2±10 | 46.3±18 | 69.5±1.4 | 57.6±3.3 |
| K-Center Greedy | 84.4±0.9 | 66.9±0.1 | 85.4±2.5 | 85.8±0.6 |
| Least Confidence | 67.7±4.4 | 59.6±12 | 56.5±17 | 57.2±13 |
| Entropy | 62.3±3.1 | 59.8±12 | 55.3±13 | 56.3±9.5 |
| Margin | 71.2±6.0 | 65.5±1.5 | 66.3±9.6 | 61.9±2.8 |
| GraNd | 72.5±6.2 | 66.9±0.1 | 49.6±6.0 | 52.2±4.6 |
| CAL | 79.6±6.9 | 62.9±7.1 | 81.9±4.2 | 74.1±0.4 |
| Craig | 84.6±0.6 | 68.8±0.2 | 84.8±1.1 | 83.1±1.2 |
| Glister | 85.8±0.8 | 69.5±0.1 | 87.0±1.3 | 84.7±0.1 |
| FL | 84.7±0.5 | 69.3±0.6 | 86.8±1.4 | 84.7±0.5 |
| GraphCut | 84.9±0.2 | 68.9±1.0 | 87.1±0.7 | 84.9±0.2 |
| RS2 w/ repl | 87.9±2.5 | 70.5±2.2 | 89.9±5.1 | 87.5±3.0 |
| RS2 w/ repl (stratified) | 89.7±0.8 | **72.7±0.4** | 95.2±0.5 | 90.0±0.3 |
| RS2 w/o repl | **90.7±0.6** | 71.6±1.4 | **95.3±0.1** | **90.2±0.2** |

Table 7: Accuracy achieved by different data pruning methods when training ViT with patch size four on CIFAR10 for varying subset selection sizes. Best method bolded; Next best underlined.

| Selection Ratio ($r$) | 1% | 5% | 10% | 20% | 30% | 40% | 50% | 100% |
|---|---|---|---|---|---|---|---|---|
| Random | 37.1±1.2 | 50.6±0.5 | 59.2±0.9 | 66.1±0.5 | 70.6±0.6 | 72.5±1.1 | 74.4±1.1 | 78.0±0.9 |
| CD | 28.2±1.5 | 38.4±1.2 | 44.6±2.4 | 52.9±3.7 | 61.2±1.2 | 63.6±0.7 | 67.3±2.0 | 78.0±0.9 |
| Herding | 25.7±7.5 | 40.7±3.5 | 47.5±4.1 | 57.6±2.8 | 66.8±3.0 | 68.6±2.4 | 72.9±3.0 | 78.0±0.9 |
| K-Center Greedy | 35.5±0.8 | 48.0±1.5 | 57.6±0.4 | 64.6±1.2 | 68.9±0.3 | 72.3±0.6 | 74.5±0.3 | 78.0±0.9 |
| Least Confidence | 21.3±1.6 | 34.2±3.6 | 39.7±3.2 | 47.6±4.3 | 57.2±1.7 | 61.7±1.0 | 66.1±2.5 | 78.0±0.9 |
| Entropy | 23.7±4.2 | 36.9±2.7 | 44.3±2.9 | 54.3±1.4 | 61.4±1.3 | 66.2±0.1 | 69.2±0.4 | 78.0±0.9 |
| Margin | 27.7±2.3 | 37.7±1.4 | 45.0±0.4 | 50.7±3.3 | 58.6±3.0 | 62.6±3.4 | 67.4±2.2 | 78.0±0.9 |
| GraNd | 27.5±0.9 | 38.4±1.2 | 44.8±1.3 | 53.1±1.9 | 61.6±0.7 | 65.0±0.3 | 68.4±0.4 | 78.0±0.9 |
| CAL | 35.4±2.1 | 48.7±3.1 | 55.5±4.5 | 62.7±4.0 | 66.7±4.0 | 69.6±3.7 | 72.0±2.8 | 78.0±0.9 |
| Craig | 34.3±0.6 | 46.7±0.3 | 56.9±1.0 | 63.8±1.0 | 68.1±0.5 | 70.0±1.8 | 71.4±0.1 | 78.0±0.9 |
| Glister | 28.8±1.1 | 48.5±0.3 | 58.9±0.1 | 66.2±0.2 | 69.8±0.7 | 72.6±0.3 | 74.0±1.2 | 78.0±0.9 |
| FL | 37.5±1.2 | 49.7±0.7 | 59.6±0.6 | 65.7±0.9 | 69.9±0.6 | 72.8±0.7 | 74.0±0.6 | 78.0±0.9 |
| GraphCut | 40.2±1.6 | 53.0±1.3 | 59.6±0.3 | 65.9±0.8 | 70.9±0.5 | 72.7±0.6 | 73.6±0.6 | 78.0±0.9 |
| RS2 w/ repl | 51.6±0.1 | 62.7±0.9 | 64.9±1.0 | 71.3±0.2 | 73.5±0.9 | 75.3±2.1 | 75.5±0.4 | 78.0±0.9 |
| RS2 w/ repl (stratified) | **52.5±0.6** | **63.6±1.4** | 65.6±1.1 | 70.5±0.8 | 72.7±0.6 | 75.1±1.7 | **77.3±1.0** | 78.0±0.9 |
| RS2 w/o repl | 50.0±0.6 | 63.4±0.7 | **67.0±0.2** | **72.7±0.3** | **74.2±1.5** | **75.8±2.4** | 74.8±1.3 | 78.0±0.9 |

## C.5    END-TO-END DATA PRUNING TRAINING TIME

We now focus on additional results to accompany the runtime and time-to-accuracy results presented in the main body of the paper. Specifically, in Table 8, we show the total time needed for subset selection on CIFAR10 across all rounds for RS2 and compare to the total time needed for subset selection for existing data pruning methods which sample a static subset once before learning begins. In Table 9 we show the same measurement for our baseline methods which utilize repeated sampling between each round. Note that the differences presented in these tables are the dominant factor leading to differences in end-to-end runtime between methods: Once a subset has been selected for training at each round, all methods train on the same number of examples, and thus have the same per-round training time (assuming there is no noise). Thus the method with the lowest subset selection overhead will also be the fastest method for end-to-end training.

## C.6    TOTAL TIME FOR SUBSET SELECTION ON CIFAR10

Table 8 shows that sampling a static random subset once before training leads to the lowest total subset selection time, but that repeated random sampling (RS2) also has low subset selection overhead, i.e., generally less than one second on CIFAR10. The subset selection overhead of RS2 is orders-of-magnitude less than existing methods, even though they sample the subset only once at the beginning of training. For example, most existing methods require over 200 seconds for subset selection because they require pretraining an auxiliary model on the full dataset for a few epochs in order to rank example importance. Some methods, however, require even more time for subset selection; Active Learning based methods can require more than 32,000 seconds to select a subset with $r = 50\%$. Once example importance has been calculated, Table 9 shows that this information can be used to resample the subset for training between each round (our -RS baseline methods, see Section 5.2) with little additional overhead. All such methods, however, still require orders of magnitude more time for subset selection compared to RS2 due to the initial pretraining [1]. On the other hand, recomputing the most important examples between each round (our -RC methods), leads to increased subset selection overhead. The reason for this is that reranking example importance requires computing the model forward pass for all training examples between each round. Thus, such methods generally are unable to significantly reduce the end-to-end runtime compared to simply training on the full dataset each round; Even with a selection ratio of 5%, the fastest -RC method requires more than 3500 seconds for subset selection, yet end-to-end training, each round on the full dataset, requires only 4500 seconds.

---

We note that the pretraining overhead of GraNd in Table 8 uses the default hyperparameters from (Guo et al., 2022) in which the results from 10 pretrained auxiliary models are averaged, but for GraNd-RS in Table 9 we use only one model for consistency across all -RS methods.

Table 8: Comparison of the total time needed for subset selection for different data pruning methods when training on CIFAR10. Time reported in seconds. The overhead of repeated random sampling is considerably less than existing data pruning methods. For reference, training on the full dataset for 200 epochs takes roughly 4500 seconds. Best method bolded; Next best underlined.

| Select Ratio ($r$) | 1% | 5% | 10% | 20% | 30% | 40% | 50% |
|---|---|---|---|---|---|---|---|
| Random | **0.001±0.0** | **0.001±0.0** | **0.001±0.0** | **0.001±0.0** | **0.001±0.0** | **0.001±0.0** | **0.001±0.0** |
| CD | 237.78±3.08 | 243.73±6.06 | 247.01±13.72 | 244.39±3.58 | 243.42±2.18 | 254.46±9.87 | 254.72±2.28 |
| Herding | 238.29±3.84 | 241.31±2.37 | 253.0±5.84 | 258.49±12.35 | 255.08±0.8 | 268.91±8.13 | 263.16±2.84 |
| K-Center Greedy | 238.42±2.75 | 243.16±0.16 | 243.12±5.1 | 246.71±5.54 | 252.07±3.21 | 260.44±5.2 | 259.34±1.68 |
| Least Confidence | 238.61±2.6 | 238.08±2.59 | 241.96±5.89 | 239.92±3.72 | 237.07±1.64 | 239.47±6.53 | 240.01±5.08 |
| Entropy | 239.44±0.91 | 242.48±1.41 | 239.57±5.57 | 242.79±7.49 | 235.59±1.44 | 240.67±2.19 | 239.68±2.0 |
| Margin | 241.71±2.58 | 245.28±5.89 | 246.17±4.65 | 240.83±1.5 | 243.12±1.24 | 241.4±4.05 | 243.45±1.32 |
| Forgetting | 235.34±1.82 | 238.09±6.87 | 237.93±5.57 | 235.44±1.96 | 234.9±4.17 | 235.97±7.32 | 234.78±1.57 |
| GraNd | 2372.95±22.89 | 2406.41±79.95 | 2384.34±13.3 | 2377.09±16.19 | 2396.31±27.83 | 2375.14±17.84 | 2389.62±34.32 |
| CAL | 559.68±1.43 | 562.32±22.51 | 558.97±1.96 | 557.83±10.9 | 568.95±10.64 | 559.37±5.04 | 553.13±9.83 |
| Craig | 296.27±2.94 | 322.16±3.83 | 362.8±10.12 | 438.21±13.49 | 506.38±10.17 | 572.06±2.95 | 642.26±12.05 |
| Glister | 244.66±3.99 | 242.02±4.24 | 247.44±3.41 | 248.03±8.07 | 254.79±1.62 | 259.26±5.81 | 259.58±5.36 |
| FL | 330.79±20.94 | 587.43±16.07 | 764.18±86.51 | 1261.9±165.55 | 1863.98±241.04 | 2151.46±435.54 | 2722.44±145.36 |
| GraphCut | 325.66±9.37 | 551.81±46.75 | 728.66±45.93 | 1251.72±187.75 | 1601.92±202.89 | 2335.62±495.53 | 2672.69±643.3 |
| AL (Conf) | 408.3±8.4 | 908.1±9.8 | 2152.8±23.7 | 6694.8±80.6 | 13358.8±184.5 | 22120.2±329.5 | 32940.6±418.7 |
| AL (LL) | 398.5±5.4 | 879.1±9.8 | 2087.2±21.0 | 6592.5±43.9 | 13206.8±172.9 | 21933.8±362.6 | 32763.1±596.6 |
| AL (Margin) | 396.3±19.9 | 875.3±36.2 | 2107.2±73.6 | 6634.9±149.1 | 13298.5±241.1 | 22062.8±275.7 | 32871.6±425.9 |
| SSP-Easy | 265.67±5.87 | 269.89±8.48 | 268.44±8.51 | 263.84±6.49 | 264.85±5.58 | 268.21±5.82 | 269.39±4.78 |
| SSP-Hard | 285.91±9.25 | 288.47±8.07 | 290.3±26.43 | 284.52±27.16 | 293.41±21.92 | 287.28±25.74 | 271.59±6.09 |
| SP-Easy | 229.04±2.46 | 231.09±3.94 | 231.47±4.36 | 233.61±6.34 | 233.86±3.38 | 231.58±5.65 | 233.6±4.32 |
| SP-Hard | 227.68±1.65 | 234.39±1.2 | 230.85±3.65 | 227.67±2.23 | 231.66±2.64 | 230.97±3.1 | 233.12±5.94 |
| Moderate-DS | 5528.12±75.71 | 5528.39±75.78 | 5528.30±74.78 | 5528.18±75.10 | 5527.85±75.41 | 5528.19±75.67 | 5528.19±75.41 |
| AUM (CCS) | 10106.01±136.97 | 10106.01±136.97 | 10106.01±136.97 | 10106.01±136.97 | 10106.01±136.97 | 10106.01±136.97 | 10106.01±136.97 |
| RS2 w/ repl | 0.16±0.01 | 0.16±0.01 | 0.16±0.01 | 0.16±0.01 | 0.16±0.01 | 0.16±0.01 | 0.16±0.01 |
| RS2 w/ repl (stratified) | 0.68±0.02 | 0.72±0.03 | 0.75±0.01 | 0.84±0.03 | 0.93±0.05 | 1.0±0.03 | 1.09±0.03 |
| RS2 w/o repl | 0.09±0.01 | 0.1±0.01 | 0.11±0.01 | 0.12±0.01 | 0.14±0.01 | 0.15±0.01 | 0.19±0.01 |

Table 9: Comparison of the total time (in seconds) needed for subset selection for our dynamic data pruning methods when training on CIFAR10. The training subset is update for all methods after each round, either by resampling from a static example importance distribution (RS, left) or by recomputing example importance based on model updates (RC, right). For reference, training on the full dataset for 200 epochs takes roughly 4500 seconds. Best method bolded; Next best underlined.

| Selection Ratio ($r$) | 5% | 10% | 30% | Selection Ratio ($r$) | 5% | 10% | 30% |
|---|---|---|---|---|---|---|---|
| CD-RS | - | - | - | CD-RC | 3581.05±61.15 | 3860.4±31.36 | 4860.18±55.95 |
| Herding-RS | - | - | - | Herding-RC | 3851.82±37.15 | 4332.73±31.25 | 6578.17±9.45 |
| K-Center Greedy-RS | - | - | - | K-Center Greedy-RC | 3854.89±39.84 | 4384.02±38.72 | 6282.79±35.03 |
| Least Confidence-RS | 238.9±2.66 | 244.12±3.5 | 243.75±9.7 | Least Confidence-RC | 3698.25±47.31 | 3674.73±37.97 | 3630.66±30.48 |
| Entropy-RS | 241.26±2.63 | 240.41±1.33 | 247.27±3.19 | Entropy-RC | 3651.39±15.2 | 3677.18±32.94 | 3690.08±27.57 |
| Margin-RS | 243.46±3.29 | 239.53±4.26 | 239.27±3.36 | Margin-RC | 3715.31±75.59 | 3686.48±24.21 | 3760.33±91.99 |
| Forgetting-RS | 236.3±5.61 | 239.38±2.36 | 236.24±4.24 | Forgetting-RC | 3756.12±25.54 | 3732.81±33.9 | 3723.22±39.52 |
| GraNd-RS | 397.92±0.56 | 409.93±9.32 | 406.89±4.74 | GraNd-RC | 38035.57±1212.62 | 37390.35±939.82 | 29134.04±16123.62 |
| CAL-RS | 555.75±20.87 | 549.93±10.61 | 547.82±3.33 | CAL-RC | 69994.0±200.65 | 66947.73±2645.88 | 67086.71±1213.57 |
| Craig-RS | 1025.14±70.46 | 987.05±16.13 | 1021.92±13.77 | Craig-RC | 20517.31±955.04 | 27497.62±359.82 | 55305.63±988.66 |
| Glister-RS | - | - | - | Glister-RC | 4358.65±56.65 | 4966.57±20.48 | 6393.25±33.59 |
| SP-Easy-RS | 326.76 ± 63.77 | 371.23 ± 63.71 | 497.39 ± 8.69 | SP-Easy-RC | - | - | - |
| RS2 w/ repl (stratified) | 0.72 ± 0.03 | 0.75 ± 0.01 | 0.93 ± 0.05 | RS2 w/ repl (stratified) | 0.72 ± 0.03 | 0.75 ± 0.01 | 0.93 ± 0.05 |
| RS2 w/o repl | **0.1±0.01** | **0.11±0.01** | **0.14±0.01** | RS2 w/o repl | **0.1±0.01** | **0.11±0.01** | **0.14±0.01** |

Table 10: The total time required for RS2 and baseline data pruning methods to reach a target accuracy (time-to-accuracy) when training with varying selection ratios on CIFAR10. Time is reported in seconds. Part 1/3. The best method(s) is bolded.

| Target | Select Ratio ($r$) | 1% | 5% | 10% | 20% | 30% | 40% | 50% | 60% | 70% | 80% | 90% |
|---|---|---|---|---|---|---|---|---|---|---|---|---|
| | Random | 73 | **16** | 28 | **23** | 15 | **20** | 13 | 14 | 16 | 18 | 20 |
| | CD | - | 332 | 372 | 289 | 297 | 284 | 290 | 283 | 286 | 294 | 283 |
| | Herding | - | 295 | 296 | 275 | 283 | 279 | 273 | 278 | 281 | 285 | 291 |
| | K-Center Greedy | 353 | 284 | 304 | 273 | 267 | 280 | 283 | 281 | 283 | 290 | 296 |
| | Least Confidence | - | - | 393 | 330 | 299 | 278 | 275 | 265 | 259 | 262 | 264 |
| | Entropy | - | 484 | 398 | 346 | 289 | 270 | 274 | 267 | 256 | 259 | 262 |
| | Margin | - | 324 | 381 | 297 | 297 | 271 | 279 | 264 | 272 | 259 | 263 |
| | Forgetting | 302 | 263 | 299 | 264 | 258 | 264 | 258 | 249 | 257 | 252 | 253 |
| | GraNd | - | 2485 | 2461 | 2422 | 2428 | 2404 | 2425 | 2397 | 2376 | 2471 | 2387 |
| | CAL | - | 596 | 597 | 574 | 385 | 579 | 565 | 605 | 605 | 591 | 595 |
| | Craig | 365 | 342 | 395 | 461 | 521 | 583 | 668 | 740 | 816 | 909 | 952 |
| | Glister | - | 280 | 274 | 264 | 270 | 278 | 283 | 278 | 280 | 291 | 293 |
| 30% acc | FL | 377 | 605 | 803 | 1279 | 1878 | 2179 | 2734 | 3239 | 3671 | 4696 | 4513 |
| | GraphCut | 371 | 567 | 771 | 1268 | 1617 | 2355 | 2684 | 3462 | 3771 | 4427 | 4458 |
| | AL (Conf) | 395 | 395 | 395 | 395 | 395 | 395 | 395 | 395 | 395 | 395 | 395 |
| | AL (LL) | 386 | 386 | 386 | 386 | 386 | 386 | 386 | 386 | 386 | 386 | 386 |
| | AL (Margin) | 383 | 383 | 383 | 383 | 383 | 383 | 383 | 383 | 383 | 383 | 383 |
| | SSP-Easy | 320 | 295 | 300 | 280 | 288 | 287 | 281 | 282 | 282 | 287 | 289 |
| | SSP-Hard | 403 | 325 | 330 | 307 | 309 | 308 | 283 | 283 | 286 | 293 | 289 |
| | SP-Easy | 295 | 246 | 258 | 249 | 246 | 254 | 240 | 243 | 249 | 243 | 253 |
| | SP-Hard | 315 | 260 | 262 | 250 | 247 | 250 | 257 | 250 | 251 | 248 | 252 |
| | SP-Easy-RS | 284 | 249 | 256 | 262 | 249 | 250 | 257 | 249 | 250 | 249 | 252 |
| | RS2 w/ repl | 67 | **19** | 36 | 21 | **15** | **19** | 12 | 14 | 16 | **18** | **20** |
| | RS2 w/ repl (stratified) | **59** | 20 | 43 | 27 | **15** | **19** | 12 | 14 | 15 | **18** | **20** |
| | RS2 w/o repl | 83 | 17 | **14** | **11** | 15 | **19** | **11** | 14 | 16 | **18** | **20** |
| | Random | - | 78 | 109 | 65 | 55 | 69 | 49 | 56 | 48 | **54** | 61 |
| | CD | - | - | 617 | 354 | 366 | 384 | 361 | 325 | 318 | 349 | 304 |
| | Herding | - | - | 564 | 399 | 345 | 319 | 322 | 320 | 314 | 323 | 311 |
| | K-Center Greedy | - | 390 | 397 | 322 | 314 | 320 | 318 | 323 | 331 | 327 | 317 |
| | Least Confidence | - | - | - | 438 | 383 | 357 | 347 | 321 | 325 | 335 | 306 |
| | Entropy | - | - | - | 513 | 412 | 339 | 345 | 324 | 321 | 314 | 303 |
| | Margin | - | - | 755 | 439 | 407 | 359 | 351 | 305 | 305 | 314 | 304 |
| | Forgetting | - | - | 455 | 366 | 350 | 333 | 306 | 305 | 306 | 306 | 295 |
| | GraNd | - | - | 2811 | 2550 | 2531 | 2484 | 2498 | 2453 | 2408 | 2509 | 2428 |
| | CAL | - | 881 | 764 | 652 | 633 | 630 | 612 | 648 | 638 | 628 | 635 |
| | Craig | - | 458 | 523 | 537 | 568 | 633 | 704 | 782 | 885 | 946 | 993 |
| | Glister | - | 393 | 372 | 313 | 308 | 319 | 307 | 334 | 312 | 328 | 313 |
| 50% acc | FL | - | 695 | 909 | 1341 | 1917 | 2219 | 2771 | 3267 | 3704 | 4733 | 4535 |
| | GraphCut | - | 623 | 876 | 1321 | 1664 | 2395 | 2720 | 3506 | 3805 | 4463 | 4478 |
| | AL (Conf) | - | 895 | 895 | 895 | 895 | 895 | 895 | 895 | 895 | 895 | 895 |
| | AL (LL) | - | 867 | 867 | 867 | 867 | 867 | 867 | 867 | 867 | 867 | 867 |
| | AL (Margin) | - | 862 | 862 | 862 | 862 | 862 | 862 | 862 | 862 | 862 | 862 |
| | SSP-Easy | - | 375 | 394 | 336 | 333 | 336 | 316 | 323 | 314 | 324 | 329 |
| | SSP-Hard | - | 432 | 433 | 365 | 357 | 358 | 332 | 326 | 335 | 330 | 329 |
| | SP-Easy | - | 340 | 362 | 306 | 301 | 284 | 287 | 271 | 282 | 261 | 293 |
| | SP-Hard | - | 277 | 354 | 312 | 301 | 290 | 306 | 294 | 284 | 285 | 291 |
| | SP-Easy-RS | - | 308 | 355 | 335 | 289 | 298 | 304 | 290 | 282 | 267 | 273 |
| | RS2 w/ repl | 279 | 77 | 137 | 66 | 61 | **48** | 47 | 56 | 48 | **55** | 61 |
| | RS2 w/ repl (stratified) | **250** | 87 | 123 | 83 | 69 | 60 | 72 | 57 | 64 | **54** | **40** |
| | RS2 w/o repl | - | **64** | **44** | **46** | **46** | 58 | **35** | **41** | **32** | 53 | 60 |

## C.7 Time-to-Accuracy on CIFAR10 and ImageNet

Finally, as our primary focus is on reducing time-to-accuracy, we include in Tables 10-14 the time for select baseline methods and RS2 to reach a set of accuracy targets when training with varying selection ratios on CIFAR10 and ImageNet. For the active learning time-to-accuracy results in these tables, we report the runtime of the smallest selection ratio that reached the given accuracy. This prevents active learning time-to-accuracies from being dominated by large subset selection overheads as the selection ratio increases (e.g., Table 8), when these selection ratios are not strictly needed to reach the desired accuracy. As shown in the main body of the paper, RS2 provides the fastest time-to-accuracy compared to existing methods. Dashes indicate that the given method and selection ratio failed to reach the target accuracy. We leave a detailed study of these results for future work. In particular, an interesting question is how to decide what selection ratio $r$ one should use in order to minimize runtime to reach a desired accuracy.

Table 11: The total time required for RS2 and baseline data pruning methods to reach a target accuracy (time-to-accuracy) when training with varying selection ratios on CIFAR10. Time is reported in seconds. Part 2/3. The best method(s) is bolded.

| Target | Select Ratio ($r$) | 1% | 5% | 10% | 20% | 30% | 40% | 50% | 60% | 70% | 80% | 90% |
|---|---|---|---|---|---|---|---|---|---|---|---|---|
| | Random | - | - | 364 | 216 | 184 | 139 | 136 | 139 | 130 | 126 | 143 |
| | CD | - | - | - | 584 | 504 | 473 | 456 | 396 | 382 | 421 | 406 |
| | Herding | - | - | - | 926 | 760 | 456 | 406 | 418 | 395 | 415 | 393 |
| | K-Center Greedy | - | - | 552 | 420 | 384 | 397 | 414 | 380 | 412 | 493 | 379 |
| | Least Confidence | - | - | - | 911 | 514 | 455 | 453 | 420 | 374 | 372 | 369 |
| | Entropy | - | - | - | 954 | 565 | 457 | 464 | 423 | 371 | 387 | 385 |
| | Margin | - | - | - | 705 | 564 | 479 | 473 | 387 | 402 | 387 | 366 |
| | Forgetting | - | - | - | 549 | 473 | 430 | 366 | 376 | 374 | 360 | 356 |
| | GraNd | - | - | - | 2727 | 2689 | 2573 | 2583 | 2523 | 2489 | 2584 | 2489 |
| | CAL | - | - | - | 1025 | 801 | 763 | 720 | 748 | 719 | 720 | 695 |
| | Craig | - | - | 775 | 664 | 693 | 724 | 803 | 854 | 966 | 1020 | 1075 |
| | Glister | - | - | 601 | 429 | 409 | 418 | 390 | 406 | 393 | 383 | 396 |
| 70% acc | FL | - | - | 1144 | 1467 | 2009 | 2299 | 2855 | 3326 | 3785 | 4807 | 4597 |
| | GraphCut | - | - | 1136 | 1465 | 1734 | 2484 | 2804 | 3578 | 3889 | 4536 | 4538 |
| | AL (Conf) | - | - | 2884 | 2884 | 2884 | 2884 | 2884 | 2884 | 2884 | 2884 | 2884 |
| | AL (LL) | - | - | 2804 | 2804 | 2804 | 2804 | 2804 | 2804 | 2804 | 2804 | 2804 |
| | AL (Margin) | - | - | 2833 | 2833 | 2833 | 2833 | 2833 | 2833 | 2833 | 2833 | 2833 |
| | SSP-Easy | - | - | 716 | 445 | 440 | 404 | 412 | 407 | 410 | 399 | 390 |
| | SSP-Hard | - | - | 689 | 532 | 461 | 438 | 392 | 411 | 416 | 404 | 411 |
| | SP-Easy | - | - | 705 | 431 | 409 | 403 | 370 | 355 | 363 | 332 | 373 |
| | SP-Hard | - | - | 586 | 428 | 379 | 398 | 402 | 380 | 365 | 357 | 372 |
| | SP-Easy-RS | - | 440 | 508 | 415 | 383 | 374 | 377 | 358 | 378 | 339 | 354 |
| | RS2 w/ repl | - | **218** | 271 | 205 | 162 | 147 | 130 | 154 | **112** | 128 | 142 |
| | RS2 w/ repl (stratified) | - | **213** | 260 | 179 | 155 | 151 | 144 | **126** | 145 | 126 | **121** |
| | RS2 w/o repl | - | **201** | **168** | **128** | **116** | **127** | **105** | **125** | **112** | **107** | **122** |
| | Random | - | - | - | 553 | 470 | 395 | 303 | **266** | 277 | **199** | 306 |
| | CD | - | - | - | 917 | 713 | 654 | 601 | 494 | 495 | 567 | 508 |
| | Herding | - | - | - | - | - | 1522 | 1222 | 909 | 686 | 598 | 536 |
| | K-Center Greedy | - | - | - | 692 | 695 | 635 | 606 | 535 | 557 | 511 | 563 |
| | Least Confidence | - | - | - | - | 738 | 622 | 621 | 587 | 537 | 518 | 534 |
| | Entropy | - | - | - | - | 912 | 675 | 584 | 521 | 516 | 498 | 529 |
| | Margin | - | - | - | - | 975 | 678 | 606 | 538 | 548 | 496 | 509 |
| | Forgetting | - | - | - | 719 | 641 | 616 | 511 | 518 | 541 | 451 | 538 |
| | GraNd | - | - | - | 2984 | 2831 | 2671 | 2681 | 2636 | 2585 | 2716 | 2650 |
| | CAL | - | - | - | - | 1378 | 1132 | 995 | 978 | 968 | 921 | 816 |
| | Craig | - | - | - | 1057 | 1063 | 1075 | 986 | 1111 | 1116 | 1242 | 1218 |
| | Glister | - | - | - | 788 | 650 | 687 | 614 | 604 | 586 | 494 | 559 |
| 80% acc | FL | - | - | - | 1765 | 2331 | 2525 | 3122 | 3528 | 3997 | 4936 | 4742 |
| | GraphCut | - | - | - | 1779 | 2068 | 2681 | 3049 | 3782 | 4106 | 4735 | 4697 |
| | AL (Conf) | - | - | 2884 | 2884 | 2884 | 2884 | 2884 | 2884 | 2884 | 2884 | 2884 |
| | AL (LL) | - | - | 2804 | 2804 | 2804 | 2804 | 2804 | 2804 | 2804 | 2804 | 2804 |
| | AL (Margin) | - | - | 2833 | 2833 | 2833 | 2833 | 2833 | 2833 | 2833 | 2833 | 2833 |
| | SSP-Easy | - | - | - | 814 | 712 | 639 | 638 | 629 | 553 | 583 | 592 |
| | SSP-Hard | - | - | - | 840 | 786 | 739 | 584 | 625 | 626 | 570 | 573 |
| | SP-Easy | - | - | - | 839 | 702 | 699 | 617 | 466 | 461 | 493 | 595 |
| | SP-Hard | - | - | - | 766 | 673 | 605 | 558 | 578 | 496 | 503 | 514 |
| | SP-Easy-RS | - | 564 | 634 | 637 | 585 | 501 | 590 | 567 | 490 | 538 | 516 |
| | RS2 w/ repl | - | **331** | 383 | 334 | 308 | 343 | 274 | 293 | 337 | 312 | 324 |
| | RS2 w/ repl (stratified) | - | **331** | 339 | 320 | 317 | 341 | 289 | 295 | 275 | 272 | 303 |
| | RS2 w/o repl | - | **333** | **278** | **257** | **278** | **264** | **212** | **264** | **241** | 270 | **264** |

Table 12: The total time required for RS2 and baseline data pruning methods to reach a target accuracy (time-to-accuracy) when training with varying selection ratios on CIFAR10. Time is reported in seconds. Part 3/3. The best method(s) is bolded.

| Target | Select Ratio ($r$) | 1% | 5% | 10% | 20% | 30% | 40% | 50% | 60% | 70% | 80% | 90% |
|---|---|---|---|---|---|---|---|---|---|---|---|---|
| | Random | - | - | - | - | 1473 | 1462 | 1722 | 1906 | 2037 | 2149 | 2403 |
| | CD | - | - | - | - | 1327 | 1615 | 1806 | 1917 | 2002 | 2181 | 2461 |
| | Herding | - | - | - | - | - | - | 2736 | 2801 | 2810 | 2885 | 2839 |
| | K-Center Greedy | - | - | - | - | 1412 | 1662 | 1913 | 2093 | 2310 | 2433 | 2433 |
| | Least Confidence | - | - | - | - | 1446 | 1546 | 1746 | 1940 | 2080 | 2268 | 2423 |
| | Entropy | - | - | - | - | - | 1542 | 1783 | 1892 | 2024 | 2255 | 2468 |
| | Margin | - | - | - | - | - | 1554 | 1749 | 1877 | 2020 | 2151 | 2498 |
| | Forgetting | - | - | - | - | 1359 | 1569 | 1699 | 1969 | 2228 | 2334 | 2562 |
| | GraNd | - | - | - | - | 3530 | 3645 | 3845 | 3889 | 4042 | 4465 | 4531 |
| | CAL | - | - | - | - | - | 2201 | 2292 | 2459 | 2658 | 2692 | 3042 |
| | Craig | - | - | - | - | 1798 | 2024 | 2381 | 2613 | 2982 | 3134 | 3030 |
| | Glister | - | - | - | - | 1718 | 1734 | 1885 | 2222 | 2368 | 2485 | 2639 |
| 90% acc | FL | - | - | - | - | 3203 | 3591 | 4454 | 5229 | 5715 | 6903 | 6707 |
| | GraphCut | - | - | - | - | 2846 | 3782 | 4428 | 5387 | 5860 | 6459 | 6782 |
| | AL (Conf) | - | - | - | 7845 | 7845 | 7845 | 7845 | 7845 | 7845 | 7845 | 7845 |
| | AL (LL) | - | - | - | 7740 | 7740 | 7740 | 7740 | 7740 | 7740 | 7740 | 7740 |
| | AL (Margin) | - | - | - | 7787 | 7787 | 7787 | 7787 | 7787 | 7787 | 7787 | 7787 |
| | SSP-Easy | - | - | - | - | 1534 | 1719 | 1932 | 2152 | 2329 | 2582 | 2546 |
| | SSP-Hard | - | - | - | - | 1740 | 1760 | 1961 | 2155 | 2273 | 2452 | 2499 |
| | SP-Easy | - | - | - | - | - | 1699 | 1936 | 2126 | 2324 | 2323 | 2647 |
| | SP-Hard | - | - | - | - | - | 1712 | 1906 | 2157 | 2206 | 2407 | 2650 |
| | SP-Easy-RS | - | - | - | 1080 | 1284 | 1505 | 1721 | 1962 | 2173 | 2365 | 2619 |
| | RS2 w/ repl | - | - | - | 777 | 1028 | **1220** | 1435 | 1645 | 1893 | **1979** | 2378 |
| | RS2 w/repl (stratified) | - | - | - | 785 | 995 | 1267 | 1483 | **1577** | **1786** | 2099 | **2127** |
| | RS2 w/o repl | - | - | 566 | 723 | 953 | 1211 | **1291** | 1637 | 1866 | 2106 | 2357 |
| | Random | - | - | - | - | - | - | - | - | - | - | 3582 |
| | CD | - | - | - | - | - | - | - | - | 3124 | 3416 | 3784 |
| | Herding | - | - | - | - | - | - | - | - | - | - | 3815 |
| | K-Center Greedy | - | - | - | - | - | - | - | - | - | - | 3952 |
| | Least Confidence | - | - | - | - | - | - | - | - | 3084 | 3432 | 3838 |
| | Entropy | - | - | - | - | - | - | - | 2900 | 3171 | 3464 | 3822 |
| | Margin | - | - | - | - | - | - | - | - | 3057 | 3522 | 3797 |
| | Forgetting | - | - | - | - | - | - | - | - | - | - | 3835 |
| | GraNd | - | - | - | - | - | - | - | - | 5133 | 5721 | 5956 |
| | CAL | - | - | - | - | - | - | - | - | - | - | 4153 |
| | Craig | - | - | - | - | - | - | - | - | - | - | - |
| | Glister | - | - | - | - | - | - | - | - | - | - | 4012 |
| 95% acc | FL | - | - | - | - | - | - | - | - | - | - | 8221 |
| | GraphCut | - | - | - | - | - | - | - | - | - | - | 8028 |
| | AL (Conf) | - | - | - | - | - | - | - | 48708 | 48708 | 48708 | 48708 |
| | AL (LL) | - | - | - | - | - | - | - | 48549 | 48549 | 48549 | 48549 |
| | AL (Margin) | - | - | - | - | - | - | - | 48326 | 48326 | 48326 | 48326 |
| | SSP-Easy | - | - | - | - | - | - | - | - | - | - | - |
| | SSP-Hard | - | - | - | - | - | - | - | - | - | - | 3908 |
| | SP-Easy | - | - | - | - | - | - | - | - | - | - | 4214 |
| | SP-Hard | - | - | - | - | - | - | - | - | - | - | 3922 |
| | SP-Easy-RS | - | - | - | - | - | - | - | - | 3180 | 3505 | 3810 |
| | RS2 w/ repl | - | - | - | - | - | - | 2296 | **2498** | **2856** | 3245 | 3696 |
| | RS2 w/repl (stratified) | - | - | - | - | - | - | - | 2585 | 3003 | 3300 | **3501** |
| | RS2 w/o repl | - | - | - | - | - | - | **2153** | 2633 | 2943 | **3147** | 3569 |

Table 13: The total time required for RS2 and select baseline data pruning methods to reach a target accuracy (time-to-accuracy) when training with varying selection ratios on ImageNet. Time is reported in seconds. Part 1/2. The best method(s) is bolded.

| Target | Select Ratio ($r$) | 1% | 5% | 10% |
|---|---|---|---|---|
| 5% acc | Random | 919 | **289** | **231** |
| | Herding | 22222 | 27794 | 24243 |
| | Least Confidence | - | 15292 | 19362 |
| | Entropy | - | 19535 | 19285 |
| | Margin | 19494 | 14121 | 15785 |
| | Forgetting | 19081 | 18394 | 15978 |
| | GraNd | 146059 | 143640 | 147931 |
| | FL | 67907 | 67789 | 220647 |
| | GraphCut | 66336 | 224755 | 303318 |
| | RS2 w/ repl | 542 | 347 | **231** |
| | RS2 w/ repl (stratified) | **403** | **291** | 350 |
| | RS2 w/o repl | 530 | 347 | 347 |
| 10% acc | Random | 2003 | **462** | 579 |
| | Herding | - | 29468 | 24822 |
| | Least Confidence | - | 16043 | 20056 |
| | Entropy | - | 20517 | 19864 |
| | Margin | - | 14698 | 16364 |
| | Forgetting | 19799 | 18625 | 16094 |
| | GraNd | 14743 | 143755 | 148047 |
| | FL | 69144 | 220878 | 308052 |
| | GraphCut | 66831 | 224928 | 303434 |
| | RS2 w/ repl | 931 | 635 | **463** |
| | RS2 w/ repl (stratified) | **686** | 638 | 581 |
| | RS2 w/o repl | 907 | 635 | **463** |
| 20% acc | Random | - | 1906 | **1157** |
| | Herding | - | - | 36974 |
| | Least Confidence | - | 22684 | 23065 |
| | Entropy | - | 27158 | 22757 |
| | Margin | - | 19029 | 17984 |
| | Forgetting | - | 20877 | 16557 |
| | GraNd | - | 146123 | 148741 |
| | FL | - | 221802 | 308630 |
| | GraphCut | - | 225794 | 304012 |
| | RS2 w/ repl | 1355 | 1502 | 1273 |
| | RS2 w/ repl (stratified) | **1145** | **1388** | **1160** |
| | RS2 w/o repl | 1308 | **1386** | **1157** |
| 30% acc | Random | - | 7219 | 4282 |
| | Herding | - | - | - |
| | Least Confidence | - | - | 32555 |
| | Entropy | - | - | 32131 |
| | Margin | - | 43750 | 25159 |
| | Forgetting | - | 25959 | 18293 |
| | GraNd | - | 151840 | 158347 |
| | FL | - | 227519 | 313954 |
| | GraphCut | - | 307600 | 307600 |
| | RS2 w/ repl | 1626 | 3927 | 4514 |
| | RS2 w/ repl (stratified) | **1546** | **3583** | 3937 |
| | RS2 w/o repl | 1673 | 4100 | **3166** |

Table 14: The total time required for RS2 and select baseline data pruning methods to reach a target accuracy (time-to-accuracy) when training with varying selection ratios on ImageNet. Time is reported in seconds. Part 2/2. The best method(s) is bolded.

| Target | Select Ratio ($r$) | 1% | 5% | 10% |
|---|---|---|---|---|
| 40% acc | Random | - | - | 14814 |
| | Herding | - | - | - |
| | Least Confidence | - | - | - |
| | Entropy | - | - | - |
| | Margin | - | - | 32682 |
| | Forgetting | - | - | 28824 |
| | GraNd | - | - | 166680 |
| | FL | - | - | 322518 |
| | GraphCut | - | 234399 | 317553 |
| | RS2 w/ repl | 2015 | 6815 | **11689** |
| | RS2 w/ repl (stratified) | **1864** | **6586** | 11923 |
| | RS2 w/o repl | 2015 | 6699 | 12152 |
| 50% acc | Random | - | - | - |
| | Herding | - | - | - |
| | Least Confidence | - | - | - |
| | Entropy | - | - | - |
| | Margin | - | - | - |
| | Forgetting | - | - | 35305 |
| | GraNd | - | - | - |
| | FL | - | - | - |
| | GraphCut | - | - | - |
| | RS2 w/ repl | - | 8605 | 16550 |
| | RS2 w/ repl (stratified) | - | **8492** | 16552 |
| | RS2 w/o repl | - | 8605 | **16434** |
| 60% acc | Random | - | - | - |
| | Herding | - | - | - |
| | Least Confidence | - | - | - |
| | Entropy | - | - | - |
| | Margin | - | - | - |
| | Forgetting | - | - | - |
| | GraNd | - | - | - |
| | FL | - | - | - |
| | GraphCut | - | - | - |
| | RS2 w/ repl | - | 10337 | 20021 |
| | RS2 w/ repl (stratified) | - | **10282** | 20024 |
| | RS2 w/o repl | - | **10280** | **19790** |
| 65% acc | Random | - | - | - |
| | Herding | - | - | - |
| | Least Confidence | - | - | - |
| | Entropy | - | - | - |
| | Margin | - | - | - |
| | Forgetting | - | - | - |
| | GraNd | - | - | - |
| | FL | - | - | - |
| | GraphCut | - | - | - |
| | RS2 w/ repl | - | - | 22105 |
| | RS2 w/ repl (stratified) | - | - | 22107 |
| | RS2 w/o repl | - | - | **21873** |

Table 15: Accuracy achieved by different data pruning methods when training ResNet-18 on the CIFAR10 dataset with $p$ percent of the train set labels randomly flipped (noise ratio). Data pruning methods use a selection ratio of 10%. We test on the normal test set. We report raw end-model accuracy/accuracy drop compared to $p = 0$/relative accuracy drop compared to $p = 0$ (as a percentage of the $p = 0$ accuracy). Best method (highest accuracy) bolded; Next best underlined. Most robust method (lowest relative accuracy drop) starred.

| Noise Ratio | 10% | 30% | 50% |
|---|---|---|---|
| Random | 50.7±1.0/27.7/35.4 | 40.9±2.3/37.5/47.8 | 37.2±1.1/41.2/52.6 |
| CD | 31.2±3.9/27.6/47.0 | 30.3±0.6/28.5/48.4 | 29.7±2.9/29.1/49.5 |
| Herding | 15.9±2.2/47.6/74.9 | 17.1±1.1/46.4/73.1 | 16.1±1.8/47.4/74.7 |
| K-Center Greedy | 41.6±1.7/33.6/44.6 | 33.1±0.7/42.1/56.0 | 31.1±0.4/44.1/58.6 |
| Least Confidence | 27.4±0.7/30.2/52.4 | 25.2±2.1/32.4/56.2 | 24.5±2.6/33.1/57.5 |
| Entropy | 26.9±4.2/30.7/53.2 | 24.9±3.7/32.7/56.8 | 23.3±2.0/34.3/59.5 |
| Margin | 29.2±2.2/44.0/60.1 | 27.4±2.2/45.8/62.6 | 28.5±3.3/44.7/61.0 |
| Forgetting | 47.5±1.6/31.5/39.9 | 48.3±1.7/30.7/38.9 | 49.0±1.6/30.0/37.9 |
| GraNd | 34.5±3.2/40.9/54.3 | 49.4±5.6/26.0/34.5 | 60.0±1.3/15.4/20.4* |
| CAL | 45.4±2.5/26.4/36.8 | 39.9±1.2/31.9/44.4 | 35.5±0.4/36.3/50.5 |
| Craig | 50.3±0.7/9.9/16.5 | 39.5±0.8/20.7/34.4 | 37.3±1.5/22.9/38.0 |
| Glister | 49.8±2.5/25.9/34.2 | 40.1±1.5/35.6/47.1 | 37.9±1.5/37.8/50.0 |
| GraphCut | 50.6±1.4/23.4/31.7 | 42.0±0.7/32.0/43.2 | 37.7±1.1/36.3/49.0 |
| FL | 50.4±2.1/24.3/32.5 | 41.6±0.8/33.1/44.3 | 36.8±0.7/37.9/50.7 |
| AL (Conf) | 53.5±3.1/30.1/36.0 | 47.1±0.9/36.5/43.7 | 37.2±1.4/46.4/55.5 |
| AL (LL) | 57.1±0.4/27.9/32.8 | 45.1±1.8/39.9/46.9 | 38.2±0.6/46.8/55.1 |
| AL (Margin) | 57.6±0.5/26.9/31.8 | 46.1±1.2/38.4/45.4 | 36.9±1.0/47.6/56.3 |
| SSP-Easy | 51.1±1.5/20.9/29.0 | 40.7±1.9/31.3/43.4 | 36.7±1.0/35.3/49.1 |
| SSP-Hard | 50.4±0.3/23.9/32.2 | 40.9±1.5/33.4/44.9 | 36.3±1.9/38.0/51.2 |
| SP-Easy | 48.4±2.4/23.9/33.0 | 40.0±0.4/32.3/44.6 | 37.7±1.0/34.6/47.8 |
| SP-Hard | 47.5±2.4/26.6/35.9 | 39.7±1.3/34.4/46.4 | 34.3±2.0/39.8/53.7 |
| SP-Easy-RS | 74.2±0.6/14.2/16.0 | 63.4±1.0/25.0/28.3 | 57.8±0.7/30.6/34.7 |
| RS2 w/ repl | 77.5±1.0/12.2/13.6* | 69.9±0.4/19.8/22.1 | 64.6±1.5/25.1/28.0 |
| RS2 w/ repl (stratified) | 76.1±0.5/13.7/15.3 | 68.7±0.6/21.1/23.5 | 65.0±1.4/24.8/27.7 |
| RS2 w/o repl | **78.7±0.8/13.0/14.1** | **74.4±0.6/17.3/18.9*** | **69.0±0.9/22.7/24.8** |

## C.8    ROBUSTNESS OF RS2 TO NOISY LABELS

In Table 15 we show the robustness of RS2 and existing data pruning methods against noisy labels. We include existing methods which sample static subsets, as well as our modified version of the recent prototype-based data pruning method which utilizes repeated subset selection between each round (SP-Easy-RS) (see Section 5.2). As discussed in Section 5.4 in the main paper, we evaluate the robustness of data pruning methods as follows: We randomly flip some percentage $p$ of the labels in CIFAR10 and then run data pruning methods with these labels. We use a subset selection ratio $r = 10\%$ for all methods and evaluate on the regular test set. For each method we report end-model accuracy/raw accuracy drop compared to $p = 0$/relative accuracy drop compared to $p = 0$ (as a percentage of the $p = 0$ accuracy). Table 15 shows that RS2 achieves higher end-model accuracy than existing data pruning methods in the presence of noisy labels. RS2 is also the most robust method (lowest relative accuracy drop) when the noise ratio is 10% and 30%. Interestingly, the GraNd baseline actually gets better as the noise ratio increases. While surprising, the overall end-model quality of GraNd is still limited, however, as the GraNd accuracy begins to decrease again as the noise increases beyond 50% and all noise ratios result in lower accuracy than training on clean data. We leave a detailed study of these observations and robust data pruning methods for future work.

## C.9    RS2-BASED PRETRAINING OF LANGUAGE MODELS

Next, in Table 16, we use RS2 to reduce the cost of pretraining a large GPT2 language model. We extend RS2 to this setting as follows: we repeatedly sample random subsets of text from the OpenWebText (Gokaslan & Cohen, 2019) dataset and use this data for next token prediction (the standard GPT2 pretraining task). We train RS2 for $r \cdot 600k$ iterations for $r = [0.1, 0.3]$ and compare to training with the full dataset for 600k iterations (recall the connection between RS2 and the number of SGD iterations). We also compare RS2 to random data pruning, i.e., training for $r \cdot 600k$ iterations on a static fraction $r$ of the full dataset selected once before learning begins. *We are not aware of existing data pruning methods being evaluated in this setting and present these results as an initial baseline.* We report accuracy (higher is better) and perplexity (lower is better) on the

Table 16: Zero-shot results of GPT2 pretrained using RS2, a static random subset, and the full dataset. We report accuracy (ACC; higher is better) and perplexity (PPL; lower is better).

| Method | Selection Ratio ($r$) | 2023 AWS Training Cost | LAMBADA (ACC ↑) | Task LAMBADA (PPL ↓) | WikiText103 (PPL ↓) |
|--------|------------------------|------------------------|------------------|-----------------------|----------------------|
| Random | 10% | $520 | 43.37 | 45.02 | 53.36 |
| Random | 30% | $1,560 | 44.63 | 41.99 | 46.44 |
| RS2 w/ repl | 10% | $520 | 44.42 | 41.67 | 45.72 |
| RS2 w/ repl | 30% | $1,560 | 45.29 | 40.51 | 42.58 |
| Full Dataset | - | $5,200 | 46.61 | 40.30 | 40.55 |

LAMBADA (Paperno et al., 2016) benchmark as well as perplexity on WikiText103 (Merity et al., 2016). RS2 leads to better model quality but not cost compared to a static random sample. Moreover, we see that RS2 leads to near matching accuracy and perplexity compared to training using the full dataset for $r = 30\%$. This result highlights the practical potential of RS2 to enable faster and cheaper training, hyperparameter tuning, or neural architecture search for large language model pretraining, currently one of the most expensive and time consuming training paradigms in machine learning.

## D   ADDITIONAL RS2 PSEUDOCODE

In this section, we include additional RS2 pseudocode algorithms to accompany Algorithm 1 presented in the main body of the paper and to present additional details useful for the RS2 theoretical analysis.

In Algorithm 3 we describe RS2 without replacement when training with accelerated mini batch SGD (Ghadimi & Lan, 2016; Nesterov, 1983). Nesterov's accelerated gradient introduces three different sets of parameters that are updated at each iteration $t$. We denote them as $w^t, w_{ag}^t$, and $w_{md}^t$. Furthermore, the algorithm introduces learning rate parameters $\alpha_t, \beta_t$, and $\lambda_t$. In later sections, we specialize the learning rate parameters for obtaining a convergence rate bound. Finally, $g(w, \xi_t; m)$ at step $t$ represents the gradient estimate on a batch of data $m$ that is used for updating the model. $\xi_t$ are random vectors whose distributions are supported on $\Xi_t \in \mathbb{R}^d$.

---

**Algorithm 3** RS2 w/o Replacement With Accelerated Mini batch SGD

---

**Require:** Dataset $S = \{x_i, y_i\}_{i=1}^N$, selection ratio $r \in (0, 1]$, batch size $b$, initial model $w^0$, $X$ rounds, learning rate parameters $\{\alpha_t\}$ s.t. $\alpha_1 = 1, \alpha_t \in (0, 1), \forall t \geq 2, \{\beta_t > 0\}, \{\lambda_t > 0\}$, gradient estimate function for batch $m$ and parameters $w$ with noise $\xi$: $g(w, \xi; m)$
1: $T \leftarrow \lceil N/b \rceil$
2: $t \leftarrow 1$
3: $w_{ag}^0 = w^0$
4: **for** round $j = 1$ to $X$ **do**
5:     **if** $t\%T == 0$ **then**                    ▷ Shuffle after full dataset has been seen
6:         $shuffle(S)$
7:     $S' \leftarrow S[(j-1) \cdot rN : j \cdot rN]$         ▷ Select the subset across rounds without replacement
8:     **for** $k = 1$ to $r \cdot T$ **do**
9:         batch $m \leftarrow S'[(k-1) \cdot b : k \cdot b]$
10:         $w_{md}^t \leftarrow (1 - \alpha_t)w_{ag}^{t-1} + \alpha_t w^{t-1}$
11:         $w^t \leftarrow w^{t-1} - \lambda_t g(w_{md}^t, \xi_t; m)$      ⎫
                                                                      ⎬ ▷ train on batch for Nesterov mini batch SGD
12:         $w_{ag}^t \leftarrow w_{md}^t - \beta_t g(w_{md}^t, \xi_t; m)$   ⎭
13:         $t \leftarrow t + 1$
    **return** $w_{md}^t$

---

In Algorithm 4, we also show RS2 without replacement, but using standard mini batch SGD. We also write this algorithm using a different perspective: instead of iterating over rounds, selecting the training subset for each round, and then iterating over batches in the selected subset, RS2 without replacement can be equivalently implemented by iterating directly over batches from the full dataset, as long as these batches are correctly selected and the full dataset is shuffled as necessary. This perspective can be more useful for understanding the generalization error of RS2 without replacement as it more closely matches the common algorithms in related works analyzing SGD (Ghadimi & Lan, 2016; Nikolakakis et al., 2023).

---

**Algorithm 4** RS2 w/o Replacement With Mini batch SGD; Single For Loop Perspective

---

**Require:** Dataset $S = \{x_i, y_i\}_{i=1}^N$, selection ratio $r \in (0, 1]$, batch size $b$, initial model $w^0$, $X$ rounds, learning rate $\eta_t$
1: $T \leftarrow \lceil N/b \rceil$
2: **for** iterate $t = 1$ to $r \cdot T \cdot X$ **do**
3:     **if** $t\%T == 0$ **then**                    ▷ Shuffle after full dataset has been seen
4:         $shuffle(S)$
5:     batch $m \leftarrow S[(t-1)\%T \cdot b : t\%T \cdot b]$
6:     $w^t \leftarrow w^{t-1} - \frac{\eta_t}{b} \sum_{(x,y) \in m} \nabla f(w^{t-1}; x, y)$      ▷ train on batch for mini batch SGD
    **return** $w^t$

---

## E   RS2 CONVERGENCE RATE

Performance of accelerated mini batch SGD has been well studied for convex functions (Lan, 2012; Dekel et al., 2012; Cotter et al., 2011). It has been shown that mini batch SGD using batch of size $b$,

after $X$ rounds with $T$ batches per round returns a solution $w$ satisfying

$$\mathbb{E}[l(w) - l(w^*)] \leq \mathcal{O}\left(\frac{L||w^0 - w^*||^2}{T^2 X^2} + \frac{\sigma||w^0 - w^*||}{\sqrt{bTX}}\right). \tag{3}$$

Furthermore, Ghadimi & Lan (2016) have analyzed the convergence rate of mini batch SGD for nonconvex $\beta$-smooth functions. After $TX$ mini batch steps of size $b$ the algorithm guarantees a solution $w$ such that

$$\mathbb{E}||\nabla l(w)||^2 \leq \mathcal{O}\left(\frac{\beta(l(w^0) - l(w^*))}{TX} + \frac{\sigma\sqrt{\beta(l(w^0) - l(w^*))}}{\sqrt{bTX}}\right). \tag{4}$$

We provide convergence analysis for accelerated mini batch SGD using RS2 without replacement as shown in Algorithm 3 following the analysis of Ghadimi & Lan (2016).

**Corollary E.1.** *Suppose the loss $l(w)$ is nonconvex, has $\beta$-Lipschitz continuous gradients, and is bounded below. Let $g(w, \xi_t)$ at step $t$ represent the gradient estimate used when updating the model as in Algorithm 3 in the Appendix. Assume the gradient estimate satisfies $\mathbb{E}\left[||g(w, \xi_t) - \nabla l(w)||^2\right] \leq \sigma^2$, and $\mathbb{E}[g(w, \xi_t)] = \nabla l(w)$, where $\xi_t$ are random vectors whose distributions are supported on $\Xi_t \in \mathbb{R}^d$. With the previous assumptions, using a selection ratio $r \in (0, 1]$ and mini batch of size $b$, RS2 produces an iterate $w$ after $X$ rounds, with $rT$ batches per round, such that:*

$$\mathbb{E}\left[||\nabla l(w)||^2\right] \leq \mathcal{O}\left(\frac{\beta(l(w^0) - l(w^*))}{r \cdot T \cdot X} + \frac{\sigma\sqrt{\beta(l(w^0) - l(w^*))}}{\sqrt{b \cdot r \cdot T \cdot X}}\right). \tag{5}$$

*Furthermore, assuming that $l(w)$ is convex it holds that*

$$\mathbb{E}[l(w) - l(w^*)] \leq \mathcal{O}\left(\frac{\beta||w^0 - w^*||^2}{r^2 \cdot T^2 \cdot X^2} + \frac{\sigma||w^0 - w^*||}{\sqrt{b \cdot r \cdot T \cdot X}}\right). \tag{6}$$

We find that the convergence rate of RS2 compared to the full dataset convergence rate (Ghadimi & Lan, 2016) has a scaling factor $r$ in front of the total number of iterates, while the bound remains consistent with respect to all the other parameters; With $r = 1$ we recover the results from previous work (Ghadimi & Lan, 2016). When $r < 1$ the gradient bound after $X$ rounds increases compared to training with the full dataset for $X$ rounds, but this is intuitive as each round contains fewer mini batches ($rT$ with $r < 1$ instead of $T$). If RS2 with $r < 1$ is instead allowed to train for more rounds, specifically $X_{new} = \frac{X}{r}$, then both training RS2 for $X_{new}$ rounds and training on the full dataset for $X$ rounds result in the same number of mini-batch iterations ($TX$). In this case, the gradient is bounded by the same value, implying that RS2 and training on the full dataset converge with respect to *mini-batch iterations* at the same rate.

While the above analysis is a straightforward extension of the convergence result for mini batch SGD with Nesterov's accelerated gradient update, it highlights the stability and guaranteed convergence properties of RS2, a property that does not necessarily hold for other data pruning methods.

*Proof.* Each round we use Nesterov's accelerated method to update the gradient:

$$w_{md}^t \leftarrow (1 - \alpha_t)w_{ag}^{t-1} + \alpha_t w^{t-1} \tag{7}$$

$$w^t \leftarrow w^{t-1} - \lambda_t g(w_{md}^t, \xi_t; m) \tag{8}$$

$$w_{ag}^t \leftarrow w_{md}^t - \beta_t g(w_{md}^t, \xi_t; m) \tag{9}$$

where $g(w_{md}^t, \xi_t)$ represents the gradient on a batch of data $m$. We assume that the following holds:

$$\mathbb{E}g(w, \xi_t) = \nabla l(w) \tag{10}$$

$$\mathbb{E}||g(w, \xi_t) - \nabla l(w)||^2 = \sigma^2, \tag{11}$$

where $\xi_t$ are random vectors whose distributions are supported on $\Xi_t \in \mathbb{R}^d$; These are the source of randomness when estimating the full data gradient.

We repeat the procedure for $X$ rounds. When using full data each round we have $T$ batches, resulting in total $TX$ gradient updates. If we perform RS2 w/o replacement. each round we will contain $rT$

updates per round, resulting in a total of $rTX$ iterations. Assume a relaxation for the learning rate parameters. For this part of the proof assume they are chosen $\{\alpha_t\}$ s.t. $\alpha_1 = 1, \alpha_t \in (0,1), \forall t \geq 2$, $\{\beta_t > 0\}, \{\lambda_t > 0\}$, such that the following holds:

$$\Gamma^t = \begin{cases} 1 & t = 1 \\ (1 - \alpha_t)\Gamma^{t-1} & t \geq 2 \end{cases} \tag{12}$$

$$C^t := 1 - \beta\lambda_t - \frac{\beta(\lambda_t - \beta_t)^2}{2\alpha_t\Gamma^t\lambda_t}\left(\sum_{\tau=t}^{rTX}\Gamma^\tau\right) > 0 \tag{13}$$

$$p_t = \frac{\lambda_t C^t}{\sum_{t=1}^{rTX}\lambda_t C^t}, \ t = 1, ..., rTX. \tag{14}$$

Furthermore, let $R$ represent an index chosen randomly in all the iterate updates from 1 to $rTX$, chosen such that $Prob\{R = t\} = p_t$.

First we want to show the following holds:

$$\mathbb{E}||\nabla l(w_{md}^R)||^2 \leq \frac{1}{\sum_{t=1}^{rTX}\lambda_t C^t}\left[l(w^0) - l(w^*) + \frac{\beta\sigma^2}{2}\sum_{t=1}^{rTX}\lambda_t^2\left(1 + \frac{(\lambda_t - \beta_t)^2}{\alpha_t\Gamma^t\lambda_t^2}\sum_{\tau=t}^{rTX}\Gamma^\tau\right)\right]. \tag{15}$$

Let us define, for ease of writing, the following: $\delta_t := g(w_{md}^t, \xi_t) - \nabla l(w_{md}^t)$ and $\Delta^t := \nabla l(w^{t-1}) - \nabla l(w_{md}^t)$. Since $l(w)$ is bounded from below and a differentiable nonconvex $\beta$-smooth function it holds that (see (Nesterov, 2003)):

$$|l(y) - l(x) - \langle\nabla l(x), y - x\rangle| \leq \frac{\beta}{2}||y - x||^2 \quad \forall x, y \in \mathbb{R}^n. \tag{16}$$

We start from the assumption that the loss function $l$ is $\beta$-smooth:

$$l(w^t) \leq l(w^{t-1}) + \langle\nabla l(w^{t-1}), w^t - w^{t-1}\rangle + \frac{\beta}{2}||w^t - w^{t-1}||. \tag{17}$$

Then, using the update step eq. (8) and the definitions of $\delta_t, \Delta^t$:

$$l(w^t) \leq l(w^{t-1}) + \langle\Delta^t + \nabla l(w_{md}^t), -\lambda_t[\nabla l(w_{md}^t) + \delta^t]\rangle + \frac{\beta\lambda_t^2}{2}||\nabla l(w_{md}^t) + \delta^t||^2$$

$$= l(w^{t-1}) + \langle\Delta^t + \nabla l(w_{md}^t), -\lambda_t\nabla l(w_{md}^t)\rangle - \lambda_t\langle\nabla l(w^{t-1}), \delta^t\rangle + \frac{\beta\lambda_t^2}{2}||\nabla l(w_{md}^t) + \delta^t||^2. \tag{18}$$

Now using the inequality eq. (16) we get:

$$l(w^t) \leq l(w^{t-1}) - \lambda_t\left(1 - \frac{\beta\delta_t}{2}\right)||\nabla l(w_{md}^t)||^2 + \lambda_t||\Delta^t||||\nabla l(w_{md}^t)|| + \frac{\beta\lambda_t^2}{2}||\delta^t||^2$$

$$- \lambda_t\langle\nabla l(w^{t-1}) - \beta\lambda_t\nabla l(w_{md}^t), \delta^t\rangle. \tag{19}$$

Since $l$ is $\beta$-smooth and by the update rule eq. (7) we have:

$$||\Delta^t|| = ||\nabla l(w^{t-1}) - \nabla l(w_{md}^t)|| \leq \beta||w^{t-1} - w_{md}^t|| = \beta(1 - \alpha_t)||w_{ag}^t - w^{t-1}||. \tag{20}$$

Continuing from eq. (18) and inserting eq. (20):

$$l(w^t) \leq l(w^{t-1}) - \lambda_t\left(1 - \frac{\beta\delta_t}{2}\right)||\nabla l(w_{md}^t)||^2 + \lambda_t\beta(1 - \alpha_t)||w_{ag}^t - w^{t-1}||||\nabla l(w_{md}^t)||$$

$$+ \frac{\beta\lambda_t^2}{2}||\delta^t||^2 - \lambda_t\langle\nabla l(w^{t-1}) - \beta\lambda_t\nabla l(w_{md}^t), \delta^t\rangle. \tag{21}$$

Using the general fact that $xy \leq \frac{(x^2 + y^2)}{2}$ holds, we bound the previous inequality:

$$l(w^t) \leq l(w^{t-1}) - \lambda_t(1 - \beta\lambda_t)||\nabla l(w_{md}^t)||^2 + \frac{\beta(1 - \alpha_t)^2}{2}||w_{ag}^{t-1} - w^{t-1}||^2 + \frac{\beta\lambda_t^2}{2}||\delta^t||^2$$

$$- \lambda_t\langle\nabla l(w^{t-1}) - \beta\lambda_t\nabla l(w_{md}^t), \delta^t\rangle. \tag{22}$$

Now we take a small digression from the main flow of the proof. We want to show that the following inequality holds:

$$||w_{ag}^{t-1} - w^{t-1}||^2 \leq \Gamma^{t-1} \sum_{\tau=1}^{t-1} \frac{(\lambda_\tau - \beta_\tau)^2}{\Gamma^\tau \alpha_\tau} ||\nabla l(w_{md}^\tau) + \delta^t||^2$$

$$= \Gamma^{t-1} \sum_{\tau=1}^{t-1} \frac{(\lambda_\tau - \beta_\tau)^2}{\Gamma^\tau \alpha_\tau} \left[ ||\nabla l(w_{md}^\tau)||^2 + 2\langle \nabla l(w_{md}^\tau), \delta^\tau \rangle + ||\delta^\tau||^2 \right]. \quad (23)$$

We show that in the following way. First, let us combine the update steps eqs. (7) to (9). Performing change of variable we have:

$$w_{ag}^t - w^t = (1 - \alpha_t)w_{ag}^{t-1} + \alpha_t w^{t-1} - \beta_t \nabla l(w_{md}^t) - [w^{t-1} - \lambda_t \nabla l(w_{md}^t)]$$

$$= (1 - \alpha_t)(w_{ag}^{t-1} - w^{t-1}) + (\lambda_t - \beta_t)\nabla l(w_{md}^t). \quad (24)$$

Following from eq. (24) and using Lemma 1 stated in Ghadimi & Lan (2016) it is implied that:

$$w_{ag}^t - w^t = \Gamma^t \sum_{\tau=1}^t \frac{\lambda_\tau - \beta_\tau}{\Gamma^\tau} \nabla l(w_{md}^\tau). \quad (25)$$

Furthermore, we have:

$$||w_{ag}^t - w^t||^2 = \left\| \Gamma^t \sum_{\tau=1}^t \frac{\lambda_\tau - \beta_\tau}{\Gamma^\tau} \nabla l(w_{md}^\tau) \right\|^2. \quad (26)$$

From the definition in eq. (12) we have:

$$\sum_{\tau=1}^t \frac{\alpha_\tau}{\Gamma^\tau} = \frac{\alpha_1}{\Gamma^1} + \sum_{\tau=2}^t \frac{1}{\Gamma^\tau}\left(1 - \frac{\Gamma^\tau}{\Gamma^{\tau-1}}\right) = \frac{1}{\Gamma^1} + \sum_{\tau=2}^t \left(\frac{1}{\Gamma^\tau} - \frac{1}{\Gamma^{\tau-1}}\right) = \frac{1}{\Gamma^t}. \quad (27)$$

Inserting that into eq. (26) we get:

$$||w_{ag}^t - w^t||^2 = \left\| \Gamma^t \sum_{\tau=1}^t \frac{\alpha_\tau}{\Gamma^\tau} \frac{\lambda_\tau - \beta_\tau}{\alpha^\tau} \nabla l(w_{md}^\tau) \right\|^2. \quad (28)$$

Applying Jensen's inequality to eq. (28) we have:

$$||w_{ag}^t - w^t||^2 \leq \Gamma^t \sum_{\tau=1}^t \frac{\alpha_\tau}{\Gamma^\tau} \left\| \frac{\lambda_\tau - \beta_\tau}{\alpha^\tau} \nabla l(w_{md}^\tau) \right\|^2 = \Gamma^t \sum_{\tau=1}^t \frac{(\lambda_\tau - \beta_\tau)^2}{\Gamma^\tau \alpha_\tau} ||\nabla l(w_{md}^\tau)||^2. \quad (29)$$

Hence, eq. (23) holds.

Coming back to the main flow of the proof. We combine the previous two inequalities eq. (22) and eq. (23). Also, we use the fact that $\Gamma^{t-1}(1 - \alpha_t)^2 \leq \Gamma^t$:

$$l(w^t) \leq l(w^{t-1}) - \lambda_t(1 - \beta\lambda_t)||\nabla l(w_{md}^t)||^2 + \frac{\beta\lambda_t^2}{2}||\delta^t||^2 - \lambda_t\langle \nabla l(w^{t-1}) - \beta\lambda_t \nabla l(w_{md}^t), \delta^t \rangle$$

$$+ \frac{\beta\Gamma^t}{2} \sum_{\tau=1}^t \frac{(\lambda_\tau - \beta_\tau)^2}{\Gamma^\tau \alpha_\tau} \left[ ||\nabla l(w_{md}^\tau)||^2 + 2\langle \nabla l(w_{md}^\tau), \delta^\tau \rangle + ||\delta^\tau||^2 \right]. \quad (30)$$

Summing up the above inequalities (eq. (30)) up to the rTX iterate, we get:

$$l(w^{rTX}) \leq l(w^0) - \sum_{t=1}^{rTX} \lambda_t(1 - \beta\lambda_t)||\nabla l(w_{md}^t)||^2 - \sum_{t=1}^{rTX} \lambda_t\langle \nabla l(w^{t-1}) - \beta\lambda_t \nabla l(w_{md}^t), \delta^t \rangle$$

$$+ \sum_{t=1}^{rTX} \frac{\beta\lambda_t^2}{2}||\delta^t||^2 - \frac{\beta}{2} \sum_{t=1}^{rTX} \Gamma^t \sum_{\tau=1}^t \frac{(\lambda_\tau - \beta_\tau)^2}{\Gamma^\tau \alpha_\tau} \left[ ||\nabla l(w_{md}^\tau)||^2 + 2\langle \nabla l(w_{md}^\tau), \delta^\tau \rangle + ||\delta^\tau||^2 \right]$$

$$= l(w^0) - \sum_{t=1}^{rTX} \lambda_t C^t ||\nabla l(w_{md}^t)||^2 + \frac{\beta}{2} \sum_{t=1}^{rTX} \lambda_t^2 \left(1 + \frac{(\lambda_t - \beta_t)^2}{\alpha_t \Gamma^t \lambda_t^2} \sum_{\tau=t}^{rTX} \Gamma^\tau \right) ||\delta^t||^2 - \sum_{t=1}^{rTX} b_t, \quad (31)$$

where $b_t = \langle \lambda_t \nabla l(w^{t-1}) - \left[ \beta \lambda_t^2 + \frac{\beta(\lambda_t - \beta_t)^2}{\Gamma^t \alpha_t} \left( \sum_{\tau=t}^{rTX} \Gamma^\tau \right) \right] \nabla l(w_{md}^t), \delta^t \rangle$. Due to the fact that under assumptions eqs. (10) and (11) $\mathbb{E}||\delta^t||^2 \leq \sigma^2$ and $\{b_t\}$ is a martingale difference, when taking expectation on both sides we obtain:

$$\sum_{t=1}^{rTX} \lambda_t C^t \mathbb{E}||\nabla l(w_{md}^t)||^2 \leq l(w^0) - l(w^{rTX}) + \frac{\beta \sigma^2}{2} \sum_{t=1}^{rTX} \lambda_t^2 \left( 1 + \frac{(\lambda_t - \beta_t)^2}{\alpha_t \Gamma^t \lambda_t^2} \sum_{\tau=t}^{rTX} \Gamma^\tau \right). \quad (32)$$

Using the fact that $l(w^t) \geq l(w^*)$, $\mathbb{E}||\nabla l(w_{md}^R)||^2 = \frac{\sum_{t=1}^{rTX} \lambda_t C^t \mathbb{E}||\nabla l(w_{md}^t)||^2}{\sum_{t=1}^{rTX} \lambda_t C^t}$, and by dividing both sides by $\sum_{t=1}^{rTX} \lambda_t C^t$, we obtain:

$$\mathbb{E}||\nabla l(w_{md}^R)||^2 \leq \frac{1}{\sum_{t=1}^{rTX} \lambda_t C^t} \left[ l(w^0) - l(w^*) + \frac{\beta \sigma^2}{2} \sum_{t=1}^{T} \lambda_t^2 \left( 1 + \frac{(\lambda_t - \beta_t)^2}{\alpha_t \Gamma^t \lambda_t^2} \sum_{\tau=t}^{rTX} \Gamma^\tau \right) \right]. \quad (33)$$

Hence, we have proven the wanted eq. (15) holds.

For the remainder of the proof for the nonconvex case we specialize the previously obtained result. Let us assume the following:

$$\alpha_t = \frac{2}{t+1} \quad (34)$$

$$\lambda_t \in \left[ \beta_t, \left( 1 + \frac{\alpha_t}{4} \right) \beta_t \right] \quad (35)$$

$$\Gamma^t = \frac{2}{t(t+1)} \quad (36)$$

$$\beta_t = \min \left\{ \frac{8}{21\beta}, \frac{\tilde{D}}{\sigma \sqrt{rTX}} \right\} \text{ for some } \tilde{D} > 0. \quad (37)$$

Now, we want to prove:

$$\mathbb{E}||\nabla l(w_{md}^R)||^2 \leq \frac{21\beta(l(w^0) - l(w^*))}{4rTX} + \frac{2\sigma}{\sqrt{rTX}} \left( \frac{l(w^0) - l(w^*)}{\tilde{D}} + \beta \tilde{D} \right). \quad (38)$$

From definition of eq. (34), eq. (35) let us make a claim about $C^t$. For that, from eq. (35) we observe $0 \leq \lambda_t - \beta_t \leq \alpha_t \beta_t / 4$. Now we have:

$$C^t = 1 - \beta \left[ \lambda_t + \frac{(\lambda_t - \beta_t)^2}{2\alpha_t \Gamma^t \lambda_t} \left( \sum_{\tau=t}^{rTX} \Gamma^\tau \right) \right] \quad (39)$$

$$\geq 1 - \beta \left[ \left( 1 + \frac{\alpha_t}{4} \right) \beta_t + \frac{\alpha_t^2 \beta_t^2}{16} \frac{1}{t \alpha_t \Gamma^t \beta_t} \right] \quad (40)$$

$$= 1 - \beta_t \beta (1 + \frac{\alpha_t}{4} + \frac{1}{16}) \quad (41)$$

$$\geq 1 - \beta_t \beta \frac{21}{16}. \quad (42)$$

Multiplied by $\lambda_t$ we have $\lambda_t C^t \geq \frac{11\beta_t}{32}$.

Now we make the following claim about $\Gamma^t$. From eq. (36):

$$\sum_{\tau=t}^{rTX} \Gamma^\tau = \sum_{\tau=t}^{rTX} \frac{2}{\tau(\tau+1)} = 2 \sum_{\tau=t}^{rTX} \left( \frac{1}{\tau} - \frac{1}{\tau+1} \right) \leq \frac{2}{t}. \quad (43)$$

From the eq. (35), eq. (37), eq. (42), we have:

$$C^t \geq 1 - \frac{21}{16} \beta \beta_t \geq \frac{1}{2} > 0 \quad \text{and} \quad \lambda_t C^t \geq \frac{\beta_t}{2}. \quad (44)$$

Furthermore, from eq. (35), eq. (36), eq. (37), and eq. (43), we obtain:

$$\lambda_t^2 \left[ 1 + \frac{(\lambda_t - \beta_t)^2}{\alpha_t \Gamma^t \lambda_t^2} \left( \sum_{\tau=t}^{rTX} \Gamma^\tau \right) \right] \le \lambda_t^2 \left[ 1 + \frac{1}{\alpha_t \Gamma^t \lambda_t^2} \left( \frac{\alpha_t \beta_t}{4} \right)^2 \frac{2}{t} \right] = \lambda_t^2 + \frac{\beta_t^2}{8}$$

$$\le \left[ \left( 1 + \frac{\alpha_t}{4} \right)^2 + \frac{1}{8} \right] \beta_t^2 \le 2\beta_t^2. \tag{45}$$

Together with eq. (33) it holds that:

$$\mathbb{E}||\nabla l(w_{md}^R)||^2 \le \frac{2}{\sum_{t=1}^{rTX} \beta_t} \left( l(w^0) - l(w^*) + \beta \sigma^2 \sum_{t=1}^{rTX} \beta_t^2 \right)$$

$$\le \frac{2(l(w^0) - l(w^*))}{rTX\beta_1} + 2\beta\sigma^2\beta_1$$

$$\le \frac{2(l(w^0) - l(w^*))}{rTX} \left\{ \frac{21\beta}{8} + \frac{\sigma\sqrt{rTX}}{\tilde{D}} \right\} + \frac{2\beta\tilde{D}\sigma}{\sqrt{rTX}}, \tag{46}$$

which implies:

$$\mathbb{E}||\nabla l(w_{md}^R)||^2 \le \frac{21\beta(l(w^0) - l(w^*))}{4rTX} + \frac{2\sigma}{\sqrt{rTX}} \left( \frac{l(w^0) - l(w^*)}{\tilde{D}} + \beta\tilde{D} \right). \tag{47}$$

Hence, we have shown that eq. (47) holds. Continuing from that, minimizing eq. (47) with respect to $\tilde{D}$, the optimal choice is $\tilde{D} = \sqrt{\frac{l(w_{ag}^0) - l(w^*)}{\beta}}$. Inserting that value for $\tilde{D}$, eq. (47) becomes:

$$\mathbb{E}||\nabla l(w_{md}^R)||^2 \le \frac{21\beta(l(w^0) - l(w^*))}{4rTX} + \frac{4\sigma\sqrt{\beta(l(w^0) - l(w^*)))}}{\sqrt{rTX}}. \tag{48}$$

Until now we have assumed that $\mathbb{E}||g(w, \xi_t) - \nabla l(w)||^2 = \sigma^2$ for the ease of the proof. However, if we assume that the gradient is calculated on a batch of size $b$, the variance of the stochastic gradient reduces to $\sigma^2/b$ (see (Wang & Srebro, 2019)). The entire previous results follow with that assumption without loss of generality. Therefore we conclude it holds that:

$$\mathbb{E}||\nabla l(w)||^2 \le \mathcal{O} \left( \frac{\beta(l(w^0) - l(w^*))}{rTX} + \frac{\sigma\sqrt{\beta(l(w^0) - l(w^*))}}{\sqrt{brTX}} \right). \tag{49}$$

**Convex case** Now, let us consider the case for convex functions. First, in order to prove eq. (4), we want to show that, assuming:

$$\alpha_t \lambda_t \le \beta\beta_t^2, \quad \beta_t < \frac{1}{\beta}, \tag{50}$$

$$p_t = \frac{\frac{1}{\Gamma^t}\beta_t(1 - \beta\beta_t)}{\sum_{t=1}^{rTX} \frac{1}{\Gamma^t}\beta_t(1 - \beta\beta_t)}, \tag{51}$$

and

$$\frac{\alpha_1}{\lambda_1 \Gamma^1} \ge \frac{\alpha_2}{\lambda_2 \Gamma^2} \ge \dots \tag{52}$$

the following holds:

$$\mathbb{E}[l(w_{ag}^R - l(w^*))] \le \frac{\sum_{t=1}^{rTX} \beta_t(1 - \beta\beta_t) \left[ (2\lambda_1)^{-1}||w^0 - w^*||^2 + \beta\sigma^2 \sum_{j=1}^{t} \frac{\beta_j^2}{\Gamma^j} \right]}{\sum_{t=1}^{rTX} \frac{\beta_t}{\Gamma^t}(1 - \beta\beta_t)}. \tag{53}$$

Starting from the update rule eq. (7) and using the convexity of $l(\cdot)$ we have:

$$l(w_{md}^t) - [(1 - \alpha_t)l(w_{ag}^{t-1}) + \alpha_t l(w)] = \alpha_t \left[ l(w_{md}^t) - l(w) \right] + (1 - \alpha_t) \left[ l(w_{md}^t - l(w_{ag}^{t-1})) \right]$$

$$\le \alpha_t\langle\nabla l(w_{md}^t), w_{md}^t - w\rangle + (1 - \alpha_t)\langle\nabla l(w_{md}^t), w_{md}^t - w_{ag}^{t-1}\rangle$$

$$= \langle\nabla l(w_{md}^t), \alpha_t(w_{md}^t - w) + (1 - \alpha_t)(w_{md}^t - w_{ag}^{t-1})\rangle$$

$$= \alpha_t\langle\nabla l(w_{md}^t), w^{t-1} - w\rangle. \tag{54}$$

Similar to before, we now start with the smoothness eq. (16) and use the update step eq. (9) to obtain:

$$l(w_{ag}^t) \leq l(w_{md}^t) + \langle \nabla l(w_{md}^t), w_{ag}^t - w_{md}^t \rangle + \frac{\beta}{2} ||w_{ag}^t - w_{md}^t||^2$$

$$= l(w_{md}^t) - \beta_t ||\nabla l(w_{md}^t)||^2 + \beta_t \langle \nabla l(w_{md}^t), \delta^t \rangle + \frac{\beta \beta_t^2}{2} ||\nabla l(w_{md}^t) + \delta^t||^2. \quad (55)$$

Inserting eq. (54) into the previous inequality, we have:

$$l(w_{ag}^t) \leq (1 - \alpha_t) l(w_{ag}^{t-1}) + \alpha_t l(w) + \alpha_t \langle \nabla l(w_{md}^t), w^{t-1} - w \rangle$$

$$- \beta_t ||\nabla l(w_{md}^t)||^2 + \beta_t \langle \nabla l(w_{md}^t), \delta^t \rangle + \frac{\beta \beta_t^2}{2} ||\nabla l(w_{md}^t) + \delta^t||. \quad (56)$$

From eq. (9) we have:

$$||w^{t-1} - w||^2 - 2\lambda_t \langle \nabla l(w_{md}^t) + \delta^t, w^{t-1} - w \rangle$$
$$+ \lambda_t^2 ||\nabla l(w_{md}^t) + \delta^t||^2 = ||w^{t-1} - \lambda_t(\nabla l(w_{md}^t) + \delta^t) - w||^2 = ||w^t - w||^2. \quad (57)$$

From the previous equation, we have:

$$\alpha_t \langle \nabla l(w_{md}^t) + \delta^t, w^{t-1} - w \rangle = \frac{\alpha_t}{2\lambda_t} \left[ ||w^{t-1} - w||^2 - ||w^t - w||^2 \right] + \frac{\alpha_t \lambda_t}{2} ||\nabla l(w_{md}^t) + \delta^t||^2. \quad (58)$$

Combining eqs. (56) and (58) and the fact that $||\nabla l(w_{md}^t) + \delta^t||^2 = ||\nabla l(w_{md}^t)||^2 + ||\delta^t||^2 + 2\langle \nabla l(w_{md}^t), \delta^t \rangle$, we get:

$$l(w_{ag}^t) \leq (1 - \alpha_t) l(w_{ag}^{t-1}) + \alpha_t l(w) + \frac{\alpha_t}{2\lambda_t} \left[ ||w^{t-1} - w||^2 - ||w^t - w||^2 \right]$$

$$- \beta_t \left( 1 - \frac{\beta \beta_t}{2} - \frac{\alpha_t \lambda_t}{2\beta_t} \right) ||\nabla l(w_{md}^t)||^2 + \left( \frac{\beta \beta_t^2 + \alpha_t \lambda_t}{2} \right) ||\delta^t||^2$$

$$+ \langle \delta^t, (\beta_t + \beta \beta_t^2 + \alpha_t \lambda_t) \nabla l(w_{md}^t) + \alpha_t(w - w^{t-1}) \rangle. \quad (59)$$

Due to the fact that $\alpha_1 = \Gamma^1 = 1$ and by eq. (52) it holds that:

$$\sum_{t=1}^{rTX} \frac{\alpha_t}{\lambda_t \Gamma^t} \left[ ||w^{t-1} - w||^2 - ||w^t - w||^2 \right] \leq \frac{\alpha_1 ||w^0 - w||^2}{\lambda_1 \Gamma^1} = \frac{||w^0 - w||^2}{\lambda_1}. \quad (60)$$

Using Lemma 1 from Ghadimi & Lan (2016), eq. (60) and subtracting $l(w)$ from eq. (59), we obtain:

$$\frac{l(w_{ag}^{rTX}) - l(w)}{\Gamma^{rTX}} \leq \frac{||w^0 - w||^2}{2\lambda_1} - \sum_{t=1}^{rTX} \frac{\beta_t}{2\Gamma^t} \left( 2 - \beta \beta_t - \frac{\alpha_t \lambda_t}{\beta_t} \right) ||\nabla l(w_{md}^t)||^2$$

$$+ \sum_{t=1}^{rTX} \left( \frac{\beta \beta_t^2 + \alpha_t \lambda_t}{2\Gamma^t} \right) ||\delta^t||^2 + \sum_{t=1}^{rTX} b_t', \quad (61)$$

where $b_t' = \frac{1}{\Gamma^t} \langle \delta^t, (\beta_t + \beta \beta_t^2 + \alpha_t \lambda_t) \nabla l(w_{md}^t) + \alpha_t(w - w^{t-1}) \rangle$. Together with eq. (50) the above inequality gives:

$$\frac{l(w_{ag}^{rTX}) - l(w)}{\Gamma^{rTX}} \leq \frac{||w^0 - w||^2}{2\lambda_1} - \sum_{t=1}^{rTX} \frac{\beta_t}{\Gamma^t} (1 - \beta \beta_t) ||\nabla l(w_{md}^t)||^2$$

$$+ \sum_{t=1}^{rTX} \frac{\beta \beta_t^2}{\Gamma^t} ||\delta^t||^2 + \sum_{t=1}^{rTX} b_t'. \quad (62)$$

Since $\{b_t'\}$ is a martingale difference, $\mathbb{E}||\delta^t||^2 \leq \sigma^2$ and by taking expectation with respect to $\xi_{[rTX]}$, we have:

$$\frac{1}{\Gamma^{rTX}} \mathbb{E} \left[ l(w_{ag}^{rTX}) - l(w) \right] \leq \frac{||w^0 - w||^2}{2\lambda_1} - \sum_{t=1}^{rTX} \frac{\beta_t}{\Gamma^t} (1 - \beta \beta_t) \mathbb{E}||\nabla l(w_{md}^t)||^2 + \sigma^2 \sum_{t=1}^{rTX} \frac{\beta \beta_t^2}{\Gamma^t}. \quad (63)$$

Now, assume $w = w^*$ and since by definition $l(w_{ag}^{rTX}) \geq l(w^*)$, we obtain:

$$\sum_{t=1}^{rTX} \frac{\beta_t}{\Gamma^t}(1 - \beta\beta_t)\mathbb{E}||\nabla l(w_{md}^t)||^2 \leq \frac{||w^0 - w^*||^2}{2\lambda_1} + \sigma^2 \sum_{t=1}^{rTX} \frac{\beta\beta_t^2}{\Gamma^t}, \tag{64}$$

from which, using the definition of $w_{md}^R$, it follows that

$$\mathbb{E}||\nabla l(w_{md}^R)||^2 \leq \frac{(2\lambda_1)^{-1}||w^0 - w^*||^2 + \beta\sigma^2 \sum_{t=1}^{rTX} \frac{\beta_t^2}{\Gamma^t}}{\sum_{t=1}^{rTX} \frac{\beta_t}{\Gamma^t}(1 - \beta\beta_t)}. \tag{65}$$

Also, using eq. (50) and eq. (63) in eq. (64), for $rTX \geq 1$ we have:

$$\mathbb{E}\left[l(w_{ag}^{rTX} - l(w^*))\right] \leq \Gamma^{rTX}\left(\frac{||w^0 - w||^2}{2\lambda_1} + \sigma^2 \sum_{t=1}^{rTX} \frac{\beta\beta_t^2}{\Gamma^t}\right),$$

which implies that eq. (53) holds:

$$\mathbb{E}[l(w_{ag}^R - l(w^*))] = \sum_{t=1}^{rTX} \frac{\frac{\beta_t}{\Gamma^t}(1 - \beta\beta_t)}{\sum_{t=1}^{rTX} \frac{\beta_t}{\Gamma^t}(1 - \beta\beta_t)}\mathbb{E}[l(w_{ag}^t - l(w^*))]$$

$$\leq \frac{\sum_{t=1}^{rTX} \beta_t(1 - \beta\beta_t)\left[(2\lambda_1)^{-1}||w^0 - w||^2 + \beta\sigma^2 \sum_{j=1}^{t} \frac{\beta_j^2}{\Gamma^j}\right]}{\sum_{t=1}^{rTX} \frac{\beta_t}{\Gamma^t}(1 - \beta\beta_t)}. \tag{66}$$

Now, assuming $\alpha_t$ is set as in eq. (34), $p_t$ is set as in eq. (51),

$$\beta_t = \min\left\{\frac{1}{2\beta}, \left(\frac{\tilde{D}^2}{\beta^2\sigma^2(rTX)^3}\right)^{1/4}\right\}, \tag{67}$$

and

$$\lambda_t = \frac{t\beta\beta_t^2}{2}, \tag{68}$$

we want to show that the following inequality holds:

$$\mathbb{E}[l(w_{ag}^R) - l(w^*)] \leq \frac{48\beta||w^0 - w^*||^2}{r^2T^2X^2} + \frac{12\sigma}{\sqrt{rTX}}\left(\frac{||w^0 - w^*||^2}{\tilde{D}} + \tilde{D}\right), \tag{69}$$

for some $\tilde{D} > 0$. Note, that eq. (67) and eq. (68) imply eq. (52) and eq. (50).

Since, $\Gamma^t = \frac{2}{t(t+1)}$, and by eq. (67), we obtain:

$$\sum_{t=1}^{rTX} \frac{\beta_t}{\Gamma^t}(1 - \beta\beta_t) \geq \frac{1}{2}\sum_{t=1}^{rTX} \frac{\beta_t}{\Gamma^t} = \frac{\beta_1}{2}\sum_{t=1}^{rTX} \frac{1}{\Gamma^t} \tag{70}$$

$$\sum_{t=1}^{rTX} \frac{1}{\Gamma^t} \geq \sum_{t=1}^{rTX} \frac{t^2}{2} = \frac{1}{12}rTX(rTX + 1)(2rTX + 1) \geq \frac{1}{6}r^3T^3X^3. \tag{71}$$

By $\Gamma^t = \frac{2}{t(t+1)}$, eqs. (65), (67) and (68), we obtain:

$$\mathbb{E}||\nabla l(w_{md}^R)||^2 \leq \frac{2}{\beta_1 \sum_{t=1}^{rTX} \frac{1}{\Gamma^t}}\left(\frac{||w^0 - w^*||^2}{\beta\beta_1^2} + \beta\sigma^2\beta_1^2 \sum_{t=1}^{rTX} \frac{1}{\Gamma^t}\right)$$

$$= \frac{2||w^0 - w^*||^2}{\beta\beta_1^3 \sum_{t=1}^{rTX} \frac{1}{\Gamma^t}} + 2\beta\sigma^2\beta_1 \leq \frac{12||w^0 - w^*||^2}{\beta r^3T^3X^3\beta_1^3} + 2\beta\sigma^2\beta_1$$

$$\leq \frac{96\beta^2||w^0 - w^*||^2}{r^3T^3X^3} + \frac{\beta^{1/2}\sigma^{3/2}}{(rTX)^{3/4}}\left(\frac{12||w^0 - w^*||^2}{\tilde{D}^{3/2}} + 2\tilde{D}^{1/2}\right). \tag{72}$$

Moreover, by eq. (67), it holds that:

$$1 - \beta\beta_t \leq 1 \quad \text{and} \quad \sum_{j=1}^{t} \frac{1}{\Gamma^j} = \frac{1}{2}\sum_{j=1}^{t} j(j+1) \leq \sum_{j=1}^{t} j^2 \leq t^3.$$

It is implied by eqs. (66), (67), (70) and (71) that:

$$\mathbb{E}[l(w_{ag}^R) - l(w^*)] \leq \frac{2}{\sum_{t=1}^{rTX} \frac{1}{\Gamma^t}} \left[ rTX(2\lambda_1)^{-1}||w^0 - w^*||^2 + \beta\sigma^2\beta_1^2 \sum_{t=1}^{rTX} t^3 \right]$$

$$\leq \frac{12||w^0 - w^*||^2}{r^2T^2X^2\beta\beta_1^2} + \frac{12\beta\sigma^2\beta_1^2}{r^3T^3X^3} \sum_{t=1}^{rTX} t^3$$

$$\leq \frac{12||w^0 - w^*||^2}{r^2T^2X^2\beta\beta_1^2} + 12\beta\sigma^2\beta_1^2 rTX$$

$$\leq \frac{48\beta||w^0 - w^*||^2}{r^2T^2X^2} + \frac{12\sigma}{\sqrt{rTX}} \left( \frac{||w^0 - w^*||^2}{\tilde{D}} + \tilde{D} \right). \tag{73}$$

This shows that eq. (69) holds. Minimizing the previous inequality with respect to $\tilde{D}$, the optimal choice is $\tilde{D} = ||w^0 - w^*||$. Hence, it becomes:

$$\mathbb{E}[l(w_{ag}^R) - l(w^*)] \leq \frac{48\beta||w^0 - w^*||^2}{r^2T^2X^2} + \frac{24||w^0 - w^*||\sigma}{\sqrt{rTX}}. \tag{74}$$

As in Wang & Srebro (2019), the variance of the stochastic gradient reduces to $\sigma^2/b$ when estimating with $b$ samples. Therefore we conclude it holds that:

$$\mathbb{E}[l(w) - l(w^*)] \leq \mathcal{O}\left( \frac{\beta||w^0 - w^*||^2}{r^2T^2X^2} + \frac{\sigma||w^0 - w^*||}{\sqrt{brTX}} \right). \tag{75}$$

$\square$

# F  RS2 GENERALIZATION ERROR

We proceed with the generalization error bound of RS2 for nonconvex Lipschitz and smooth losses. We start by introducing the assumptions on the function $f : \mathbb{R}^d \times \mathcal{Z} \to \mathbb{R}^+$ (see Section 4) for completeness, and then we proceed with the proof of Theorem 4.1.

**Assumption. (Smooth and Lipschitz Loss)**  There exist constants $\beta_f \geq 0$ and $L_f \geq 0$, such that for all $w, u \in \mathbb{R}^d$ and $x \in \mathcal{X}$, it is true that $||\nabla_w f(w, z) - \nabla_u f(u, z)||_2 \leq \beta_f ||w - u||_2$ and $||f(w, z) - f(u, z)||_2 \leq L_f ||w - u||_2$.

The next result follows form prior work (Nikolakakis et al., 2023) and it suffices to show that RS2 sampling is data independent and belongs to the set of general mini batch schedules that appear in (Nikolakakis et al., 2023, Definition 1).

**Theorem F.1** (Generalization error of standard gradient RS2, (Nikolakakis et al., 2023) Theorem 8)**.** *Let the function $f$ be nonconvex, $L_f$-Lipschitz and $\beta_f$-smooth. Then the generalization error of the standard gradient RS2 algorithm with a decreasing step-size $\eta_t \leq C/t$ (for $C < 1/\beta_f$), is bounded as:*

$$|\epsilon_{\text{gen}}(f, \mathcal{D}, \text{RS2})| \leq \frac{1}{N} \cdot 2Ce^{C\beta_f} L_f^2 (r \cdot T \cdot X)^{C\beta_f} \min\left\{1 + \frac{1}{C\beta_f}, \log(e \cdot r \cdot T \cdot X)\right\}. \tag{76}$$

*Proof.* Let $\{k_1^j, \ldots, k_b^j\} \subset \{1, 2, \ldots, N\}^b$ be the set of indices for mini batch selection at each gradient step $j \in \{1, \ldots, rTX\}$. We select the mini batch through the choice of indices $\{k_1^j, \ldots, k_b^j\}$ as follows. For sampling with replacement in line 4 of Algorithm 1 at round $j$ selects a subset of indices $\{k_1^j, \ldots, k_{rN}^j\}$. These indices are sampled independently from any other round $i \in$

$\{1, \ldots, X\}$, i.e., the same indices can be sampled in consecutive rounds, hence with replacement. Note, that at round $j$ sampled indices in the set are unique. The parameters are then updated using a deterministic batch schedule iterating through the sampled subset of indices resulting in $rT$ gradient updates. On the contrary, RS2 without replacement can be seen as traversing the full dataset in a deterministic round-Robin fashion. That is, the model parameters are updated by sequentially selecting indices $\{k_1^j, \ldots, k_b^j\}$. After iterating over the full dataset, i.e., after $T = N/b$ gradient updates we shuffle the full dataset array and repeat the procedure (e.g., Algorithm 4). The algorithm early stops after $rTX$ gradient updates. Thus the selection rule is non-adaptive and data-independent and it belongs to the set of the general batch schedules (Nikolakakis et al., 2023, Definition 1). As a consequence, (Nikolakakis et al., 2023, Lemma 2) and the growth recursion (Nikolakakis et al., 2023, Lemma 3 (Growth Recursion)) holds verbatim for RS2 with standard gradient training with batch size $b$. Then, we solve the recursion identically to (Nikolakakis et al., 2023, Proof of Theorem 8)) for $rTX$ total number of gradient steps. The solution of the recursion gives (the on-average stability and thus) the generalization error bound of RS2, as appears in the theorem, and completes the proof. $\quad\square$

