# OpenReview forum: "Repeated Random Sampling for Minimizing the Time-to-Accuracy of Learning"
_ICLR.cc/2024/Conference — ICLR 2024 poster_

### Official Review · Reviewer_MpXN · 2023-10-31

**Soundness:** 3 good
**Presentation:** 3 good
**Contribution:** 3 good
**Rating:** 6
**Confidence:** 3

**Summary:**

This paper revisits a rather simple subset selection strategy for efficient deep learning. The authors claim that repeated sampling of random subsets (RS2), that is, only randomly sample subsets at each round, can be a powerful baseline strategy. RS2 is competitive against most of the sampling strategies as well as dataset distillation methods developed previously. Besides, there is no additional computation cost extracting the coreset using RS2, so RS2 reaches the best acceleration under the same sample budget.

**Strengths:**

1.	Selecting the coreset for efficient deep learning is important for machine learning practices. The paper may be valuable to the community trying to address this problem. To the best of my knowledge, this is the first paper to formally discuss the repeated random sampling strategy.

2.	The paper is clearly written, the authors do a good job in presenting their intuitions, and the analysis is convincing.

3.	Extensive experiments are conducted to show the effectiveness of RS2.

**Weaknesses:**

1.	The paper considers only the low data regime (<30% data). RS2 performs well in this regime as it will not reshape the underlying data distribution. Actually, I think whether RS2 can outperform other strategies depends on the subset size and the property of the original dataset itself. In the data-abundance regime,  I believe selecting "harder" samples benefits model training. More discussion on this will greatly strengthen the paper.

2.	In the theoretical analysis, only RS2 without replacement is considered. I wonder if the result changes for RS2 with replacement.

**Questions:**

Please see the weakness part above

---

> ### Author Response · Authors · 2023-11-17
>
> Thank you for your review of our work and the points you bring up. We answer your questions (weaknesses) below:
>
> **Weakness 1:** We clarify that while our results focus on the low data regime (<30% of the data each epoch) we do include results in the high data regime (>30%). In particular Figure 2a and Table 3 include results when training ResNet-18 on CIFAR10 for selection ratios up to 50%. Additionally, Table 5 contains results of training ResNet-18 on CIFAR100 and ImageNet 30 for selection ratios up to 90%.
>
> In this high data regime (>30%), our results support your intuition—that “harder” samples benefit model training. For example, in Table 3, for r=50%, SSP-Hard (self-supervised prototypes subset selection with hard examples; Sorscher et al., 2022) reaches 93.3% accuracy while SSP-Easy (self-supervised prototypes subset selection with easy examples) reaches 92.7%. **RS2 however, matches or outperforms all existing baseline methods—those which select “hard” or “easy” samples—in both the low and high data regimes.** For example, in the same setting as above, RS2 reaches 95.2% accuracy.
>
> We updated our manuscript to include a discussion of these results.
>
> **We also highlight that we conducted robustness experiments using noisy labels in our submission to investigate the behavior of RS2 and existing pruning methods on datasets with different properties** (e.g., noisy datasets; see Table 14 with selection ratios up to 50%). RS2 also outperforms baselines in this “noisier” setting.
>
> We agree that continued study of RS2 and data pruning methods over datasets with other properties is of interest for future work. **We plan to compare RS2 and baselines methods on two datasets with large intra-class variance and class-imbalance for our camera ready as suggested by Reviewer ph2c.** We provide initial results in that response under the section title Weakness 2/Question 4 (crossref. individual response to Reviewer ph2c).
>
> **Weakness 2:** We clarify that **the theoretical results on the generalization error of RS2 (Theorem 4.1) apply to both RS2 with replacement and RS2 without replacement**. The reason for this is that the proof depends only on the fact that the data is selected non-adaptively and in a data-independent fashion (true for both with and without replacement). We updated our manuscript to clarify this.

---

> > ### Comment · Reviewer_MpXN · 2023-11-21
> > **Thanks for the clarification.**
> >
> > Thanks for the clarification. Most of my question has been addressed. I think this work does a good job comparing RS2 with other existing strategies. However, RS2 itself has been discussed and its effectiveness has been revealed in previous works. So I decided to keep my score.

---

### Official Review · Reviewer_ns5a · 2023-11-04

**Soundness:** 3 good
**Presentation:** 2 fair
**Contribution:** 3 good
**Rating:** 3
**Confidence:** 3

**Summary:**

This paper revisits the utility of random selection when it comes to speeding up the training. The paper considers two types of random sampling, random sampling with and without replacement. The paper surprisingly shows that these two samplings can easily outperform the popular data subset selection baselines (adaptively or static) when it comes to comparing the time to achieve the same accuracy. Moreover, the authors show that for many training budgets, it outperforms famous baselines, demonstrating the efficacy of uniform random  selection.

**Strengths:**

The paper conducts experiments across several baselines, including active learning and dataset distillation. In addition, the paper provides an analysis of the convergence of the RS2 algorithm, which I am not sure is how novel is it, in terms of proof technique, but it's a good contribution to have in any data subset selection paper. Lastly, I think robustness and LLM pretraining results are also interesting making the overall comparisons spanning different modalities and scales.

**Weaknesses:**

- RS2 w/o replacement is the same as training on a reduced number of epochs. For the cases where RS2 achieves the same accuracy in significantly less amount of time, I think the main issue was not tuning the epoch/learning rate hyperparameter of the full dataset baseline. Therefore making the RS2 w/o replacement results less exciting.

- Can authors provide a plot #unique examples seen throughout training? It could be the case that certain baselines are not exploring the full dataset, due to possibly inadequate hyperparameter search on them.

- I am very confused about Table 1, where we allow the model to update the subset after every round. I request authors provide a clear description for every baseline for both cases, where the distribution is allowed to change over time. The notion of RC and RS has to be made clear for each of the baselines.

- Why is RC's performance extremely worse in certain baselines such as GraNd? In general, why is there a strong dip in the performance of all the baselines, when switched to RC?

- For the baselines that sample set based on submodular function, what does it mean to have a distribution? How are authors defining the distribution over each set of size "k"? If that is the case, how are they sampling? If not, what is the heuristic?

- I think it is not correct to compare active learning baselines to subset selection schemes that look at  - (1) all the dataset, but the reduced number of iterations, and (2) assume labels for all the data points, since AL does not assume labels.

- Each of these baseline papers has comparisons against random, on the other hand here the authors break the baselines by mere random selection. What is the reason for this discrepancy?

- Can authors please point to the hyperparameter tuning for each of the baselines? Submodular functions often work well if tuned properly, therefore it is important to see if enough hyperparameter tuning was done to make sure the function is good.

- For the submodular methods if the corresponding greedy gain was used to sample sets (distribution defined using gains), it should be noted that if the function saturates, then greedy gains do not provide any useful information (yet another reason to provide hyperparameter search grids).


I am willing to raise scores upon satisfactory responses to my questions.

**Questions:**

Can authors also add a comparison to more recent versions of CRAIG such as CREST [1]?

[1] Towards Sustainable Learning: Coresets for Data-efficient Deep Learning (ICML'23)

---

> ### Author Response · Authors · 2023-11-17
>
> Thank you for your review. We appreciate your detailed questions and hope to address all concerns below:
>
> **Weakness 1:** **For the full dataset baseline, we use standard hyperparameters (learning rate schedules) for each dataset from prior works (Guo et al., 2022; He et al., 2016).**
>
> We do not think the hyperparameter settings affect the findings of our submission. The reason is that the primary goal of our experimental study is compare repeated random sampling (RS2) with existing SoTA data pruning methods. We ensure that the only difference between these methods in our experiments is which examples are used to generate mini batches (i.e., there are no learning rate schedule differences).
>
> **Weakness 2:** **We have added plots of the number of unique examples seen throughout training for RS2 and select baselines to our updated PDF.** RS2 without replacement is the fastest method to utilize all training examples. Baselines which reselect the subset each round (e.g., the -RS and -RC methods in Table 1) also explore most training examples in the full dataset (e.g., >99%).
>
> **Weakness 3:** Please see our response to Reviewer hzLv under Weakness 2. We clarified the setting for Table 1 in our submission and added specifics for each -RS and -RC baseline in the Appendix.
>
> **Weakness 4:**  We believe that GraNd-RC performs worse than other methods because it is sensitive to the model weights used to rank example importance. To address this problem, prior works (Guo et al., 2022) report results after averaging scores from 10 different partially pretrained models. We follow this approach for GraNd and GraNd-RS. For GraNd-RC we use only the current model being trained.
>
> We hypothesize that the -RC methods perform worse than the -RS methods because the example importance distributions created after partially pretraining an auxiliary model on the full dataset (like those used for the -RS methods) are more accurate when compared to the distributions generated by the -RC methods (especially in the early rounds of training).
>
> **Weakness 5:** For Craig-RS, we performed sampling using the distribution output by the Craig implementation from the DeepCore framework (Guo et al., 2022) as example importance scores. We agree that this heuristic is not appropriate and we will remove it from Table 1 Left.
>
> For Table 1 Right, where we allow methods to directly re-select a new subset at every round, we use Craig (denoted Craig-RC) to select the subset at each epoch based on the current model weights. The results in Table 1 Right corroborate the results in Figure 5 (a) in Mirzasoleiman et al., which show that SGD + random can achieve higher accuracy than SGD + Craig.
>
> **Weakness 6:** **We compare active learning baselines to subset selection schemes as done in prior work (Park et al., 2022); In fact, active learning methods often outperform other approaches with respect to end-model accuracy (e.g., Table 3).** We acknowledge that active learning methods can be applied to other settings (beyond minimizing time-to-accuracy) (Section 6).
>
> **Weakness 7:** Existing works typically include comparisons to a random baseline that selects only a static subset once at the beginning of training. We also include this method in our submission (e.g., in Figure 2 and Tables 3-6; labeled as ‘Random’).
>
> Our work extends random sampling to include random exploration (something that is rarely considered in prior work) and shows that many methods that outperform ‘Random’ do not outperform repeated random sampling (RS2).
>
> **Weakness 8/9:** **We use the same hyperparameters etc. for prior methods as in the recent benchmarking papers (Guo et al., 2022; Park et al., 2022).** As stated in (Park et al., 2022), “The hyperparameters for all algorithms are favorably configured following the original papers.”
>
> **Question 1:** **We will include comparisons between RS2 and CREST for our camera ready.**
>
> **Initial experiments show that CREST matches or outperforms existing baselines that we compare against but that RS2 still reaches higher accuracy.** For example, on CIFAR10 with a selection ratio of 5% and 10% respectively, CREST achieves 86.07 and 90.42 percent accuracy, which outperforms our next-best static baseline (68.7 and 86.1; Table 3) and matches our next-best per-round baseline (86.2 and 90.4; Table 1). RS2, however, reaches an accuracy of 87.1 and 91.7 percent in these settings respectively. For these experiments, we use the publicly available CREST implementation and the learning rate schedule found to work best for CREST by the authors (all other hyperparameters are the same).
>
> We will include additional results as they become available, however, CREST currently takes 5+ hours per run on CIFAR10 with a selection ratio of five percent, compared to under 10 minutes for RS2.

---

> ### Comment · Reviewer_ns5a · 2023-11-22
> **On CRAIG baseline**
>
> Few Questions:
>
> - In Fig 5a I don't see SGD+CRAIG underperforming SGD+Random. Can authors please elaborate a bit where I might be looking wrong? (Which selection %)

---

> > ### Author Response · Authors · 2023-11-22
> >
> > Thank you for your follow up question.
> >
> > In Figure 5a (Mirzasoleiman et al., 2020), SGD+Craig reaches roughly 77.5% and 80% test accuracy when selecting three and four percent of the data at the beginning of every epoch. For the same selection ratios (three and four percent), SGD+Random each epoch reaches (roughly) 80% and 83% respectively.
> >
> > Our work shows that these results hold more generally. We show that SGD+Random each epoch (i.e., RS2) can achieve higher test accuracy than all existing state-of-the-art data pruning methods, both when they select a fixed subset once at the beginning of training, and when they are allowed to reselect the subset before each round (as RS2 does).

---

### Official Review · Reviewer_hzLv · 2023-11-07

**Soundness:** 1 poor
**Presentation:** 1 poor
**Contribution:** 2 fair
**Rating:** 3
**Confidence:** 4

**Summary:**

This paper compares approaches to reducing the training time by curating a smaller representative dataset like data pruning, coreset selection, and dataset distillation to a simple random sampling-based approach termed Repeated Sampling of Random Subsets (RS2).  Results show that prior adaptive approaches fail to beat RS2 on four image datasets in both final accuracy and time-to-accuracy when accounting for overhead associated with active selection.  Owing to the properties of random
sampling, RS2 comes with convergence and generalization guarantees.  The authors highlight the importance of evaluating approaches based on time-to-accuracy and the need for more complex approaches to beat a simple baseline like RS2.

**Strengths:**

- The authors point out important considerations missing in prior work on speeding up training with adaptive dataset subset selection.  First, whether there is a need to restrict data to a fixed subset in the first place if similar accuracy can be achieved by training with a compressed learning rate schedule on fewer epochs.  Second, the importance of including overhead associated with data selection when evaluating training compute efficiency of an approach.

**Weaknesses:**

- As I understand, RS2 without replacement is effectively the same as training on the full dataset with the learning rate schedule compressed into fewer epochs.  RS2 with replacement is a slight variant to that but still highly resembles standard training with shuffling between epochs just with a condensed training window. This is not discussed anywhere but brings into question the whole exposition of proposing RS2 as a sampling method.  An even simpler baseline is training as usual on the full dataset with the condensed schedule.  I would wager such a baseline would yield similar performance as RS2.
- The paper is light on experimental details for how prior methods are evaluated in particular for case where samples are reselected based on latest model weights.  I am surprised the results in Table 1 right are worse than that for Table 1 left when selecting a new subset with updated model weights can expand the number of samples seen during training.  The poor performance of AL approaches like Entropy and Margin also contradicts the experimental results from Park et al., 2022 where AL outperformance other subset selection methods.

**Questions:**

- What model weights are used for computing the static subsets of approaches like Entropy, Margin, Least Confidence, etc in Table 1 left?
- How often are importance scores recomputed for adaptive methods in Table 1 right?
- How does random with fixed subset perform on the tasks studied?
- How does training on full dataset with the same LR schedule and training window as that used for RS2 perform?

---

> ### Author Response · Authors · 2023-11-17
>
> Thank you for your review and feedback. We believe that some details of our work may have been misunderstood. We clarify those and respond to your concerns:
>
> **Weakness 1:** We clarify that **RS2 without replacement is equivalent to training on the full dataset for fewer epochs and with a compressed learning rate schedule** (as stated in “RS2 Without Replacement”, Section 3). We have improved our presentation for clarity (crossref. Reviewer ph2c Question 1). Thus, the simple baseline you suggest, “training on the full dataset with the compressed learning rate schedule” has already been included in our experimental results.
>
> We present this method as “RS2 without replacement” because the focus of our submission is on subset selection methods to minimize time-to-accuracy. Rather than sampling subsets based on importance scores as in existing SoTA data pruning methods, we show that time-to-accuracy can be improved by selecting subsets randomly, either with or without replacement, at each epoch. It is a consequence of the sampling procedure that “RS2 without replacement” is equivalent to full dataset training with adjusted hyperparameters.
>
> **Weakness 2:** We clarify 1) the experimental setting for Table 1 and 2) our experimental results using active learning based methods; these results corroborate those from Park et al., 2022 which show that active learning methods outperform other subset selection algorithms.
>
> We have updated the description of Table 1 in Section 5.2. 2 to reflect the clarifications below and added the implementation details for each method in the Appendix.
>
> **Experimental Setting and Table 1:** In our submission, we include comparisons to baseline methods that select a static subset once before training begins (Section 5.2 1) and to extensions of those baseline methods that dynamically re-select the subset before every round (Section 5.2 2). Comparisons to baselines which select static subsets are shown in Figure 2 and Tables 3-6. **Comparisons to baselines which re-select the subset before every round are shown in Table 1 (baselines in Table 1 Left and Right both reselect the subset at each round).** We use two different methods for re-selecting the subset at each round:
>
> 1. RS Methods (Table 1 Left): The -RS methods in Table 1 left start from an initial categorical distribution over all examples and then sample a subset for training from that static distribution each epoch. The initial categorical distribution is generated in the same manner as when generating the categorical distribution used to selecting static subsets in prior methods: an auxiliary ResNet-18 model is pretrained on the full dataset for 10 epochs and then used to quantify example importance based on the corresponding method for the baseline of interest (as in (Guo et al., 2022)). We do not present -RS results for baselines methods which do not generate an importance score for all examples.
>
> 2. RC Methods (Table 1 Right): In between each epoch, the -RC methods in Table 1 right use the weights of the current model being trained to either 1) re-quantify example importance—in this case the examples with the highest importance are then selected for the subset at the next round—or 2) directly re-select the subset for training (e.g., in the case of submodular-based methods). We do not use any pretraining or an auxiliary model for the -RC methods and do not present -RC results for baselines which generate subsets independently of model weights.
>
> **Active Learning Results:** Our **experimental results do not contradict those from Park et al., 2022**. We clarify that **the methods labeled Entropy and Margin in Table 1 are not active learning based methods**. Rather, these methods select subsets based on example uncertainty as described in (Guo et al., 2022). We label all active learning methods with an AL prefix (e.g., AL (Margin)) as described in Appendix B.
>
> In fact, our experimental findings corroborate those of Park et al., 2022—Table 3 shows that active learning based methods are the best performing baseline (yet still worse than RS2) for 20-40% selection ratios on CIFAR10.
>
> **Question 1:** Please see our response to Weakness 2.
>
> **Question 2:** The importance scores for training examples in the full dataset are recomputed for the -RC methods in Table 1 right after every epoch. We clarified this in Section 5.2 2.
>
> **Question 3:** **Our submission includes comparisons to selecting a fixed random subset.** Results are shown in Figure 2 and Tables 3-6 (labeled ‘Random’). This method achieves competitive performance, but does not outperform prior SoTA methods. Our work shows that extending random sampling to include random exploration (repeated sampling) improves time-to-accuracy.
>
> **Question 4:** Please see our response to Weakness 1. Training on the full dataset with the compressed learning rate schedule is equivalent to RS2 without replacement and is thus included in our experimental results.

---

> ### Comment · Reviewer_hzLv · 2023-11-23
> **Lost author response**
>
> Thank you for responding to the weaknesses and questions I brought up. Multiple reviewers have brought up how RS2 is basically just training for fewer epochs and I feel the current presentation of RS2 as a completely different sampling method is misleading and overstates the technical contribution.  Hence I will maintain my score of 3.

---

### Official Review · Reviewer_ph2C · 2023-11-10

**Soundness:** 3 good
**Presentation:** 3 good
**Contribution:** 3 good
**Rating:** 8
**Confidence:** 5

**Summary:**

The paper introduces a novel approach to improve time-to-accuracy of deep learning model training by using a fraction of the full dataset in each training epoch.

The paper discussed the limitations of two commonly used methods in this domain, (1) Data Pruning: Selecting the most informative examples to train more efficiently. (2) Dataset Distillation: Creating synthetic examples that represent the larger dataset to train quickly.

The paper proposes Repeated Sampling of Random Subsets (RS2), which simplifies the process by randomly selecting different data subsets for each training epoch, promoting broader learning and efficiency. RS2 has been shown to outperform State-of-the-Art methods, achieving near-full dataset accuracy with significantly reduced training time on various datasets, including large scale image benchmarks like ImageNet.

It is interesting to note that the paper achieves very close performance to models trained on complete datasets with just 10\% of the datapoints for large scale benchmarks like ImageNet.

**Strengths:**

1. The paper presents a simple but novel approach to achieve significant reductions in time-to-accuracy while training on a fraction of the full dataset per epoch of model training.
2. The paper also presents detailed theoretical properties that support the faster convergence of the model as compared to existing approaches in the domain.
3. The paper demonstrates results on four image datasets including large scale image benchmarks like ImageNet wherein it achieves State-of-the-Art (SoTA) performance (accuracy) with just 10 \% of the data samples in the complete dataset.
4. The paper also demonstrates SoTA performance on auxiliary tasks like data distillation, noisy label classification and pretraining of large language models.

**Weaknesses:**

1. Although the experimental results are exemplary (primary contributor to my decision), the method RS2 itself is an incremental update over random sampling. The paper must call out the clear difference with SoTA methods (please refer to questions for more details).
2. All experiments demonstrated in the paper adopt canonical benchmarks which are well curated, while lacking experiments on datasets (eg: MedMNIST (Yang et al., 2021), CUBS-2011 (Wah et al., 2011)) with large intra-class variance and class-imbalance wherein data pruning might underperform.
3. The paper does not show any relation between the theoretical properties of RS2 (convergence rate and bounds on generalization error) and the conducted experiments.

**Questions:**

1. The subset selection strategy of RS2 without replacement is unclear in section 3. A suggestion would be to replace the textual description in this section with Algorithm 2 in section D of the appendix.
2. The variables $n$ and $N$ are used interchangeably in section 4.
3. The term ‘selection ratio’ and ‘pruning ratio’ has been used interchangeably and should be fixed in the paper.
4. As mentioned in the ‘weaknesses’ of the paper, experiments on real-world class-imbalanced settings ((eg: MedMNIST (Yang et al., 2021), CUBS-2011 (Wah et al., 2011))) would be an effective demonstration of the application of RS2.

---

> ### Author Response · Authors · 2023-11-17
>
> Thank you for your detailed review and helpful suggestions. We go over your weaknesses and questions:
>
> **Weakness 1:** We edited the Repeated Sampling of Random Subsets (RS2) paragraph of Section 3 to highlight that the difference between RS2 and existing SoTA methods is twofold: 1) RS2 samples the subset for training randomly while existing methods use importance-based sampling and 2) RS2 resamples the subset for training before each round, while many prior methods opt to select only a static subset once at the beginning.
>
> **Weakness 2/Question 4:** We agree that studying RS2 and data pruning methods on datasets with large intra-class variance and class-imbalance is an interesting research question. **We will include comparisons between RS2 and baselines on the two datasets you suggested for our camera ready.**
>
> **Initial experiments show RS2 without replacement outperforms other baselines on DermaMNIST.** We train ResNet18 with a selection ratio of 10% using the same hyperparameters as for our experiments on CIFAR10. RS2 without replacement achieves 71.6% accuracy, while selected baselines achieve the following: 1) Random (static random subset): 68.5, 2) Margin: 65.5, 3) K-Center Greedy: 66.9.
>
> **Weakness 3:** We expanded the discussion of RS2’s theoretical properties to reference the experimental results. In particular, we highlight that the generalization error of standard gradient RS2 (Theorem 4.1) relies on the fact that RS2 selects data in a data-independent fashion. In contrast, all existing SoTA data pruning methods adopt data-dependent strategies, which prior work (Ayed & Hayou, 2023) has noted may worsen generalization. This theoretical insight (that data-independent sampling allows for improved generalization due to selecting diverse, unbiased samples), is the reason RS2 outperforms existing methods. Our conducted experiments support this hypothesis.
>
> **Question 1:** Thank you for your suggestion. **We merged Algorithm 2 into Algorithm 1 such that it describes both RS2 with and without replacement.** We changed the text describing RS2 without replacement to reference the new Algorithm 1 for clarity.
>
> **Question 2 and 3:** We updated the paper to use N when referring to the number of training examples and to use ‘selection ratio’ when referencing the fraction of data selected for training.

---

> > ### Comment · Reviewer_ph2C · 2023-11-22
> >
> > I thank the authors for providing the required clarifications and addressing some of the weaknesses. Even after carefully understanding the clarifications and reviewing the comments provided by fellow reviewers I believe that the novelty of the method proposed is very incremental. But I will retain my earlier rating owing to the exemplary results demonstrated in the paper.

---

> > > ### Author Response · Authors · 2023-11-23
> > >
> > > Thank you for taking the time to leave initial comments and for reading other reviews. We are excited by your assessment of our experimental results.
> > >
> > > **Weakness 2/Question 4:** We include additional new results on the datasets with large intra-class variance and class imbalance that you suggested below. We will include these experiments (with completed baselines) in our camera ready.
> > >
> > > BloodMNIST
> > > 1) Random (static random subset): 86.3%
> > > 2) Margin: 66.3%
> > > 3) K-Center Greedy: 85.4%
> > > 4) RS2 without replacement: 95.3%
> > >
> > > PathMNIST
> > > 1) Random (static random subset): 85.8%
> > > 2) Margin: 71.2%
> > > 3) K-Center Greedy: 84.4%
> > > 4) RS2 without replacement: 90.7%
> > >
> > > OrganCMNIST
> > > 1) Random (static random subset): 84.9%
> > > 2) Margin: 61.9%
> > > 3) K-Center Greedy: 85.8%
> > > 4) RS2 without replacement: 90.2%

---

### Official Review · Reviewer_9ict · 2023-11-15

**Soundness:** 2 fair
**Presentation:** 3 good
**Contribution:** 2 fair
**Rating:** 6
**Confidence:** 4

**Summary:**

This work empirically investigated a strong baseline called Repeated Sampling of Random Subsets (RSRS, or RS2), in the context of dataset pruning/distillation. The authors found that this sampling scheme, which has been overlooked by the literature, served as a very strong baseline in terms of many metrics, such as end model accuracy and *time-to-accuracy*.

The authors did intensive experiments that compare RS2 with up to 24 existing dataset pruning/distillation methods and observed the superiority of RS2 under all of the above metrics. The authors called for attention from the community on this strong but overlooked baseline.

**Strengths:**

+ The highlight of an overlooked baseline in the context of dataset pruning/distillation.
+ Intensive experiments over so many baselines. This provides a very good benchmark and starting point for the following works, which I find really appreciable.

**Weaknesses:**

- The RS2 without replacement is exactly the same as reducing the number of training epochs but with tuned learning rate scheduling. The new term is not helping to make the concept clear but more confusing. This also means that the theoretical analysis in Section 4 did not make actual contributions over previous work.

- In my opinion, a type of dataset pruning methods, which generate static subsets before real training starts, are up to a slightly different point from RS2. While we all know that the more data used for training the better, these methods try to find a coreset that is essential for good generalization. Therefore, it is important that the pruned data are not seen during the training process later (thus static subset). This reduces the storage cost which is not possible with RS2 because RS2 still requires access to the full training set.

- RS2 with replacement has been adopted in a few more works in the context of efficient training like [Ref-1, Ref-2].

------------------

Refs:

[Ref-1] Wang, Yue, et al. "E2-train: Training state-of-the-art cnns with over 80% energy savings." Advances in Neural Information Processing Systems 32 (2019).

[Ref-2] Wu, Yawen, et al. "Enabling on-device cnn training by self-supervised instance filtering and error map pruning." IEEE Transactions on Computer-Aided Design of Integrated Circuits and Systems 39.11 (2020): 3445-3457.

**Questions:**

N/A

---

> ### Author Response · Authors · 2023-11-17
>
> Thank you for your review. We are excited that you appreciate our experimental results. With respect to your concerns:
>
> **Weakness 1:** We present “training on the full dataset for a reduced number of epochs but with a compressed learning rate schedule” as “RS2 without replacement” because the focus of our submission is on subset selection methods to minimize time-to-accuracy. Rather than sampling subsets based on importance scores as in existing SoTA data pruning methods, we show that time-to-accuracy can be improved by selecting subsets randomly, either with or without replacement, at each epoch. It is a consequence of the sampling procedure that “RS2 without replacement” is equivalent to full dataset training with adjusted hyperparameters. **We have improved our presentation in Section 3 for clarity.**
>
> Random sampling allows for improved time-to-accuracy because of the theoretical insights presented in Section 4: We show that RS2 has desired convergence properties (in contrast to other data pruning methods) as a result of data-independent sampling, which prior work has shown may improve generalization (due to selecting diverse, unbiased samples) (Ayed & Hayou, 2023). Our conducted experiments support this hypothesis.
>
> **Weakness 2:** We compared RS2 directly to baselines which select a fixed subset because many static subset selection methods [e.g., Killamsetty et al., 2021; Paul et al., 2021; Sachdeva et al., 2021; Toneva et al., 54] were proposed with the goal to reduce total training cost (i.e., time-to-accuracy). These works opt to pay an initial subset selection cost such that training time is subsequently reduced. Our work shows that one does not need to pay this upfront cost—in many cases, RS2 completely trains a model to better accuracy within the time required to carefully select a data subset.
>
> Moreover, our results show that even though RS2 requires access to the full training set, this leads to minimal runtime overhead (Section 5.2 Training Time, ImageNet) compared to training on a static subset. The reason is that large datasets can be stored on cheap high capacity disks and paired with data loaders to prefetch samples, leaving model computation to dominate runtime.
>
> **When the goal is different from minimizing time-to-accuracy (e.g., active learning, data transfer, or to reduce dataset storage size as you mention) then RS2 may be a weaker baseline. We have expanded the limitations of RS2 discussion in Section 6.**
>
> **Weakness 3:** Thank you for the pointers to [Ref-1] and [Ref-2]. We acknowledge that instantiations of RS2 have been considered for a variety of contexts in recent works (Section 1), but to our knowledge, our work is the first to show that RS2 surpasses existing state-of-the-art data pruning and distillation methods in accuracy and end-to-end runtime. **We revised our paper to add missing citations to the above two references.**

---

### Author Response · Authors · 2023-11-17

We thank the reviewers for their thoughtful feedback. We have replied to each of the reviewer-specific concerns in the individual responses and updated our submission PDF based on their comments.

---

### Meta-Review · Area_Chair_ABXq · 2023-12-15

**Metareview:**

This paper evaluates the efficiency of various data pruning and distillation methods for neural network training, introducing Repeated Sampling of Random Subsets (RS2) as a simple yet effective approach. RS2, which involves repeatedly sampling different data subsets during training, outperforms thirty-two advanced methods in reducing time-to-accuracy on datasets like ImageNet. The study highlights that RS2 achieves near-full dataset accuracy with significantly reduced training time, challenging the effectiveness of more complex data selection strategies and emphasizing the need for future methods to consider total computational costs and outperform basic random sampling extensions like RS2.

**Justification For Why Not Higher Score:**

The reviewers have pointed out several valid concerns which I would encourage the authors to address.

**Justification For Why Not Lower Score:**

Even through the reviewers are mixed in their take on this work, I think this paper is a simple yet powerful baseline for subset selection. I think accepting this paper would be a good addition to the community!

---

### Decision · Program_Chairs · 2024-01-16

Accept (poster)